# ADVANCING MULTIMODAL FUSION ON HETEROGENEOUS DATA WITH PHYSICS-INSPIRED ATTENTION

## ABSTRACT

Multimodal fusion learning (`MFL`) paradigm (a framework to jointly learn from heterogeneous data sources) has shown great potential in various fields such as Medicine, Science, Engineering, etc. It is extremely desirable in the medical domain, where we are faced with disparate data modalities such as imaging, clinical records, and omics. However, existing `MFL` strategies face several major challenges. First, they struggle to capture complex cross-modal interactions effectively. Second, they are often designed and evaluated for narrow, fixed modality configurations (e.g., imaging-only, or specific pairs such as image and omics or image and clinical text), which limits evidence of their adaptability and generalizability to broader collections of heterogeneous medical modalities. Finally, they incur high computational costs, restricting their applicability in resource-constrained healthcare `AI`. To address these challenges, we propose a novel `MFL` framework – **E**fficient **H**ybrid-fusion **P**hysics-inspired **A**ttention **L**earning **Net**work (`EHPAL-Net`) – a lightweight and scalable framework that integrates various modalities through novel Efficient Hybrid Fusion (`EHF`) layers. Each `EHF` layer captures rich modality-specific multi-scale spatial information, followed by a Physics-inspired Cross-modal Fusion Attention module to model fine-grained, structure-preserving cross-modal interactions, thereby learning robust complementary shared representations. Furthermore, `EHF` layers are sequentially learned for each modality, making them adaptable and generalizable. Extensive evaluations on 15 public datasets show that `EHPAL-Net` outperforms leading multimodal fusion methods, boosting performance by up to $3.97\%$ and lowering computational costs by up to $87.8\%$, ensuring more effective and reliable predictions. Our code is available at: `https://github.com/Submission-01/EHPAL`.

## 1 INTRODUCTION

Multimodal fusion learning (`MFL`) remains a key challenge in machine learning. It is of great importance in healthcare research as there is a strong need to integrate heterogeneous data of disparate modalities, such as radiology, dermoscopy, multi-omics, and electronic health records (`EHRs`). The goal in `MFL` is to capture shared representations for various modalities while preserving modality-specific structural information (Steyaert et al., 2023; Hemker et al., 2024). Although `MFL` methods exist for modalities that share semantic spaces, such as audio-visual text tasks (Goyal et al., 2016), image captioning (Yu et al., 2019), multimodal dialogue (Liang et al., 2022), etc. – healthcare data is far more diverse and complex (Hemker et al., 2024). This is because cross-modal relationships in medical contexts are often opaque, and modalities do not share semantics. As a result, learning robust shared representations across such modalities remains an open research problem.

Hybrid of early [1] and intermediate [2] fusion methods (Hybrid Early Fusion Layer in Fig. 1(B)) has recently emerged as a promising new approach for `MFL`. However, broader adoption of hybrid multi-

---

[1] Early fusion methods combine raw inputs (e.g., via concatenation or Kronecker products) at the model's input, allowing end-to-end training but risking structural information loss and feature dimensionality explosion (Chen et al., 2020). Unlike, Late fusion methods preserve modality-specific structures by training separate sub-models and aggregating their outputs, but this incurs high computational cost and lacks fine-grained cross-modal interaction (Liang et al., 2022).

[2] Intermediate fusion methods aim to learn shared representations at the feature level. They conflate modality-specific and modality-shared semantics. This strategy is, however, computationally expensive (Cui et al., 2023a).

Figure 1: Comparative high-level overview of various `MFL` methods – **(A)** Architecture of intermediate or late fusion methods (`DRIFA-Net`, `MuMu`, `MOTCAT`, etc.) **(B)** Architecture of hybrid early and intermediate fusion methods (`HEALNet` ). **(C)** Architecture of our proposed `EHPAL-Net`.

modal methods in `AI`-driven healthcare faces three main challenges – **performance, generalization and efficiency**. The performance issue mainly stem from the fact that existing architectures of hybrid methods lack an effective fusion mechanism – e.g., they do not have capacity for representational diversification to retain modality-specific context before fusion; they do not have specialized attention modules to preserve modality-specific structural details and model complex cross-modal interactions; and they cannot reintroduce high-level, modality-specific semantics after learning a shared representation (Challenge 1). The efficiency issue stems from the fact that existing hybrid methods rely on large, high-dimensional attention matrices during intermediate fusion. This imposes significant computational overhead, limiting the applicability of hybrid methods in resource-constrained medical `AI` environments (Challenge 2). The generalizability concern stems from the fact that existing hybrid models are often restricted in their ability to scale across diverse medical data sources, as they are mostly trained on limited data modalities (Challenge 3). Existing works do not have the capability to leverage heterogeneous medical data during a single training time due to increased computational overhead. Ideally, one should adopt a strategy to leverage heterogeneous medical modalities data during a single training cycle to reduce computational cost and to ensure better generalization. So, in a way, the challenge of generalization is related to that of performance and efficiency.

To address these concerns, we propose **E**fficient **H**ybrid-fusion **P**hysics-inspired **A**ttention **L**earning **Net**work (`EHPAL-Net`) – a novel framework of multimodal fusion learning to *enable efficient, generalizable and scalable integration in resource-constrained `AI`-driven healthcare applications*. The main component of `EHPAL-Net` is **E**fficient **H**ybrid **F**usion (`EHF`) layer that exploits hybrid intermediate and late fusion strategies – effectively balancing optimal performance with minimal computational cost. This design facilitates the learning of robust complementary shared representations and ensures scalability across any number of heterogeneous modalities[3] for improved generalization. `EHPAL-Net` is a single-pass non-iterative multimodal fusion pipeline that processes modalities sequentially and incrementally fuses them via successive `EHF` layers (Fig. 1 (C)). Unlike prior methods (Hemker et al., 2024; Dhar et al., 2025; Islam & Iqbal, 2020; 2022) requiring additional encoders for iterative refinement, which result in high-cost architectures (as shown in Fig. 1 (A–B)), our approach eliminates these computational bottlenecks. In our proposed model, two modalities are first processed by an `EHF` layer, which captures multi-scale spatial information to learn diversity in representations. The representations are further refined in an `EHF` layer using a **physics-inspired attention mechanism**, that models fine-grained cross-modal interactions while preserving the complex structural properties of each modality. Finally, a late-fusion strategy is used that integrates the refined outputs into a shared representation – thereby learning more expressive multimodal information. This information (shared representation), along with the next input modality, is then fed into the subsequent `EHF` layer. This process is repeated until all input modalities are processed. The main contributions of our work are as follows:

---

[3]Definition of modality and setting. Throughout this paper we adopt a slightly broader, dataset-level notion of *modality*. A modality is an input stream corresponding to a particular data-generating process and dataset (e.g., dermoscopy images, Pap-smear cytology, colorectal histology, multi-omics profiles, or tabular `EHR` time series). Different datasets—even when all are "images"—are treated as distinct modalities because they arise from different acquisition protocols and have different label spaces. Importantly, `EHPAL-Net` does not require modalities to be paired at the patient level; in our experiments, modalities are unpaired heterogeneous datasets (Yang et al., 2022; Dou et al., 2020; Valindria et al., 2018) and the `EHPAL-Net` is trained in a multi-task fashion.

- We introduce `EHPAL-Net`, a novel light-weight end-to-end `MFL` framework that addresses the challenges of performance, generalizability, and efficiency in existing frameworks – making it well-suited for `AI`-driven healthcare applications with many modalities in low-resource environments.
- We have proposed a novel `EHF` layer that is based on a physics-inspired attention mechanism utilizing an intermediate fusion strategy by integrating hyperbolic and quantum-inspired embeddings. As a result, it learn much better spatial and frequency-domain features, preserves modality-specific structural details, and captures complementary cross-modal dependencies.
- We conduct extensive evaluations against state-of-the-art methods across *fifteen heterogeneous medical datasets*, demonstrating significant performance improvements.

## 2 RELATED WORKS

We focus on multimodal learning problems in biomedical data, where the modalities are structurally heterogeneous – i.e., using imaging modalities (e.g., dermoscopy, Pap smear) along with tabular data (e.g., multi-omics and clinical records). This is in contrast with existing works that are limited to homogeneous modalities only. While multimodal fusion learning strategies based on early, late, intermediate, and

Table 1: Characteristics of existing `MFL` methods.

| Model | Shared Representations | Cost-Effective | General-ization | Reli-ability |
|---|:---:|:---:|:---:|:---:|
| `Perceiver` (Jaegle et al., 2021) | ✓ | | | |
| `GLORIA` (Huang et al., 2021) | ✓ | | | |
| `HAMLET` (Islam & Iqbal, 2020) | ✓ | | | |
| `MuMu` (Islam & Iqbal, 2022) | ✓ | | | |
| `M³Att` (Liu et al., 2023) | ✓ | | | |
| `MOTCAT` (Xu & Chen, 2023) | ✓ | | | |
| `DRIFA-Net` (Dhar et al., 2025) | ✓ | | ✓ | ✓ |
| `HEALNet` (Hemker et al., 2024) | ✓ | | | |
| `EHPAL-Net` (Ours) | ✓ | ✓ | ✓ | ✓ |

hybrid methods have driven significant research for homogeneous modalities (e.g., natural vision tasks (Abdelhalim et al., 2021; Peng et al., 2022; Wang et al., 2020a;b; Joze et al., 2020; Ma et al., 2021)) , their adoption for structurally heterogeneous modalities (e.g., biomedical data) remains limited (Cheng et al., 2022). As we discussed earlier, each strategy of multimodal learning has their own strengths and weaknesses. E.g., early fusion methods, such as `Perceiver` (Jaegle et al., 2021), naïvely concatenate raw inputs and apply iterative self and cross-attention layers for end-to-end learning. However, such naïve fusion dilutes the modality-specific signals and increases feature dimensionality (Wang et al., 2020b). Late fusion methods (e.g., `MTTU-Net` (Cheng et al., 2022), `GLORIA` (Huang et al., 2021), etc.) encode each modality independently – combining `CNNs` and transformers for glioma segmentation or aligning images with radiology reports via global local attention. Its variants like `HAMLET` (Islam & Iqbal, 2020) and `MuMu` (Islam & Iqbal, 2022) include multi-head self-attention for richer context learning. Despite preserving modality-specific structure, they fail to capture fine-grained cross-modal interactions, incur progressive information loss from cascaded attention layers, and remain computationally intensive due to heavy convolutional and attention operations. Intermediate fusion methods (e.g., `CAF` (He et al., 2023), `DRIFA-Net` (Dhar et al., 2025), `MOTCAT` (Xu & Chen, 2023), etc.), employ cross-modal, cascaded dual attention (or optimal transport co-attention) to fuse modalities within their attention module. However, their reliance on modality training constraints as well as high computational overhead, along with progressive information loss, limits their ability to learn robust complementary representations. Finally, hybrid models such as `HEALNet` (Hemker et al., 2024) bridge early and intermediate fusion through a hybrid early fusion layer (Fig. 1 (B)) to enrich shared representations. However, its reliance on high-dimensional attention matrices incurs high computational costs, limiting its capacity in low-resource healthcare `AI`. A summary of these techniques is presented in Table 1 (more details in A).

Several recent methods for clinical prognosis explicitly model the interplay between modality-specific and modality-shared representations. Pathomic Fusion (Chen et al., 2020) employs separate encoders for histology and genomics and fuses them via Kronecker products and gated attention, yielding distinct unimodal and pairwise interaction terms for cancer diagnosis and survival prediction. (Steyaert et al., 2023) and (Cui et al., 2023a) systematically evaluate early, intermediate, late and hybrid fusion strategies on `TCGA`-style cohorts, emphasizing that preserving modality-specific structure while learning a shared latent space is crucial for biomarker discovery. More recent architectures such as `CA-MLIF` (An et al., 2025) and LegoFuse (Hemker et al., 2025a) extend this idea: `CA-MLIF` interleaves cross-attention with low-rank interaction fusion to jointly refine per-modality features and shared risk signatures, while LegoFuse composes pre-trained unimodal encoders with lightweight fusion blocks to capture cross-modal dependencies at linear cost in the number of modalities.

Our Approach: `EHPAL-Net` overcomes these limitations with the following key innovations. First, it has a modular design that *scales seamlessly to any number of modalities* by adding a novel `EHF`

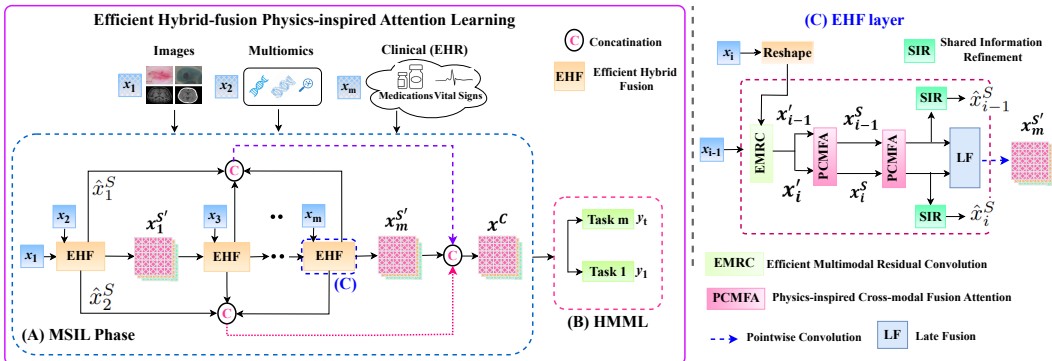

Figure 2: Overview of `EHPAL-Net` comprising of `MSIL` and `HMML` phases. **(A)** Overview of `MSIL` phase utilizing `EHF` layer. **(B)** Overview of `HMML` phase. **(C)** Layout of `EHF` layer.

layer, which captures *cross-modal interactions* while preserving each *modality's structural details*, thereby learning *robust complementary shared representations*. Secondly, it has a novel module that leverages hyperbolic and quantum attentions in parallel and fuses their outputs to prevent *progressive information loss* and capture *complex hierarchical relationships* across modalities. `EHPAL-Net` delivers improved *generalizability* across diverse medical modalities and more holistic explanations than existing state-of-the-art (`SOTA`) fusion methods.

## 3    PROPOSED METHOD

**Problem Formulation:**  `EHPAL-Net` enhances `MFL` through cascaded spatial- and frequency-domain information integration. Let $\mathcal{X} = \{x_i\}_{i=1}^{m}$ denote $m$ heterogeneous input modalities, where each $x_i \in \mathbb{R}^{H_i \times W_i \times C_i} \cup \mathbb{R}^{D_i}$ represents either imaging data (with spatial dimensions height $H_i$, width $W_i$, channels $C_i$) or non-imaging data (e.g., multi-omics or `EHR`) as $D_i$-dimensional vectors. Let $\mathcal{Y} = \{y_j\}_{j=1}^{t}$ denote $t$ multi-task labels. Notably, non-imaging vectors are reshaped into pseudo-image tensors in $\mathbb{R}^{H_i \times W_i \times C_i}$ to enable uniform fusion. Our aim is to learn a function $-\mathcal{F}(\cdot)$ to learn robust complementary shared representations: $x^C$ through the mapping $\mathcal{F} : \mathcal{X} \longrightarrow \mathcal{Y}$, optimizing both performance as well as computational efficiency.

In this work, each $x_i$ corresponds to one dataset-level modality stream. For example, we use dermoscopy, cytology, histology tiles, multi-omics feature vectors, and clinical time-series. Although the notation $\mathcal{X} = \{x_i\}_{i=1}^{m}$ suggests a common index set, in practice these modalities are unpaired: the underlying subjects and label spaces are disjoint across datasets. The `MSIL` phase therefore learns a shared representation $x^C$ across heterogeneous, unpaired modalities, while the `HMML` phase attaches task-specific heads to each modality.

**Method Overview:** `EHPAL-Net` differs from conventional `MFL` methods as it leverages a sequential multimodal integration pipeline, where modality-specific inputs are processed sequentially and incrementally fused through its novel **E**fficient **H**ybrid **F**usion (`EHF`) layer. `EHPAL-Net` comprises two salient phases: (1) **Multimodal Shared Information Learning** (`MSIL`) – a sequential integration of heterogeneous modalities to learn complementary shared representations, and (2) **Heterogeneous Modality-Specific Multitask Learning** (`HMML`) – utilizes the shared representations for multiple tasks, such as multi-disease classification and patient survival prediction. In the following, we will delve into the details of these two phases. An overview of our proposed method is shown in Figure 2. Detailed algorithm of `EHPAL-Net` is given in Algorithms 1-2 in C.

### 3.1    MULTIMODAL SHARED INFORMATION LEARNING PHASE (MSIL)

`MSIL` phase utilizes `EHF` layers to learn shared representations over $m$ modalities. At `EHF` layer $i$, the current and previous modality inputs ($x_i$ and $x_{i-1}$) are processed together through three core modules, namely a) **Efficient Multimodal Residual Convolution (`EMRC`)**, b) **Physics-inspired Cross-modal Fusion Attention (`PCMFA`)**, and c) **Shared Information Refinement (`SIR`)** modules. We will delve into the details of these three modules in the following. It is important to note that each `EHF` layer uses two `PCMFA` modules to refine shared representations, then fuses their outputs through a (learned) late fusion (`LF`) strategy, i.e., leveraging trainable weights to capture more expressive multimodal information (Figure 2). This representation, along with the next modality's input, is fed

Figure 3: (A) Illustration of the `PCMFA` module's role in learning complementary shared representations. (B) Overview of the `PCMFA` module, which employs the `HQMGA` approach – comprising the `MHDGA` and `MQIA` mechanisms and a `MAFG` block that fuses the attention maps from `MHDGA` and `MQIA` to learn shared attention maps $A_i$.

to the subsequent `EHF` layer. It can be seen that `MSIL` phase is a single-pass sequential pipeline that processes modalities in order (unlike `HEALNet`'s (Hemker et al., 2024) iterative loop-based strategy). This design facilitates robust shared representation learning by effectively optimizing performance and computational cost.

### 3.1.1 EFFICIENT MULTIMODAL RESIDUAL CONVOLUTION MODULE

Efficient Multimodal Residual Convolution (`EMRC`) module captures multi-scale spatial representation for each modality $x_i'$ while ensuring low computational cost. To achieve this, `EMRC` incorporates Modality-specific Heterogeneous Convolutions Fusion (`MHCF`) blocks, which facilitate progressive refinement of modality-specific inputs. The architectural and implementation details of the `EMRC` module are provided in D.

### 3.1.2 PHYSICS-INSPIRED CROSS-MODAL FUSION ATTENTION MODULE

Recent advances in hyperbolic neural networks (`HNNs`) and quantum neural networks (`QNNs`) have demonstrated their ability to learn richer representations than traditional Euclidean architectures (e.g., `CNNs`, Transformers), especially on complex natural visual tasks (Ganea et al., 2018; Peng et al., 2021; Cong et al., 2018; Shi et al., 2024). `HNNs` leverage non-Euclidean geometries such as the Poincaré ball (Nickel & Kiela, 2017b) and the Lorentz hyperboloid (Chen et al., 2021), to preserve hierarchical structural information efficiently, often in lower-dimensional spaces than their Euclidean counterparts. Concurrently, `QNNs` employ quantum principles like superposition and entanglement to operate in high-dimensional Hilbert spaces (Cong et al., 2018), with recent variants (e.g., `QSAN` (Shi et al., 2024) and `QSANN` (Li et al., 2022)) redefining neural layers (e.g., convolutions, attentions) using parameterized quantum states, improving representational capacity while potentially reducing computational cost.

*Despite their respective complementary strengths – hierarchical structure preservation in HNNs and expressive representational capacity in QNNs – these paradigms have not been unified effectively within a single framework to benefit MFL.* To bridge this gap, we propose a unified attention mechanism – Physics-inspired Cross-modal Fusion Attention (`PCMFA`) module, which is the key component of our proposed `EHPAL-Net` framework. Like `EMRC`, `PCMFA` module (Figure 3) is also designed to learn robust complementary representations ($x_i^s$), however, it does it by jointly optimizing cross-modal interactions in hyperbolic and quantum spaces. This way, it captures rich structural relationships by leveraging both non-Euclidean geometry and quantum states. It processes the refined multimodal inputs $x_{i \in [1:m]}'$ (output of the `EMRC` module) and learns physics-inspired shared attention maps ($A_{i \in [1:m]}$). `PCMFA` module consists of two core blocks: a) **Hyperbolic Quantum Mutual Guidance Attention** (`HQMGA`) block, that captures complex hierarchical structures across modalities by employing mutually guided attention streams in hyperbolic and quantum spaces (Figures 3 (B) and Figure 6), and b) **Multimodal Attention Fusion Gating** (`MAFG`) block that fuses information from hyperbolic-quantum streams across modalities, facilitating the learning of refined complementary shared representations. The working of these two blocks can be written in the following form:

$$A_i = \texttt{PCMFA}(x_i^s, x_{i+1}') = \texttt{MAFG}\left(\texttt{HQMGA}\left(x_i^s, x_{i+1}'\right)\right). \tag{1}$$

where:

$$x_i^s = x_i' \odot A_{i-1} \odot \alpha_i. \tag{2}$$

The above equation captures cross-model interaction where $\alpha_i$ represents modality-specific learnable parameters and $\odot$ denotes the Hadamard product. Let us discuss HQMGA and MAFG blocks in the following:

- HQMGA block is designed by integrating two main attention mechanisms (Figure 3 (B)) that we described in the following:

  1. **Multimodal Hyperbolic Dual-Geometry Attention (MHDGA)** mechanism (Figure 5 (A)) that exploits complementary properties of the Poincaré ball and Lorentz models to learn rich hierarchical representations across modalities. MHDGA mechanism is implemented with two sub-blocks – Poincare Information Learning (PIL($\cdot$)) and Lorentz Information Learning (LIL($\cdot$)). These two sub-blocks respectively compute Poincaré attention weights ($A_i^P$) and Lorentzian attention weights ($A_i^L$) [4]. The resulting geometry-specific weights are fused via learnable parameters $L$ and $P$, followed by a sigmoid activation ($\sigma(\cdot)$) to learn dual-geometry attention maps (for $x_i'$ and $x_i^{s'}$) – in the following we will denote both $x_i'$ and $x_i^{s'}$ as $x_i'$:

$$
\begin{aligned}
A_i^D &= \text{MHDGA}(x_i'), \\
&= \sigma\big(L \times A_i^L + P \times A_i^P\big), \\
&= \sigma\big(L \times \text{LIL}(\psi_i) + P \times \text{PIL}(\psi_i)\big), \text{ where } \psi_i = \text{GAP}(\text{DCT}(x_i')).
\end{aligned}
\tag{3}
$$

     Here $\psi_i = \text{GAP}\big(\text{DCT}(x_i')\big) \in \mathbb{R}^e$ represents the application of Global Average Pooling (GAP) followed by a Discrete Cosine Transform (DCT) to capture frequency components.

     **Poincaré Information Learning** sub-block (Figure 5 (B)) is designed based on the principles of the Poincaré ball model (Nickel & Kiela, 2017b). Given multimodal frequency-domain inputs, i.e., $\psi_i$, we define $\text{MPE}^{\tilde{c}}(\cdot)$ as Multimodal Poincare Exponential map with learnable curvature $\tilde{c}$ and each $\psi_i$ is projected onto the $e$-dimensional hyperbolic manifold with learnable curvature $\tilde{c}$ as follows:

$$
\text{MPE}^{\tilde{c}}(\psi_i) = \tanh\Big(\sqrt{\tilde{c}}\,\|\psi_i\|\Big)\,\frac{\psi_i}{\|\psi_i\| + \varepsilon}.
\tag{4}
$$

     Here $\|\cdot\|$ denotes the Euclidean norm and $\varepsilon = 10^{-6}$ ensures numerical stability. This ensures $\|\text{MPE}^{\tilde{c}}(\psi_i)\| < 1$, exploiting hyperbolic volume growth to compactly encode hierarchical relationships in low-dimensional manifolds. To enable adaptive curvature $\tilde{c}$, we modulate a base curvature $c$ with fractal-scaling weights $\mathbf{f} \in \mathbb{R}^e$: $\tilde{c} = c \times \frac{1}{e}\sum_{j=1}^e \sigma\big(f_j\big)$. The base curvature is parameterized as: $c = \text{clip}\big(e^k, 0.1, 10.0\big)$, with $k \in \mathbb{R}$. This bounded formulation ensures that the geometry remains stable – avoiding degenerate flatness ($c \to 0$) or excessive sharpness ($c \to \infty$) – across both Poincaré and Lorentz models. The Poincaré attention weights are computed via the Hadamard product between the hyperbolic projection and the original frequency components:

$$
A_i^P = \text{PIL}(\psi_i) = \sum_i \big(\text{MPE}^{\tilde{c}}(\psi_i)\big) \odot \psi_i.
\tag{5}
$$

     **Lorentz Information Learning** sub-block (Figure 6 (A)) is designed based on the Lorentzian hyperboloid model (Ganea et al., 2018). Given multimodal frequency-domain inputs, i.e., $\psi_i$, we define Multimodal Lorentz Embedding denoted as $\text{MLE}(\psi_i)$, and split it into a temporal axis ($t_i$) and frequency components ($f_i$) as follows:

$$
t_i, f_i = \underbrace{\text{MLE}(\psi_i)[:1]}_{\text{time axis}}, \underbrace{\text{MLE}(\psi_i)[1:]}_{\text{frequency axes}}; \ \forall i \in [1, m].
\tag{6}
$$

     Here we define $\text{MLE}(\psi_i)$ as:

$$
\text{MLE}(\psi_i) = \theta\left(\left(\frac{\sqrt{1 + \tilde{c}\,\|\psi_i\|_2^2}}{\tilde{c}}\right), \psi_i\right),
\tag{7}
$$

     where $\theta$ represents concatenation. Next, we obtain a channel-wise bias ($\delta_i \in \mathbb{R}^e$) from the Multimodal Quantum-Inspired Attention (MQIA) mechanism (discussed next). This bias is

---

[4]These components aim to preserve complex hierarchical structural cues across modalities in non-Euclidean spaces.

injected into the Lorentz manifold through a Minkowski inner-product modulation (Figure 6 (A-B)), guiding the computation of Lorentzian attention weights:

$$A_i^L = \mathtt{LIL}(t_i, f_i) = \alpha_i\Big(-t_i^2 + \sum_{l=1}^{i} \beta_l \left(f_{i,l} + \delta_{i,l}\right)^2\Big), \tag{8}$$

where $\alpha_i$ and $\beta_l$ are learnable scalars modulating the contributions of temporal and frequency components.

2. **Multimodal Quantum-Inspired Attention (`MQIA`)** mechanism (Figure 6 (B)) embeds modality-specific inputs into a complex Hilbert space via quantum-inspired mappings to capture long-range dependencies. *In parallel with `MHDGA`, the `MQIA` stream is designed to infuse quantum-inspired frequency priors to guide the Lorentz information learning block.* For each modality $i$, we leverage modality-specific frequency components $\psi_i$ to compute complex quantum states: $q_i = \psi_i \cdot \left(\eta_{\text{real}} + j\,\eta_{\text{imag}}\right)$, where $\eta_{\text{real}}, \eta_{\text{imag}}$ are learnable parameters. By applying the Born rule (Hall, 2013; Nielsen & Chuang, 2010), we compute the element-wise amplitudes of the quantum states: $|q_i|^2 = q_i\, q_i^*$, where $q_i^*$ denotes the complex conjugate of $q_i$. We then project the real component of $q_i$ into the Lorentz manifold using Eqs. 6-7, allowing quantum states to be interpreted in hyperbolic spaces via mutual guidance (Figure 6 (A-B)), thus improving the learning of complex representational structural details. Let $\lambda_i = \|\mathtt{MLE}(\psi_i)_{1:}\|$ be the hyperbolic Lorentz-norm of $\mathtt{MLE}(\cdot)$ – drawing on fractal-geometry principles, we define the quantum attention weights, followed by attention maps as:

$$A_i^Q = \mathtt{MQIA}(\psi_i) = \mathtt{Softmax}\big(|q_i|^2 \,\big\| \mathtt{MLE}(\psi_i)\big\|^{h-2}\big). \tag{9}$$

where $h$ is a scalar hyperparameter controlling the strength of the quantum-inspired interaction. The resulting $\mathtt{MQIA}(\cdot)$ yields the bias $\delta_i$ used in `LIL` sub-block. Through mutual guidance between `LIL` and `MQIA`, our `PCMFA` module captures richer, geometry-aware hierarchical structural details across modalities.

- `MAFG` (Figure 3 (B)) block dynamically fuse geometry-aware attention maps from `MHDGA` mechanism with quantum-guided maps from `MQIA` mechanism. For each modality $i$, we use modality-specific learnable weights $c_i$, by adaptively weighting each modality's contribution to learn attention maps $A_i$. The `MFAG` is formally defined as:

$$\begin{aligned}
A_i &= \mathtt{MAFG}(\mathtt{HQMGA}(x_i')), \\
&= \big(c_i \odot \mathtt{MHDGA}\,(x_i')\big) + \big(c_i \odot \mathtt{MQIA}\,(x_i')\big), \\
&= \big(c_i \odot \sigma\big(L \times \mathtt{LIL}(\psi_i) + P \times \mathtt{PIL}(\psi_i)\big)\big) + \big(c_i \odot \mathtt{MQIA}(\psi_i)\big). \tag{10}
\end{aligned}$$

This adaptive fusion ensures robust integration of hyperbolic and quantum structural priors, enhancing cross-modal interactions and enabling the learning of complementary shared representations.

Throughout this work, we use the term *physics-inspired* in a mechanistic sense: `PCMFA` combines (i) a geometric branch implemented as exponential maps on negatively curved manifolds (Poincaré ball and Lorentz hyperboloid) and (ii) a quantum-inspired branch whose scores are squared amplitudes in a complex Hilbert space, fused through a probability-preserving, norm-stable gate; Appendix B formalizes this definition (Definition 1) and proves that `PCMFA` satisfies it (Theorem 1).

### 3.1.3 LEARNABLE LATE FUSION

Given shared representations $\{x_i^S\}$ from the `PCMFA` module, where each $x_i^S \in \mathbb{R}^{H \times W \times C}$, we design learnable Late-Fusion (`LF`) layer, which learns *scalar*, content-aware gates that (i) adapt to the current sample, (ii) reduce exactly to zero whenever a modality is missing, and (iii) incur negligible overhead (one global-pool and a tiny `MLP` per modality), thereby capturing a more expressive multimodal representation ($x_{j\in[1:m]}^{S'}$). See E for more details.

### 3.1.4 SHARED INFORMATION REFINEMENT MODULE

We design the Shared Information Refinement (`SIR`) module to further refine complementary shared representations from `PCMFA`, thereby improving the representational diversity as: $\hat{x}_i^S$. Details of this module are given in F.

## 3.2 Heterogeneous Modality-specific Multitask Learning Phase

The `HMML` phase is designed to leverage the robust complementary shared representations $x^C$ from the `MSIL` phase of `EHPAL-Net` for multiple tasks across $m$ modalities. It maps input $\mathcal{X}$ to output predictions $\mathcal{Y}$ using a loss function $\mathcal{L}_{\text{HMML}}$, where $\lambda_t^m$ controls the weight of each task-modality-specific loss $\mathcal{L}_t^m$. The best model parameters $\beta^*$ are found by minimizing $\mathcal{L}_{\text{HMML}}$:

$$\mathcal{L}_{\text{HMML}} = \sum_{t=1}^{T} \sum_{m=1}^{M} \lambda_t^M \cdot \mathcal{L}_t^m \big( \mathcal{F}(x^C; \beta), \mathcal{Y} \big) \quad \text{and} \quad \beta^* = \arg\min_{\beta} \big( \mathcal{L}_{\text{HMML}} \big) \tag{11}$$

where $\beta$ signifies the `EHPAL-Net` parameters. *For formal definitions and proofs that precisely formalize (i) the concept of `PCMFA` as a physics-inspired attention mechanism, (ii) the dual-geometry interaction implemented by `MHDGA`, and (iii) how this interaction preserves hierarchical structure, please refer to Sections B.1, B.2, and B.3.* A comprehensive analysis of the efficient, robust shared-representation learning enabled by the `EHF` layer is provided in Sections B.4 and B.5.

## 4 Experimental Analysis and Results

**Datasets.** We evaluate `EHPAL-Net` on 15 heterogeneous medical datasets, grouped as follows: (1) **Imaging datasets** (`D1-D8`): HAM10000 (Tschandl et al., 2018), SIPaKMeD (Plissiti et al., 2018), PathMNIST and OrganAMNIST (MedMNIST) (Yang et al., 2023), BraTS-2021 (Baid et al., 2021), SARS-CoV-2 CT-Scan (Angelov & Soares, 2020), CNMC-2019 (Mourya et al., 2019), Chest X-ray Pneumonia (Kermany, 2018a). (2) **Multi-omics datasets** (`D9-D13`): TCGA's BRCA, UCEC, GBMLGG, KIRP, and BLCA. (3) **Clinical EHR datasets** (`D13-D16`): MIMIC-III (Johnson et al., 2016), MHEALTH (Banos et al., 2014), UCI-HAR (Anguita et al., 2013). In all experiments each dataset $D_k$ is treated as one modality stream in the `MSIL` phase (except `BLCA`, which is paired modality[5]). The datasets are unpaired: they originate from different cohorts and institutions, and each defines its own prediction task and label space. Images were resized to $128 \times 128 \times 3$, split 80/10/10 into train/val/test sets, with standard augmentations applied. *All experiments are run with five random seeds, and we report the mean, the standard deviation, or both for each experiment.*

**Baselines.** We compare `EHPAL-Net` against state-of-the-art (`SOTA`) single-modal learning and multimodal fusion learning (`MFL`) methods. For `D1-D15` datasets: (1) *Single-modal learning models* (`M1{M5`): POTTER (Zheng et al., 2023), NAT (Hassani et al., 2023), DDA-Net (Cui et al., 2023b), MFMSA (from MADGNet (Nam et al., 2024)), MSCAM (from EMCAD (Rahman et al., 2024)); (2) *Early fusion `MFL` model* (`M6`): Perceiver; (3) *Late fusion `MFL` models* (`M7{M11`): Gloria, HAMLET, MuMu, MTTU-Net (Cheng et al., 2022), M³Att; (4) *Intermediate fusion MFL models* (`M12{M13`): MOTCAT, DRIFA-Net; (5) *Hybrid early fusion `MFL` model* (`M14`): HEALNet. Our `EHPAL-Net` is instantiated with four backbones—ResNet18, ResNet50 (He et al., 2016), Inception-v3 (Szegedy et al., 2016), ViT-Tiny (Steiner et al., 2021) and ShuffleNet (Zhang et al., 2018) – yielding variants `EHPAL-Net-18`, `EHPAL-Net-50`, `EHPAL-Net-IN`, `EHPAL-Net-V`, and `EHPAL-Net-SN`.

**Evaluation Metrics.** We report the following metrics: accuracy (`ACC`), AUC, concordance index (`C-Index`), number of parameters (`#P`) in millions, and floating-point operations (`#F`) in `GFLOPs`.

**Training Details.** All models are trained for 200 epochs on a single `NVIDIA A100 40GB GPU` running on a Ubuntu machine. (1) Loss: Negative log-likelihood (`NLL`) for survival prediction (multi-omics), and cross-entropy for all classification tasks (imaging, `EHR`). (2) Optimizer: Adam with initial learning rate $1 \times 10^{-3}$. (3) Scheduler: ReduceLROnPlateau down to $1 \times 10^{-6}$. Further details on the datasets and implementations are provided in G, H, and I.

### 4.1 Performance Comparisons

The multi-disease classification, survival, and mortality prediction, and human activity recognition results are summarized in Tables 2 and 5-6 (see J). Our `EHPAL-Net` demonstrates exceptional performance ranging from 68.22% to 100% across fifteen heterogeneous medical datasets (`D1-D15`). `EHPAL-Net` consistently outperforms both single-modal and multimodal fusion baselines by 0.05%–11.44% while reducing model parameters up to $\approx 98.3\%$ and `FLOPs` up to $\approx 97.6\%$. Qualitative results on `D1-D2` datasets (ref. N) illustrate its ability to capture highly discriminative contexts. These results highlight `EHPAL-Net` empowers resource-limited healthcare `AI` with strong

---

[5]Whole-slide histopathology images (WSI) and multi-omics profiles

Table 2: Performance comparison of `EHPAL-Net` variants—`EHPAL-Net-18`, `EHPAL-Net-50`, `EHPAL-Net-IN`, and `EHPAL-Net-SN` – against `SOTA` models on heterogeneous datasets. **Bold** and underlined indicate the best and the second-best results, respectively.

| Models | Datasets → Backbone | HAM10000 ACC | AUC | SIPaKMeD ACC | AUC | BRCA C-Index | MORT ACC | AUC | ICD9 ACC | AUC | PATHMNIST ACC | AUC | OrganAMNIST ACC | AUC | UCEC C-Index | Overall #P↓ | #F↓ |
|---|---|---|---|---|---|---|---|---|---|---|---|---|---|---|---|---|---|
| POTTER | ResNet18 | 91.35 | 91.72 | 92.40 | 92.66 | 60.41 | 85.18 | 86.55 | 67.20 | 90.17 | 91.46 | 99.58 | 96.06 | 99.85 | 66.28 | 12 | 0.95 |
| NAT | Swin-T | 93.11 | 93.25 | 91.87 | 91.62 | 61.34 | 86.25 | 88.29 | 68.40 | 92.18 | 91.76 | 99.87 | 95.72 | 99.60 | 68.37 | 20 | 1.1 |
| DDA-Net | ResNet18 | 92.83 | 92.14 | 92.39 | 92.61 | 63.46 | 87.16 | 88.68 | 66.85 | 91.22 | 92.24 | 99.65 | 95.73 | 99.75 | 68.20 | 12.1 | 1.12 |
| MFMSA | ResNet50 | 97.90 | 97.90 | 94.76 | 95.38 | 66.75 | 89.52 | 92.90 | 69.63 | 94.32 | 92.57 | 99.70 | 96.90 | 99.80 | 70.27 | 26.9 | 1.4 |
| MSCAM | PVT2-B2 | 97.55 | 97.71 | 94.25 | 95.05 | 65.84 | 90.78 | 93.90 | 70.10 | 94.84 | 93.18 | 99.75 | 96.54 | 99.90 | 70.13 | 26.9 | 1.3 |
| Gloria | ResNet50 | 93.75 | 94.58 | 94.32 | 94.50 | 64.63 | 89.15 | 92.28 | 65.0 | 88.60 | 92.41 | 99.58 | 95.85 | 99.75 | 68.31 | 30.8 | 1.54 |
| HAMLET | ResNet50 | 93.25 | 93.20 | 92.84 | 93.32 | 63.22 | 88.35 | 91.40 | 67.12 | 90.58 | 92.20 | 99.80 | 95.64 | 99.72 | 68.70 | 57.3 | 3.52 |
| MTTU-Net | ResNet50 | 97.45 | 97.18 | 91.90 | 92.56 | 62.23 | 89.58 | 92.80 | 68.5 | 92.81 | 92.48 | 99.50 | 95.45 | 99.60 | 69.18 | 38.1 | 6.8 |
| MuMu | ResNet50 | 92.80 | 93.12 | 92.15 | 92.78 | 64.92 | 88.74 | 91.85 | 66.88 | 90.90 | 91.87 | 99.45 | 95.64 | 99.80 | 69.27 | 56.6 | 2.97 |
| $M^3 Att$ | Swin-B | 95.80 | 95.94 | 92.32 | 92.88 | 66.38 | 90.75 | 94.05 | 68.10 | 92.48 | 93.20 | 99.90 | 95.51 | 99.65 | 70.25 | 183 | 12.14 |
| Perceiver | ResNet50 | 92.42 | 92.57 | 91.55 | 91.70 | 64.78 | 87.20 | 88.81 | 71.30 | 96.52 | 93.45 | 99.80 | 96.27 | 99.90 | 70.9 | 31.0 | 11.7 |
| MOTCAT | ResNet50 | 95.35 | 95.80 | 93.56 | 93.94 | 65.35 | 90.27 | 93.78 | 66.22 | 89.56 | 92.80 | 99.50 | 95.10 | 99.58 | 68.80 | 3.9 | 1.48 |
| HEALNet | ResNet50 | 98.24 | 98.17 | 94.75 | 94.80 | 67.30 | 91.24 | 94.30 | 70.66 | 95.72 | 93.58 | 99.83 | 96.12 | 99.90 | 70.25 | 27.2 | 3.84 |
| DRIFA-Net | ResNet18 | 98.33 | 98.51 | 95.58 | 95.75 | 66.47 | 91.32 | 94.10 | 70.28 | 95.10 | 93.45 | 99.75 | 96.45 | 99.90 | 70.48 | 53.8 | 4.83 |
| EHPAL-Net-18 | ResNet18 | 99.75 | 99.99 | 95.97 | 99.67 | 71.27 | 92.34 | 95.55 | 73.64 | 98.24 | 94.25 | 99.95 | 97.78 | **99.98** | 71.74 | 7.71 | 0.59 |
| EHPAL-Net-50 | ResNet50 | **99.81** | 99.99 | 96.32 | 99.90 | 71.59 | 92.52 | 96.10 | 76.53 | 99.30 | 94.63 | 99.98 | 98.12 | **99.98** | 72.50 | 14.4 | 1.83 |
| EHPAL-Net-IN | Inception-v3 | 98.62 | 98.81 | 96.62 | 99.90 | 71.85 | 91.55 | 94.81 | 74.20 | 98.85 | 94.40 | 99.96 | 97.15 | 99.20 | 74.82 | 13.7 | 2.82 |
| EHPAL-Net-V | ViT-Tiny | 99.75 | 99.99 | 97.10 | 99.95 | 72.22 | 93.37 | 97.08 | 76.85 | 99.47 | 95.02 | 99.95 | 98.51 | **99.98** | 73.16 | 10.8 | 1.25 |
| EHPAL-Net-SN | ShuffleNet | 97.84 | 98.64 | 95.25 | 97.73 | 68.22 | 90.37 | 93.64 | 71.41 | 97.10 | 92.87 | 99.55 | 96.85 | 99.0 | 70.1 | 3.1 | 0.29 |

generalization to achieve optimal performance at minimal computational cost – surpassing leading competitors (`DRIFA-Net`, `HEALNet`, `MTTU-Net`) – across diverse medical imaging modalities.

*To tackle challenge 1* (**effective learning of richer complementary shared representations**), we adopt an efficient hybrid fusion strategy. First, `EMRC` captures multi-scale spatial details across modalities. Next, `PCMFA` focuses on learning cross-modal interactions through intermediate fusion, followed by learned late fusion, which integrates complementary cues across each modality's inputs – preserving both shared cross-modal patterns to capture multimodal information. Finally, `SIR` enhances representational diversity by refining these shared representations. This cascaded design outperforms existing `MFL` approaches, which rely solely on early, intermediate, late, or hybrid early fusion schemes, in capturing richer and more effective shared representations. *To address challenge 2* (**efficient and effective design**), `EHPAL-Net` integrates three efficient and effective modules – `EMRC`, `PCMFA`, and `SIR` – that jointly optimize performance-computation trade-offs.

Table 3: Missing-modality evaluation and performance comparison of `EHPAL-Net`—trained with all modalities—against leading uni-modal and multi-modal fusion baselines. At test time each sample supplies only one modality (omics or whole-slide histopathology images (`WSI`)). We report performance in four settings: (i) Omics-only, (ii) `WSI`-only, (iii) mixed (50% of samples randomly drawn from each modality), and (iv) full (100% of samples containing both modalities).

| Model | Omics | WSI | 50% of Both | 100% of Both |
|---|---|---|---|---|
| MFMSA | 61.7 | 52.7 | 57.2 | 57.2 |
| DRIFA-Net | 56.9 | 48.6 | 56.1 | 68.7 |
| CA-MLIF | 54.5 | 47.2 | 51.1 | 70.7 |
| HealNet | 64.7 | 55.4 | 61.4 | 71.4 |
| LegoFuse | 65.2 | 56.5 | 62.9 | 73.4 |
| **EHPAL-Net** | **66.7** | **57.9** | **64.9** | **75.1** |

The synergistic design captures rich spatial-frequency cues to achieve optimal performance with minimal computational overhead, addressing key limitations of medical `AI` in resource-limited environments. *To deal with challenge 3* (**generalization**), `EHPAL-Net` is evaluated on fifteen diverse medical datasets, demonstrating its scalability across heterogeneous data sources and its adaptability to complex multi-task learning.

**Effectiveness on paired and missing–modality settings.** To verify that `EHPAL-Net` also handles classical paired multimodal data, we further evaluate it on `TCGA-BLCA`, which provides per-patient whole-slide histopathology (`WSI`) and multi-omics profiles. Table 3 reports C-index under four test regimes. When only one modality is available at inference time, `EHPAL-Net` achieves 66.7% C-index on omics-only and 57.9% on `WSI`-only, improving over the strongest baseline (`LegoFuse`) by 1.5 and 1.4 points, respectively. In the mixed "50% of both modalities (50% `WSI` and 50% `Omics` are randomly selected" setting, `EHPAL-Net` reaches 64.9% C-index, outperforming `LegoFuse` by 2.0 points and `HEALNet` by 3.5 points. When both modalities are present for all test patients ("100% of both"), `EHPAL-Net` attains 75.1% C-index, a gain of 1.7 points over `LegoFuse` and 3.7 points over `HEALNet`. These results show that the proposed fusion architecture extends naturally to standard paired `WSI` + omics survival prediction, while maintaining strong robustness to missing modalities at test time. To demonstrate *reliability*, we present the comprehensive results in J. Further analyses – including robustness to *missing modality*, input-order permutations, and misaligned or asynchronous modalities – together with comparisons to recent baselines, are provided in L.

Table 4: (**A**) Fusion-scheme ablations: performance of alternative fusion strategies embedded in `EHPAL-Net` across heterogeneous medical datasets. (**B**) Module-wise ablations: incremental impact of each `EHPAL-Net` component on imaging-only and multi-omics benchmarks.

| | (A) Fusion Strategies | | | | | | | (B) Integrated components of EHPAL-Net | | | | | | | | | |
| | HAM10000 | | SIPaKMeD | | BRCA | MORT | | 100% Diverse Imaging Modalities | | | | | 100% Multi-omics | | | | | |
| Model | ACC | AUC | ACC | AUC | C-Index | ACC | AUC | EMRC | PCMFA | SIR | Acc | AUC | EMRC | PCMFA | SIR | C-Index | #Params | #FLOPs |
|---|---|---|---|---|---|---|---|---|---|---|---|---|---|---|---|---|---|---|
| Early | 94.9 | 95.4 | 92.1 | 95.3 | 64.2 | 86.3 | 88.8 | ✓ | × | × | 89.6 | 92.7 | ✓ | × | × | 67.2 | **5.53** | **0.38** |
| Intermediate | 98.4 | 98.3 | 94.9 | 97.9 | 68.8 | 89.7 | 92.3 | ✓ | × | ✓ | 90.9 | 94.4 | ✓ | × | ✓ | 70.5 | 6.82 | 0.47 |
| Late | 97.8 | 98.1 | 94.4 | 97.3 | 67.4 | 88.9 | 93.8 | ✓ | ✓ | × | 95.8 | 99.2 | ✓ | ✓ | × | 77.2 | 7.53 | 0.55 |
| Hybrid Early | 98.6 | 98.7 | 95.6 | 98.3 | 70.5 | 91.1 | 94.3 | ✓ | ✓ | ✓ | **96.9** | **99.9** | ✓ | ✓ | ✓ | **79.3** | 7.71 | 0.59 |
| **Efficient Hybrid Fusion** | **99.8** | **99.9** | **95.97** | **99.7** | **71.3** | **92.3** | **95.7** | × | ✓ | ✓ | 96.3 | 99.4 | × | ✓ | ✓ | 78.1 | 14.7 | 1.45 |

## 4.2 ABLATION STUDY

**Fusion Strategy.** We evaluate five fusion schemes – early, intermediate, late, hybrid early as inspired from `Perceiver`, `DRIFA-Net`, `HAMLET`, and `HEALNet`, respectively, and our efficient hybrid fusion – within `EHPAL-Net` on four heterogeneous medical datasets (HAM10000, SIPaKMeD, TCGA-BRCA, MIMIC-III) (Table 4 (A)). Our efficient hybrid fusion architecture – comprising cascaded EMRC, PCMFA, and SIR modules – achieves performance improvement of 0.8%–7.1% over all alternative fusion strategies. Existing schemes often suffer from limited capacity to learn richer, complementary shared representations across modalities due to their inability to preserve effective modality-specific structural properties. In contrast, our cascaded design preserves these crucial structures and, through an intermediate-late fusion strategy, facilitates the learning of richer and more synergistic representations – thereby effectively *addressing challenge 1* (Hemker et al., 2024). This seamless integration of efficient modules underpins the observed improvements.

**Component Integration.** We next dissect the contributions of `EHPAL-Net`'s modules – EMRC, PCMFA, and SIR – and PCMFA's core components: MHDGA, MQIA, and MAFG, while also comparing cascaded attention (CA) against parallel fusion attention (PFA). Experiments span four modality sets: (1) **Imaging modalities (four datasets):** HAM10000, SIPaKMeD, PathMNIST, and OrganAMNIST; (2) **Multi-omics modalities:** BRCA, UCEC, GBMLGG, and KIRP; (3) **EHR/time-series modalities:** MIMIC-III, MHEALTH, and UCI-HAR; and (4) **Mixed settings:** HAM10000, SIPaKMeD, UCI-HAR, and BRCA. In all cases, each dataset is treated as a separate modality stream. As shown in Tables 4 (B), 8-9 (ref. K), omitting any component incurs a 0.18%–10.1% performance drop. This highlights the effectiveness of our fully integrated design for capturing complementary shared representations and achieving optimal performance with minimal computational overhead – thereby *tackling challenges 1–2*.

In the cascaded- vs. parallel-fusion attention comparison (Table 9), our parallel fusion attention strategy outperforms the cascaded-attention baselines by 0.19%–2.26%, demonstrating its superior ability to capture complementary shared representations with minimal computational overhead. To isolate the benefit of parallel fusion, we compare the configuration of MHDGA and MQIA streams in parallel versus a cascaded setup (Table 9). The parallel fusion attention mechanism not only enhances the learning of complementary shared representations but also mitigates the progressive information loss observed in cascaded designs (Lv et al., 2024; Shen et al., 2021; Fu et al., 2019). We attribute these improvements to the direct inter-module interactions facilitated by the parallel architecture, which preserve crucial cross-modal cues that may otherwise degrade when attention blocks are applied sequentially. Comprehensive ablation studies – spanning *diverse attentions*, *integrated components*, *per-modality* (unimodal) performance, *modality imbalance* in multimodal fusion, and strategies for mitigating *modality dominance* – are summarized in K.

**Discussion.** Our method learns richer shared representations across heterogeneous modalities by preserving their structural properties and highlighting the benefit of effective and efficient cross-modal learning, and analyzing the associated computational complexity. Details appear in M.

## 5 CONCLUSION

We present `EHPAL-Net`, a novel, efficient, and effective multimodal fusion framework designed for analyzing heterogeneous medical data, making it ideal for resource-constrained, AI-driven healthcare settings. Evaluated across fifteen diverse datasets, `EHPAL-Net` achieves strong cross-modal generalization, outperforming leading state-of-the-art methods by up to 3.97% in performance, with 85.7% fewer parameters and 87.8% lower `FLOPs`. Future work will focus on strengthening adversarial defenses and evaluating the framework on non-medical vision benchmarks to demonstrate adaptability across both medical and natural-vision domains.

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

## A   APPENDIX – ADDITIONAL CONTEXT AROUND OUR PROPOSED METHOD

Quantum probability represents a physical state as a unit vector in a complex Hilbert space and converts measurement amplitudes into probabilities through the Born rule (Nielsen & Chuang, 2010; Hall, 2013). *Quantum-inspired attention* (QIA) transports this formalism to deep learning by encoding features as complex amplitudes and deriving *normalized, amplitude-based* similarities. Recent "quantum self-attention" architectures demonstrate that such amplitude-driven scoring remains fully classical yet competitive with real-valued baselines (Shi et al., 2024; Li et al., 2022). Concurrently, *hyperbolic learning* exploits the constant negative curvature of the Poincaré ball and Lorentz (hyperboloid) models to embed a hierarchical structure with low distortion, implementing gyrovector/Möbius operations in deep networks (Nickel & Kiela, 2017b; Ganea et al., 2018; Chen et al., 2021). Finally, complex-valued neural networks naturally handle magnitude and phase information – an advantage for frequency-domain inputs such as MRI data (Hirose, 2006).

**Hyperbolic and quantum-inspired modeling for medical multimodal learning:**   Hyperbolic methods have progressed from Poincaré embeddings (Nickel & Kiela, 2017b) to full neural architectures for natural-vision tasks (Ganea et al., 2018), yet their uptake in medical AI remains limited. None of the existing approaches provides a unified attention mechanism that learns robust *shared* representations for fusing heterogeneous medical modalities.

In parallel, QIA – originally developed for natural-image understanding – formulates attention in a Hilbert-space probabilistic framework, deriving weights from state amplitudes or density matrices (Shi et al., 2024; Li et al., 2022; Chen et al., 2024). Complex-valued networks have already proven effective for phase-sensitive tasks such as MRI reconstruction, demonstrating that amplitude-based modeling need not rely on quantum hardware (Cole et al., 2021; 2020). **Despite these advances, both hyperbolic and QIA research have yet to yield a computationally efficient multimodal fusion framework capable of handling diverse medical data at scale. To our knowledge, no prior work combines QIA with *dual* hyperbolic geometry to create a single, cost-effective representation space that improves multimodal fusion performance in real-world medical AI applications.**

**Our key contribution – the PCMFA module:**   The PCMFA module fills this gap. A *dual-geometry* hyperbolic stream (Poincaré **+** Lorentz) runs alongside a *quantum-inspired* amplitude stream; the two exchange guidance information before fusion via the MAFG. Instantiated twice per EHF layer, this design learns complementary shared representations at low cost. Because the streams operate in *parallel* – not cascaded –PCMFA remains both geometry-aware and computationally efficient, a novel combination in medical multimodal fusion that directly tackles diverse scales and structures.

## B   APPENDIX – THEORETICAL FOUNDATIONS OF PHYSICS-INSPIRED ATTENTION

In this section we justify the term physics-inspired in a rigorous sense. We introduce a formal definition of a physics-inspired attention mechanism that requires (i) a geometric branch realized as a Riemannian exponential-map embedding on a constant-negative-curvature manifold, (ii) a quantum-inspired branch defined on a complex Hilbert space with Born-rule–based probability scores, and (iii) a probability-preserving, non-expansive fusion of these scores. We then prove that the proposed PCMFA module satisfies this definition (Theorem 1), analyze the dual-geometry interaction between Poincaré and Lorentz attentions (Theorem 2), show that the resulting gate preserves hyperbolic hierarchical structure (Corollary 1), and establish that it acts as a contraction-like operator on feature norms (Theorem 3). Together, these results demonstrate that our "physics-inspired" design is not merely metaphorical but enforces concrete geometric and probabilistic constraints inherited from physical modeling.

In this section we make precise the notions of *physics-inspired attention*, *dual-geometry interaction*, and *hierarchical structure preservation* in the context of the EHF pipeline. Specifically, we focus on the PCMFA module and its MHDGA/MQIA components (Sec. 3.1.2), and show how they yield stable and efficient shared representations within EHF.

## B.1 PHYSICS-INSPIRED NATURE OF PCMFA

To formalize the intuition behind `PCMFA`, we frame the module as the fusion of two norm-stable branches—a geometric path operating in hyperbolic space and a quantum-inspired path operating in a complex Hilbert space. We first state a precise definition of a physics-inspired attention mechanism and then prove that `PCMFA` satisfies this definition.

**Definition 1** (Physics-inspired attention mechanism). *Let $\mathcal{X}$ be an input feature space, $\mathcal{M}$ a Riemannian manifold of constant negative curvature (e.g., a Poincaré ball or Lorentz hyperboloid), and $\mathcal{H}$ a finite-dimensional complex Hilbert space with inner product $\langle \cdot, \cdot \rangle$. An attention mechanism $T : \mathcal{X} \to [0,1]^e$ is called* physics-inspired *if there exist maps $E^{\mathrm{geo}} : \mathcal{X} \to \mathcal{M}$ and $\Psi : \mathcal{X} \to \mathcal{H}$ such that:*

> *(G1)* ***Geometric branch.*** *$E^{\mathrm{geo}}$ is realized via a Riemannian exponential-type map on $\mathcal{M}$ (e.g., Poincaré ball or Lorentz hyperboloid), and the corresponding attention scores depend only on geodesic distances or norms in $\mathcal{M}$.*

> *(Q1)* ***Quantum-inspired branch.*** *$\Psi(x)$ is a complex state whose channel-wise scores are proportional to Born-rule amplitudes $|\Psi(x)_k|^2$, i.e., squared magnitudes in $\mathcal{H}$, normalized to a probability vector, following the standard probabilistic interpretation of quantum states.*

> *(N1)* ***Physical normalization and stability.*** *The final attention weights $T(x)$ are obtained from these physically derived scores by a probability-preserving normalization (e.g., Softmax or sigmoid) and induce a non-expansive multiplicative update on features in every $\ell_p$ norm.*

**Theorem 1** (`PCMFA` is a physics-inspired attention mechanism). *For each `EHF` layer $i$, the `PCMFA` map*

$$A_i = \mathtt{PCMFA}(x_i^S, x_{i+1}')$$

*defined in Eq. (1) is a physics-inspired attention mechanism in the sense of Definition 1.*

*Proof.* We verify (G1)–(Q1)–(N1) for the `PCMFA` construction in Sec. 3.1.2.

*(G1) Geometric branch.* Within `PCMFA`, the `MHDGA` block takes $\psi_i = \mathtt{GAP}(\mathtt{DCT}(x_i'))$ (Eq. (3)) and maps it to hyperbolic embeddings in two constant-negative-curvature models: the Poincaré ball and the Lorentz hyperboloid (Eqs. (4)–(8)). Both mappings are exponential-type projections with learnable but bounded curvature (see the curvature parameterization and clipping in Sec. 3.1.2 and Lemma 2 below), so $E^{\mathrm{geo}}(\psi_i)$ is a well-defined element of $\mathcal{M}^P \times \mathcal{M}^L$. The corresponding attentions $A_i^P, A_i^L$ depend only on hyperbolic radii and Lorentzian norms, and are fused into $A_i^D$ by an affine map plus sigmoid (Eq. (3)), satisfying (G1).

*(Q1) Quantum-inspired branch.* `MQIA` maps the same $\psi_i$ into a complex vector $q_i \in \mathbb{C}^e$ via learned real and imaginary coefficients (Eq. (9)), so $q_i$ lies in a Hilbert space $\mathcal{H} \cong \mathbb{C}^e$. The channel-wise quantities $|q_{i,k}|^2$ are Born-rule amplitudes, and Softmax over a scaled version of $|q_i|^2$ together with a Lorentz-norm term, yields a probability vector $A_i^Q$ (Eq. (9)), satisfying (Q1).

*(N1) Normalization and stability.* `MAFG` fuses $A_i^D$ and $A_i^Q$ channel-wise using bounded coefficients $c_i$ (Eq. (10)). Lemma 2 below implies that all components of $A_i^D$ lie in $(0, 1)$ and $A_i^Q$ is a probability vector; under the bounded gate assumption, the fused gate $A_i$ satisfies $\|A_i\|_\infty \leq 1$. Theorem 3 then shows that the multiplicative update $z \mapsto z \odot A_i$ is non-expansive in every $\ell_p$ norm.

Together, these properties establish that `PCMFA` has (i) a geometric branch on hyperbolic manifolds, (ii) a quantum-inspired branch in a complex Hilbert space, and (iii) a normalized, non-expansive fusion of the two, so it is physics-inspired in the sense of Definition 1. $\square$

## B.2    DUAL-GEOMETRY INTERACTION IN MHDGA

We next formalize the *dual-geometry interaction* implemented by MHDGA and show that it acts as a monotone consensus operator over Poincaré and Lorentz attentions.

**Definition 2** (Dual-geometry interaction). *For modality $i$, let $\psi_i = \texttt{GAP}(\texttt{DCT}(x_i'))$ be the frequency summary (Eq. (3)). Denote by*

$$\phi_i^P(\psi_i) \in \mathcal{M}^P, \qquad \phi_i^L(\psi_i) \in \mathcal{M}^L$$

*the Poincaré and Lorentz embeddings produced by the MHDGA sub-branches (Eqs. (4)– (8)). Let $A_i^P, A_i^L \in \mathbb{R}^e$ be the corresponding geometry-specific attention vectors computed from curvature-aware norms and Minkowski inner products, and define the fused dual-geometry attention (Eq. (3)) as*

$$A_i^D = \sigma(LA_i^L + PA_i^P),$$

*where $L$ and $P$ are learnable linear maps and $\sigma$ is the element-wise sigmoid. The map $\texttt{MHDGA}_i :$ $\psi_i \mapsto A_i^D$ is called the* dual-geometry interaction operator *for modality $i$.*

A useful property of the hyperbolic projections used in MHDGA is that they are radially monotone.

**Lemma 1** (Radial monotonicity of hyperbolic projections). *Let $\psi, \tilde{\psi} \in \mathbb{R}^e$ with $\|\psi\|_2 < \|\tilde{\psi}\|_2$. Assume that the Poincaré and Lorentz embeddings in MHDGA are implemented via exponential maps at the origin as in Eqs. (4) and (7). Then:*

*1. the hyperbolic radius of $\phi^P(\psi)$ in the Poincaré model is strictly smaller than that of $\phi^P(\tilde{\psi})$;*

*2. the Lorentzian rapidity of $\phi^L(\psi)$ is strictly smaller than that of $\phi^L(\tilde{\psi})$.*

*Proof.* In both models, the exponential map at the origin has the form $\psi \mapsto f(\|\psi\|_2)\,\psi/\|\psi\|_2$, where $f$ is a smooth, strictly increasing function of the Euclidean radius $\|\psi\|_2$ (cf. Eqs. (4) and (7)). Thus a larger Euclidean norm yields a larger hyperbolic radius or rapidity, giving the claimed ordering. $\square$

**Proposition 1** (Monotone consensus dual-geometry attention). *Fix modality $i$ and channel $k$. Let*

$$A_{i,k}^D = \sigma\big(\ell_k(A_{i,k}^L, A_{i,k}^P)\big)$$

*denote the $k$-th component of $A_i^D$, where $\ell_k$ is the $k$-th component of the affine map $LA_i^L + PA_i^P$. Suppose the coefficients of $\ell_k$ with respect to $A_{i,k}^L$ and $A_{i,k}^P$ are non-negative.* **Then:**

*1. $A_{i,k}^D$ is weakly increasing in each of its arguments $A_{i,k}^L$ and $A_{i,k}^P$;*

*2. if both $A_{i,k}^L$ and $A_{i,k}^P$ increase, then $A_{i,k}^D$ increases strictly;*

*3. when one of $A_{i,k}^L, A_{i,k}^P$ increases and the other decreases, $A_{i,k}^D$ remains between the scalar responses obtained by relying on either geometry alone, i.e., it behaves as a consensus between the two branches.*

*Proof.* The derivative of $A_{i,k}^D$ with respect to $A_{i,k}^L$ is

$$\frac{\partial A_{i,k}^D}{\partial A_{i,k}^L} = \sigma'(\ell_k)\,\frac{\partial \ell_k}{\partial A_{i,k}^L}.$$

The sigmoid derivative $\sigma'$ is strictly positive, and by assumption the partial derivative of $\ell_k$ with respect to $A_{i,k}^L$ is non-negative, so $\partial A_{i,k}^D/\partial A_{i,k}^L \geq 0$. The same argument holds for $A_{i,k}^P$, proving item 1. If both $A_{i,k}^L$ and $A_{i,k}^P$ increase and at least one coefficient is strictly positive, then $\ell_k$ increases and strict monotonicity of $\sigma$ implies item 2. For item 3, note that $\ell_k$ is affine with non-negative coefficients, so for fixed values of one argument it varies monotonically with the other. Applying a sigmoid, which maps $\mathbb{R} \to (0, 1)$ monotonically, ensures that $A_{i,k}^D$ lies between the values induced by each branch, yielding the consensus behavior. $\square$

Lemma 1 shows that both geometries encode the same radial ordering of channels, while Proposition 1 shows that their fusion in $A_i^D$ respects and stabilizes this ordering. This is the sense in which MHDGA realizes a *dual-geometry interaction*: two hyperbolic representations cooperate through a monotone consensus gate to produce robust, geometry-aware attention.

## B.3 HIERARCHICAL STRUCTURE PRESERVATION

We now connect dual-geometry interaction to the notion of *hierarchical structure preservation*. Intuitively, channels (or semantic units) that are deeper in an underlying hierarchy should receive systematically stronger or weaker attention than their ancestors; hyperbolic geometry is well known to represent such tree-like structures efficiently.

**Definition 3** (Hierarchical structure and its preservation). *Let $(\mathcal{C}, \preceq)$ be a finite partially ordered set of semantic units (e.g., classes or channels) with a distinguished root $r \in \mathcal{C}$. A map $f : \mathcal{C} \to \mathcal{Z}$ into a metric space $(\mathcal{Z}, d_{\mathcal{Z}})$ is called* radially hierarchical *if there exists a scalar function $\rho : \mathcal{Z} \to \mathbb{R}_{\geq 0}$ such that:*

1. *$\rho(z)$ depends only on $d_{\mathcal{Z}}(z, z_0)$ for some fixed root point $z_0 \in \mathcal{Z}$, and*

2. *whenever $u \preceq v$ in $\mathcal{C}$, one has $\rho(f(u)) < \rho(f(v))$.*

*An attention mechanism that assigns scores $\alpha_c$ to units $c \in \mathcal{C}$ is said to be* hierarchical-structure preserving *if each $\alpha_c$ is a monotone function of a radially hierarchical representation of $c$.*

**Assumption 1** (Hyperbolic hierarchical encoding). *For each modality $i$, there exists a latent hierarchy $(\mathcal{C}, \preceq)$ over channels and a radially hierarchical embedding $f_i^\star : \mathcal{C} \to \mathcal{M}^P \times \mathcal{M}^L$ such that the learned hyperbolic mappings $\phi_i^P, \phi_i^L$ used in `MHDGA` approximate $f_i^\star$ in the sense that deeper nodes in the hierarchy are mapped to larger hyperbolic radii (up to small perturbations).*

This assumption is standard in hyperbolic representation learning and is supported by empirical evidence that Poincaré and related embeddings recover hierarchical structures in low-dimensional hyperbolic manifolds.

**Theorem 2** (Hierarchical structure preservation of dual-geometry attention). *Under Assumption 1, suppose that the geometry-specific attentions $A_i^P$ and $A_i^L$ are computed as channel-wise monotone functions of the corresponding hyperbolic radii or norms (as in the `PIL`/`LIL` definitions). Then, for each modality $i$, the dual-geometry attention $A_i^D$ produced by `MHDGA` is hierarchical-structure preserving in the sense of Definition 3: if $u \preceq v$ in the latent hierarchy, then $A_{i,u}^D \leq A_{i,v}^D$ (or the reverse inequality, depending on the chosen monotone direction).*

*Proof.* By Assumption 1 and Lemma 1, deeper nodes in the hierarchy have strictly larger hyperbolic radii in both the Poincaré and Lorentz embeddings. By construction, $A_i^P$ and $A_i^L$ are obtained by applying monotone functions (linear maps and element-wise nonlinearities) to these radii or norms, so they are monotone in depth: $u \preceq v \Rightarrow A_{i,u}^P \leq A_{i,v}^P$ and likewise for $A_i^L$. Proposition 1 shows that $A_i^D$ is monotone in each of $A_i^P$ and $A_i^L$, hence $A_i^D$ inherits the same ordering and is hierarchical-structure preserving. $\square$

Finally, we propagate this property through the full `PCMFA` gate.

**Corollary 1** (Hierarchical physics-inspired attention in PCMFA). *Under Assumptions 2 and 1, the full `PCMFA` gate $A_i$ obtained by fusing $A_i^D$ and $A_i^Q$ via `MAFG` (Eq. (10)):*

1. *preserves the hierarchical order of channels induced by the hyperbolic embeddings, and*

2. *is non-expansive as a multiplicative operator on feature tensors, as stated in Theorem 3.*

*Proof.* By Theorem 2, $A_i^D$ is hierarchical-structure preserving. The quantum-inspired attention $A_i^Q$ is a probability vector derived from the same summary $\psi_i$ and acts as an energy-based reweighting of channels (Eq. (9)). `MAFG` fuses $A_i^D$ and $A_i^Q$ with bounded coefficients $c_i$ in a channel-wise monotone manner. Thus the hierarchical ordering from $A_i^D$ is preserved in $A_i$. Boundedness of $A_i$ and non-expansiveness of the resulting multiplicative update follow from Lemma 2, Assumption 2, and Theorem 3. $\square$

Corollary 1 formalizes that `PCMFA` is not only physics-inspired and norm-stable, but also respects the hierarchical structure encoded by the hyperbolic embeddings in `MHDGA`.

### B.4 EFFICIENT ROBUST SHARED-REPRESENTATION LEARNING

We now recall the basic guarantees that connect the design of PCMFA (Eqs. (3)– (10)) and the EHF pipeline (Algorithms 1–2) to our stated objective: learning robust complementary *shared representations* at *low cost*.

**Lemma 2** (Manifold boundedness and normalized attention). *Let $\psi_i = GAP(DCT(x'_i)) \in \mathbb{R}^e$ be the frequency summary used in PCMFA. Then:*

    *(i) the Poincaré projection $MPE_{\tilde{c}}(\psi_i)$ has Euclidean norm $< 1$;*

    *(ii) the quantum-inspired attention $A_i^Q$ is simplex-normalized (nonnegative, sums to 1) because it is computed via Softmax on modulus-squared amplitudes (Born rule);*

    *(iii) the dual-geometry attention $A_i^D$ produced by MHDGA is componentwise in $(0, 1)$ because it passes through a sigmoid.*

*Consequently, all channel-wise weights that feed MAFG are bounded and well-posed.*

**Assumption 2** (Bounded effective gate). *Let the MAFG output for modality $i$ be $A_i = c_i \odot A_i^D + c_i \odot A_i^Q$ (Eq. (10)), and define the effective gate $a_i := A_i$. We assume $\|a_i\|_\infty \leq 1$. This is satisfied, for example, if $\|c_i\|_\infty \leq \frac{1}{2}$ (since $A_i^D, A_i^Q \in [0, 1]^e$ by Lemma 2); alternatively one may implement a convex gate $A_i = c_i \odot A_i^D + (1 - c_i) \odot A_i^Q$, which also guarantees $\|a_i\|_\infty \leq 1$.*

**Theorem 3** (Non-expansive PCMFA gating $\Rightarrow$ stable features). *Under Assumption 2, for any feature tensor $z \in \mathbb{R}^{H \times W \times e}$ and any $p \in [1, \infty]$,*

$$\| z \odot a_i \|_p \leq \|z\|_p.$$

*Hence, the PCMFA + MAFG update $x_i^S = x'_i \odot A_i$ is a non-expansive mapping in $\ell_p$, acting as a contraction-like regularizer that prevents channel blow-up and encourages stable, complementary shared representations across modalities.*

**Proposition 2** (Per-layer linear cost and overall $O(m)$ fusion). *Let $m$ be the number of modalities processed by the MSIL phase and let $n$ denote the per-sample spatial size ($n = H \times W \times e$). One EHF layer applies EMRC $\rightarrow$ PCMFA (MHDGA $\|$ MQIA, no token–token all-pairs) $\rightarrow$ SIR to two streams once (and PCMFA a second time as in Algorithm 2), so the work per layer is $O(n)$ up to constants fixed by the backbone. Because MSIL composes $m - 1$ such layers sequentially (Algorithms 1–2), the total fusion cost is $O(mn)$, in contrast to pairwise/co-attention designs that realize $O(m^2n)$ interactions.*

Lemma 2 formalizes that the hyperbolic and quantum-inspired branches of PCMFA produce *bounded, normalized* attention signals, which MAFG then fuses. Theorem 3 shows that the resulting gate cannot increase feature norms, stabilizing optimization and mitigating over-amplification of any single modality while promoting robust shared signals. Proposition 2 explains the empirical efficiency of our sequential EHF pipeline relative to all-pairs fusion.

### B.5 PROOFS

*Proof of Lemma 2.* From Eq. (4), the Poincaré projection $MPE_{\tilde{c}}(\psi_i)$ scales $\psi_i$ by a product of two scalar factors, each strictly less than one in magnitude, so $\|MPE_{\tilde{c}}(\psi_i)\|_2 < 1$. For $A_i^Q$, Eq. (9) computes elementwise $|q_i|^2$ from complex amplitudes and then applies SOFTMAX, which by definition yields a nonnegative vector summing to 1. For $A_i^D$, Eq. (3) applies a sigmoid to an affine combination of LIL($\psi_i$) and PIL($\psi_i$), so $A_i^D \in (0, 1)^e$. $\qquad\square$

*Proof of Theorem 3.* Let $a_i$ satisfy $\|a_i\|_\infty \leq 1$ (Assumption 2). For any index $(u, v, w)$, $|(z \odot a_i)_{u,v,w}| = |z_{u,v,w}| |a_{i,w}| \leq |z_{u,v,w}|$, so $\|z \odot a_i\|_\infty \leq \|z\|_\infty$. For $p \in [1, \infty)$,

$$\|z \odot a_i\|_p^p = \sum_{u,v,w} |z_{u,v,w}|^p |a_{i,w}|^p \leq \sum_{u,v,w} |z_{u,v,w}|^p = \|z\|_p^p,$$

hence $\|z \odot a_i\|_p \leq \|z\|_p$. The sum of two non-expansive maps remains non-expansive when the weights lie in $[0, 1]$, as in $c_i \odot A_i^D$ and $c_i \odot A_i^Q$. $\qquad\square$

*Proof of Proposition 2.* From Algorithm 2, each `EHF` layer processes two streams once (lines 5–9), applying `EMRC` and `SIR` each in $O(n)$ (pointwise/group/dilated depthwise operations) and `PCMFA` twice. `PCMFA`'s `MHDGA` computes per-channel weights from $\psi_i = \text{GAP}(\text{DCT}(\cdot))$ and closed-form projections (Eqs. (3) – (8)); `MQIA` forms $q_i$ and a single SOFTMAX over $e$ channels (Eq. (9)). All steps are linear in $n = H \times W \times e$ with no token–token quadratic interactions. `MSIL` composes $m - 1$ layers sequentially (Algorithm 1), hence $O(mn)$ total cost, whereas pairwise attention/no-attention multimodal fusion pipelines across all modality pairs typically require $O(m^2)$ interactions, each $O(n)$, i.e., $O(m^2 n)$. $\square$

## B.6 ADDITIONAL GUARANTEES FOR HYBRID FUSION

In this section, we provide a theoretical grounding for our three main challenges: (1) learning rich, structure-preserving shared representations; (2) achieving an optimal trade-off between performance and computational cost; and (3) ensuring adaptability and generalizability across heterogeneous medical data modalities.

**Challenge 1 — Richer Shared Representation Learning.**

**Definition 4** (Complementary Shared Representation). *The shared representation learned by the `EHF` pipeline (Fig. 2 C ) is complementary shared when it (i) preserves per-modality hierarchical cues via the dual-geometry branch inside `PCMFA`, (ii) captures cross-modal dependencies unavailable to any single modality through the parallel hyperbolic–quantum streams, and (iii) depends only on present modalities through the mask-aware `LF` (Appendix E, Steps 2–3; Eqs. (14) – (15). Items (i)–(ii) follow from Appendix B.1– B.3; item (iii) follows from the `LF` construction.*

**Lemma 3** (Mask-aware `LF` ⇒ missing-modality consistency). *For any `EHF` layer, the `LF` in Appendix E assigns zero weight to any missing modality and renormalizes over the present subset; the fused state equals the fusion of the available inputs only (Appendix E, Eqs. (14)–(15).*

*Proof.* Appendix E sets the logits of missing streams to a large negative constant before the softmax (Step 2), so their weights become exactly zero; the gated concatenation (Step 3) then uses only present streams. $\square$

**Proposition 3** (Dominance mitigation inside `EHF`). *No single present modality can swamp the fusion within an `EHF` layer.*

*Proof.* By Lemma 2 (Appendix B.4) the geometry and quantum branches yield bounded scores; by Assumption 2 and Theorem 3 (Appendix B.4) the multiplicative `PCMFA` update is non-expansive channel-wise; and by Lemma 3 `LF` redistributes probability mass only across present streams via Softmax (Eq. 14). Together, bounded gates + non-expansive updates + normalized `LF` weights prevent any one stream from dominating the fused state. Table 15 empirically corroborates this mechanism. $\square$

**Corollary 2** (Two-pass `PCMFA` yields deep-chain stability). *Each `PCMFA` update is non-expansive (Appendix B.4, Thm. 3). Algorithm 2 applies `PCMFA` twice per `EHF` layer; composing non-expansive maps yields a stable update, and stacking `EHF` layers in `MSIL` preserves this bound (Fig. 2 C; Algorithms 1– 2).*

*Proof.* Theorem 3 (Appendix B.4) shows each `PCMFA` map is non-expansive. The composition of non-expansive operators is non-expansive; Algorithm 2 applies `PCMFA` twice within a single `EHF` layer, hence the claim. $\square$

**Corollary 3** (Hierarchical, physics-inspired attention persists through fusion). *The hierarchy-preservation property established for `PCMFA` (Appendix B.2– B.3: Theorem 2, Corollary 1) is preserved by the mask-consistent `LF` and `SIR` refinement in `EHF`.*

*Proof.* Corollary 1 states that `PCMFA`'s gate is hierarchical and non-expansive; `LF` (Eqs. 13–15) reweights channels with probability-normalized gates without changing their monotone order, and

`SIR` is a residual refinement (Eq. 16). Thus the ordering induced by the dual-geometry attention persists. □

**Challenge 2 — Performance–Cost Trade-off.**

**Proposition 4** (Per-layer linear cost; overall $O(mn)$ sequential fusion). *Let $n$ be per-sample spatial size and $m$ the number of modalities processed in `MSIL`. One `EHF` layer has $O(n)$ work; stacking $m-1$ such layers yields $O(mn)$ fusion cost.*

*Proof.* Proposition 2 (Appendix B.4) and Table 21 show each of `EMRC`, `PIL`, `LIL`, `MQIA`, `LF`, and `SIR` is $O(n)$ per layer; Algorithm 2 executes them once per `EHF` (with two `PCMFA` calls but still linear without token–token all-pairs), hence $O(n)$ per layer; Algorithm 1 composes $m-1$ layers sequentially, hence $O(mn)$. □

**Theorem 4** ($\varepsilon$–do-no-harm under residual + gated interaction). *Consider expanding from $m$ to $m+1$ modalities. For any $\varepsilon > 0$, there exists a setting of the new stream's parameters (`MAFG` gate coefficients and `LF MLP`) that makes the predictions of the $(m+1)$-modality model differ from the $m$-modality model by at most $\varepsilon$, while the per-sample fusion cost increases by an additive $O(n)$ (Proposition 4).*

*Proof.* Keep all pre-existing parameters. In the new `EHF`: (i) set the per-stream `MAFG` coefficients for the new modality arbitrarily close to zero so its `PCMFA` gate contributes an arbitrarily small factor (cf. Eq. 10); (ii) set the `LF MLP` (Eq. 13) to produce a large negative pre-Softmax score for the new modality so its $\alpha$-weight is arbitrarily small among present modalities (Eq. 14); (iii) because `PCMFA` is non-expansive (Theorem 3), multiplying by gates with sup-norm $< \delta$ perturbs the shared state by at most $O(\delta)$; concatenation in `LF` (Eq. 15) preserves this bound. Choosing $\delta$ small enough ensures the output shift is $\leq \varepsilon$. Cost increases by one additional `EHF`, which is $O(n)$ by Proposition 4. □

**Corollary 4** (Constructive performance–cost path). *Varying the new stream's gates from the neutral setting in Theorem 4 to the learned setting traces a continuous path in prediction space while cost increases by a fixed $O(n)$. Empirically, this path improves the frontier relative to baselines at comparable `FLOPs`/parameters (Tables 4, 10, 20; Fig. 8).*

*Proof.* Gate vectors are continuous in their parameters (Eqs. 9 – 10, 13 – 15); by Theorem 3 the mapping from gates to outputs is non-expansive, so the output varies continuously with gates. Cost is fixed by the presence of the extra `EHF` (Proposition 4). Tables 4, 10, 20 and Fig. 8 report the empirical frontier at these cost points. □

**Challenge 3 — Generalization & Adaptability.**

**Lemma 4** (Missing-modality robustness (restated)). *Under the `LF` design in Appendix E, removing any subset of modalities at inference yields a well-defined fused state equal to the fusion over the remaining subset.*

*Proof.* Identical to Lemma 3; see Eqs.( 13– 15). □

**Proposition 5** (Monotone expressivity w.r.t. added modalities). *Let $\mathcal{F}_m$ be the functions realized with $m$ modalities. Then $\mathcal{F}_m$ is contained in the closure of $\mathcal{F}_{m+1}$: adding a stream cannot force worse risk because the neutral setting in Theorem 4 arbitrarily approximates the $m$-modality map, while non-neutral gates can realize strictly different mappings (Tables 2, 14).*

*Proof.* Theorem 4 gives an $\varepsilon$-approximation to the $m$-modality map inside the $(m+1)$-modality model; hence $\mathcal{F}_m \subseteq \overline{\mathcal{F}_{m+1}}$. Tables 2 and 14 show cases where using the extra stream strictly improves performance. □

**Proposition 6** (Permutation stability — sufficient conditions). *If `EHF` parameters are shared across modality positions and `PCMFA`/ `LF` depend only on content and use symmetric fusion (Eqs. 1– 2, 13– 15), then `MSIL` is permutation-equivariant in the stream order and the final predictions are invariant. Empirically, Table 19 shows near-invariance even without strict sharing.*

*Proof.* Under parameter sharing and symmetric operators, swapping the order of input streams commutes with each `EHF` update (Fig. 2 C; Algorithm 2). By induction over the $m-1$ `EHF` layers, the fused state and predictions coincide for any permutation. Table 19 reports the observed invariance. $\square$

## C  APPENDIX – EHPAL-NET ALGORITHM

---

**Algorithm 1** `EHPAL-Net` $(x_i)$

---

1: **Input:** Modalities, $x_i = [x_1, x_2, \ldots, x_m]$, where $x_i \in \mathbb{R}^{H_i \times W_i \times C_i} \cup \mathbb{R}^{D_i}$
2: **Output:** Shared representation $x^C$ and downstream multitask outputs
3: **Procedure:**
4: **if** phase $==$ `MSIL` **then**
5:    /* *MSIL phase for learning robust representations* */
6:    **for** each $x_i$, where $i = 1$ to $m$ **do**
7:      **if** $i == 2$ **then**
8:        $(\hat{x}_i^S, x_i^{S'}) \leftarrow$ `EHF`$(x_{i-1}, x_i)$
9:      **else**
10:        $(\hat{x}_i^S, x_i^{S'}) \leftarrow$ `EHF`$(x_i^{S'}, x_{i+1})$
11:      **end if**
12:      **if** $i == d \in m$ **then**
13:        $x^C \leftarrow \theta(x_i^{S'}, \theta(\hat{x}_i^S))$      // Robust Representations Eq. 16
14:      **end if**
15:    **end for**
16: **else**
17:    /* *HMML phase for multitask learning* */
18:    **for** each task $t$ **do**
19:      $\mathcal{L}_{\text{HMML}} \leftarrow \sum_{t=1}^{T} \sum_{m=1}^{M} \lambda_t^M \cdot \mathcal{L}_t^m(\mathcal{F}(x^C; \beta), \mathcal{Y})$
20:      $\beta^* \leftarrow \arg\min_\beta (\mathcal{L}_{\text{HMML}})$      // as per Eq. 11
21:    **end for**
22: **end if**

---

**Algorithm 2** `EHF`$(x_{i-1}, x_i)$

---

1: **Input:** Heterogeneous modality inputs $x_{i-1}$, $x_i$
2: **Output:** Shared representations $\hat{x}_i^S$, $x_i^{S'}$
3: **Procedure:**
4: /* *Efficient Hybrid Fusion (`EHF`) layer for multimodal input integration pipeline* */
5: $x_{i-1}', x_i' \leftarrow$ `EMRC`$(x_{i-1}, x_i)$    // Multi-scale spatial information learning Eq. 12.
6: $x_{i-1}^S, x_i^S \leftarrow$ `PCMFA`$(x_{i-1}', x_i')$    // Complementary shared representation learning Eqs. 1–10.
7: $x_{i-1}^S, x_i^S \leftarrow$ `PCMFA`$(x_{i-1}^S, x_i^S)$    // Further enriches information.
8: $x_i^{S'} \leftarrow$ `LF`$(x_{i-1}^S, x_i^S)$    // Handles missing modalities Eqs. 13–15.
9: $\hat{x}_i^S \leftarrow$ Call `SIR`$(x_i^S)$    // Enhances representational diversity Eq. 16.
10: **Return:** $\hat{x}_i^S$, $x_i^{S'}$

---

## D  APPENDIX – EFFICIENT MULTIMODAL RESIDUAL CONVOLUTION MODULE

We design the **E**fficient **M**ultimodal **R**esidual **C**onvolution (`EMRC`) module (ref. Fig. 4 (A)) to capture multi-scale spatial representations while ensuring low computational cost. The `EMRC` incorporates **M**odality-specific **H**eterogeneous **C**onvolutions **F**usion (`MHCF`) blocks (Fig. 4 (B)), along with Batch-Norm ($\text{B}(\cdot)$) ReLU ($\mathcal{R}(\cdot)$), and pointwise convolution ($\text{PC}(\cdot)$) to facilitate progressive refinement. Unlike the uniform depth-wise convolutions employed in `EMCAD`'s `MSDC` (Rahman et al., 2024), `MHCF` uses diverse branch-wise convolutions – Group-Pointwise (`GPC`), Dilated Depth-wise (`DDC`), and Depth-wise (`DWC`) – at varying scales ($1\times1$, $3\times3$, $5\times5$), capturing heterogeneous spatial contexts and promoting branch-wise heterogeneity. The resulting contexts are fused and refined using a

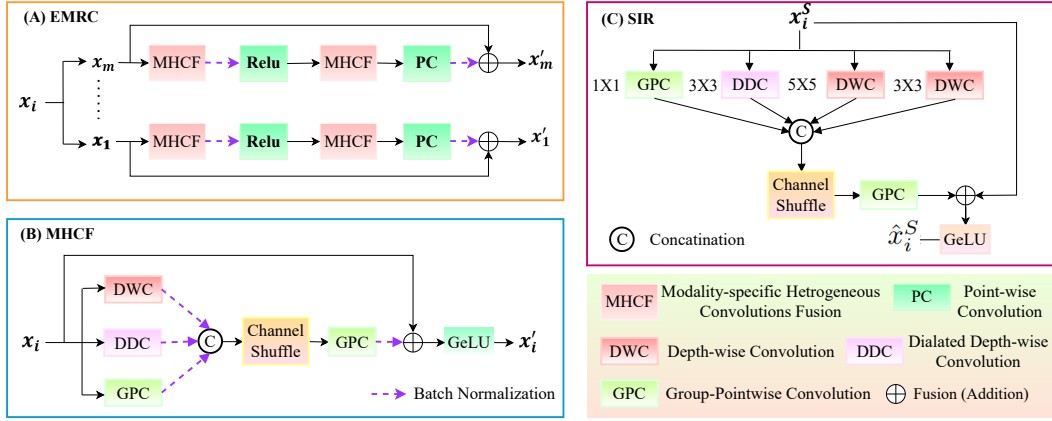

Figure 4: Architecture of the Efficient Multimodal Residual Convolution (EMRC) and Shared Information Refinement (SIR) modules. (A) The EMRC module integrates the Modality-specific Heterogeneous Convolutions Fusion (MHCF) block (shown in B) to progressively refine multimodal representations $x_i'$, where $i \in [1:m]$. (C) The SIR module takes the shared representations $x_i^S$ produced by the PCMFA module and refines them into $\hat{x}_i^S$ within each EHF layer, further enhancing representational diversity.

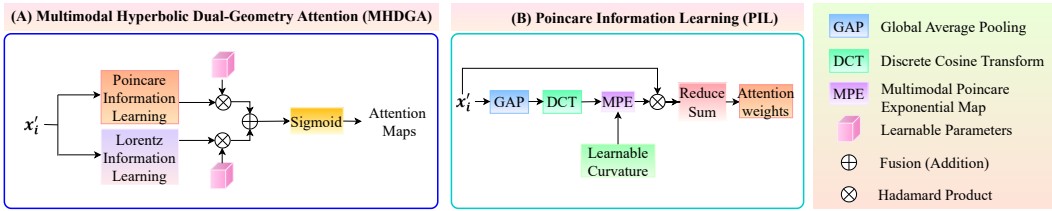

Figure 5: (A) Overview of MHDGA module. (B) Overview of the Poincaré Information Learning (PIL) sub-block.

channel shuffle (CS($\cdot$)) to facilitate inter-channel communication, thereby enhancing representational diversity, denoted as $x_i' = $ EMRC($\cdot$):

$$\text{EMRC}(x_i) = x_i + \underbrace{\text{B}\Big(\text{PC}\big(\text{MHFC}(\mathcal{R}(\text{B}(\text{MHFC}(x_i))))\big)\Big)}_{\text{Progressive Refinement}}, \quad \text{MHFC}(x_i) = \text{CS}\Big(\theta\Big(\underbrace{\forall_{sc \in \{1,3,5\}} \text{HC}_{sc}(x_i)}_{\text{Branch-wise Heterogeneity}}\Big)\Big)$$
(12)

where the skip connection ensures stable gradient flow and regularization, $\text{HC}_{sc} \in \{\text{GPC}, \text{DDC}, \text{DWC}\}$, and $\theta$ denotes concatenation (fusion).

## E   APPENDIX – LEARNABLE LATE FUSION

Given the shared representations $\{x_i^S\}$ from the PCMFA module, the overall LF workflow is presented in the following steps:

**Step 1 – Channel context learning.** For each modality $i$, we capture channel information, $z_i$, by applying global average pooling (GAP) and feeding the resulting vector, $p_i$, into a two-layer perceptron $h_i$ with a ReLU activation:

$$p_i = \text{GAP}(x_i^S) \ \in \ \mathbb{R}^C, \quad z_i = h_i(p_i) \ \in \ \mathbb{R}.$$
(13)

**Step 2 – Mask-aware Gating.** Let $mask \in \{0,1\}^m$ be a binary mask indicating which modalities are present in the current sample. We suppress missing modalities by multiplying them by a constant

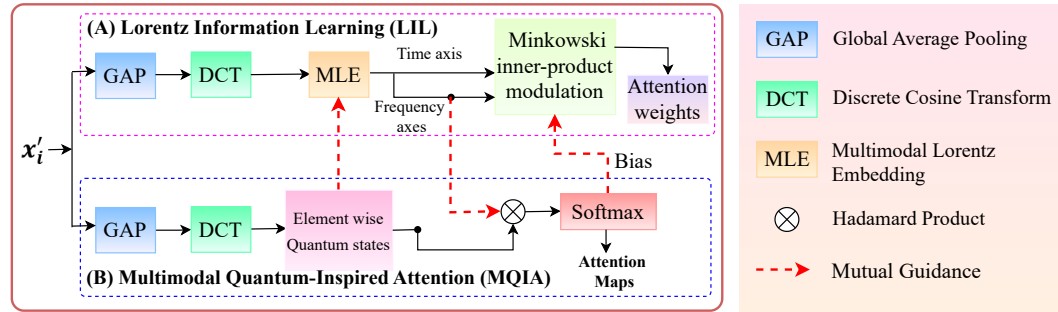

Figure 6: Schematics of the two key blocks: (A) Lorentz Information Learning (`LIL`) sub-block and (B) `MQIA` module, which interact through bidirectional mutual guidance.

$o$ and then apply softmax to compute the attention weights, $\alpha_i$:

$$\tilde{z}_i = z_i + (1 - mask_i)\, o, \quad \alpha_i = \frac{\exp(\tilde{z}_i)}{\sum_{i=1}^{m} \exp(\tilde{z}_i)}. \tag{14}$$

By construction, $\alpha_i = 0$ if $mask_i = 0$ and $\sum_{i:mask_i=1} \alpha_i = 1$.

**Step 3 – Gated concatenation.** The resulting weights are used to fuse with the shared features $\{x_i^S\}_{i=1}^m$ corresponding to each modality $i$ and to capture richer, more expressive shared representations $x_j^{S'}$ ($j \in [1:m]$):

$$x_j^{S'} = \text{LF}\big(\{x_i^S\}, mask_i\big) = \theta\big(\alpha_1\, x_1^S, \ldots, \alpha_m\, x_m^S\big) \in \mathbb{R}^{H \times W \times (mC)}, \tag{15}$$

where $\theta(\cdot)$ denotes concatenation-based fusion.

Relative to static element-wise modulation, the our `LF` layer is:

- *Content-aware*: gates depend on each modality's own features.
- *Missing-modality robust*: the masked softmax sets the contribution of any absent modality exactly to zero.
- *Efficient*: one `GAP` and a lightweight `MLP` per modality add negligible computational cost.

At inference, at least one modality must be present.

# F   APPENDIX – SIR MODULE

`SIR` (Figure 4 (C)) extends our `MHCF` block by incorporating an additional `DWC` with $3 \times 3$ scale, resulting in heterogeneous convolutions at scales—$1 \times 1$, $3 \times 3$, $5 \times 5$, and $3 \times 3$–where the `DDC` uses a dilation rate ($r = 2$) to capture broader contexts. The outputs are fused to maximize representational diversity while maintaining computational efficiency. Subsequently, a channel shuffle (`CS`($\cdot$)) and `GPC` are employed to facilitate cross-scale interactions across $m$ modalities while a skip connection fuses the refined output with $x_i^S$, followed by `GELU` activation (`G`($\cdot$)) to preserve salient information. Finally, `SIR`'s cascaded structure increases the channel dimensions to align with each `PCMFA` output, thereby learning enriched representations ($\hat{x}_i^S = \text{SIR}(x_i^S)$). The resulting representations across $i$th `EHF` layer are then fused (ref. Fig. 2 (A)) to boost representational diversity, thereby learning robust complementary shared representations ($x^C$), as mentioned in Eq. 16, thereby improving `EHPAL-Net`'s effectiveness.

$$\text{SIR}(x_i^S) = \text{G}\bigg(\forall_i\big(x_i^S + \text{GPC}\big(\text{CS}\big(\theta\big(\forall_{sc,r}\, \text{DC}_{sc,r}(x_i^S)\big)\big)\big)\big)\bigg) \text{ and } x^C = \theta\bigg(x_m^{S'}, \big(\forall_i\theta\big(\text{SIR}(x_i^S)\big)\big)\bigg) \tag{16}$$

where $\text{DC}_{sc,r} \in \{\text{GPC}_{1\times1}, \text{DWC}_{3\times3}, \text{DWC}_{5\times5}, \text{DDC}_{3\times3,2}\}$.

# G   APPENDIX – DATASET DESCRIPTIONS AND PREPROCESSING

In this work, we integrate fifteen distinct biomedical datasets spanning dermoscopy, cytology, histopathology, radiology, and multi-omics. All imaging data were uniformly resized to $128 \times 128 \times 3$ pixels, intensity-normalized to $[0, 1]$, and split patient- or sample-wise into 80% train, 10% validation, and 10% test sets. Multi-omics features were harmonized across patients, $\log_2$-transformed or $\beta$-normalized as appropriate, z-score standardized, and similarly partitioned. Below, we provide detailed descriptions of each dataset.

**HAM10000 Dataset.**   The HAM10000 dataset [6] ("Human Against Machine with $10, 000$ training images") (`D1`) (Tschandl et al., 2018) repository aggregates $10, 015$ dermoscopic images collected from multiple clinical centers worldwide to represent seven common pigmented skin lesion categories:

- **Melanocytic nevi (NV)** – $6, 705$ images
- **Melanoma (MEL)** – $1, 113$ images
- **Benign keratosis-like lesions (BKL)** – $1,099$ images (solar lentigines, seborrheic keratoses, lichen-planus–like keratoses)
- **Basal cell carcinoma (BCC)** – $514$ images
- **Actinic keratoses/intraepithelial carcinoma (AKIEC)** – $327$ images
- **Vascular lesions (VASC)** – $142$ images (angiomas, angiokeratomas, pyogenic granulomas, hemorrhages)
- **Dermatofibroma (DF)** – $115$ images

Original images vary in resolution and source; we resized each to $128 \times 128 \times 3$.

**SIPaKMeD Dataset.**   For single-cell cytology classification, we used the publicly available SIPaKMeD dataset[7] (`D2`) (Plissiti et al., 2018), which comprises 4,049 Pap-smear images cropped from 966 whole-slide scans. These images are distributed across five cytological categories:

- **Superficial–Intermediate cells (SIC)** – $831$ images (normal)
- **Parabasal cells (PC)** – $787$ images (normal)
- **Koilocytotic cells (KC)** – $825$ images (abnormal)
- **Dyskeratotic cells (DC)** – $793$ images (abnormal)
- **Metaplastic cells (MC)** – $813$ images (benign)

Each cell was segmented and center-cropped to 450×450 pixels, then resized to $128 \times 128 \times 3$ using standard resizing. All pixel values were normalized to $[0, 1]$. Finally, the dataset was partitioned into 80% training, 10% validation, and 10% test splits, preserving class proportions in each subset.

**PathMNIST Dataset.**   Derived from the `NCT-CRC-HE-100K` (Kather et al., 2019; 2018) (100,000 H&E-stained patches) and `CRC-VAL-HE-7K` (Kather et al., 2019; 2018) (7,180 patches) datasets, `PathMNIST`[8] (`D3`) (Yang et al., 2023) contains non-overlapping histology patches labeled into nine tissue types critical for colorectal cancer diagnosis: adipose (`ADI`), background (`BACK`), debris (`DEB`), lymphocytes (`LYM`), mucus (`MUC`), smooth muscle (`MUS`), normal mucosa (`NORM`), cancer-associated stroma (`STR`), and tumor epithelium (`TUM`). We resized each patch to $128 \times 128 \times 3$ and normalized all pixel values to the range $[0, 1]$.

**OrganAMNIST Dataset**   OrganAMNIST dataset[9] (`D4`) (Yang et al., 2023) comprises 58,830 axial `CT` slices drawn from the `LiTS` liver segmentation benchmark (Bilic et al., 2023), covering 11 organs: bladder; left/right femur; heart; left/right kidney; liver; left/right lung; pancreas; spleen. We converted each original $128 \times 128$ grayscale slice into a $128 \times 128 \times 3$ tensor using a trainable 1×1 convolution to produce three feature maps, then normalized all pixel values to $[0,1]$.

---

[6] https://www.kaggle.com/datasets/kmader/skin-cancer-mnist-ham10000

[7] https://www.cs.uoi.gr/ marina/sipakmed.html

[8] https://medmnist.com/

[9] https://medmnist.com/

**BraTS 2021 Dataset.**    We tackled glioma grading using the 1,251-case `BraTS 2021` training set[10] (`D5`) ([Bakas et al., 2017](#)). Each patient contributes four co-registered, skull-stripped `MRI` volumes—T1, T1ce, T2, FLAIR—resampled to 1 mm³ isotropic and pre-cropped to 240×240×155. Our pipeline:

1. Identify the axial slice with maximal tumor extent via FLAIR segmentation labels.
2. Resize each modality's 2D slice to $128 \times 128$ (bilinear).
3. Intensity-normalize by dividing by per-volume maximum.
4. Fuse the four modalities into a single three-channel image using a learned 1×1 convolutional fusion layer, yielding $128 \times 128 \times 3$.

After modality fusion, we balanced classes by undersampling `HGG` to match `LGG` counts, then split into 80% train, 10% validation, and 10% test sets.

**SARS-CoV-2 CT-Scan Dataset.**    The SARS-CoV-2 CT-Scan dataset[11] (`D6`) ([Soares et al., 2020](#)) contains 2,482 lung `CT` slices (1,252 COVID-19 positive; 1,230 negative) acquired under varied scanning protocols. Each axial slice was resized to $128 \times 128 \times 3$, normalized globally across the dataset, and organized so that no patient's scans spanned multiple splits, resulting in an 80/10/10 train/val/test split.

**C-NMC 2019 Dataset.**    For lymphocyte classification, we used the C-NMC 2019 challenge dataset[12] (`D7`) ([Mourya et al., 2019](#)), which provides 15,114 RGB cell images (450×450) from 118 subjects, with expert labels for malignant (7,272) and healthy (3,389) cells. After Jenner–Giemsa stain normalization and segmentation from $2,560 \times 1,920$ smear scans, each cell was centered in a $450 \times 450$ frame and resized to $128 \times 128 \times 3$. All pixel values were normalized to $[0, 1]$. We randomly partitioned malignant and healthy subsets independently into 80% train, 10% validation, and 10% test.

**Chest X-ray Pneumonia Dataset.**    Chest X-ray Pneumonia dataset[13] (`D8`) ([Kermany, 2018b](#)) was Collected at Guangzhou Women's and Children's Medical Center, this pediatric dataset comprises 5,863 anterior–posterior chest radiographs labeled "Pneumonia" or "Normal." Images passed dual-physician quality control and diagnosis adjudication. Each was cropped to remove borders, resized to $128 \times 128$, and intensity-normalized before an 80/10/10 split.

**TCGA Multi-Omics Datasets**    We assembled four TCGA datasets: `BRCA` (`D9`) ([Lingle et al., 2016](#)), `UCEC` (`D10`) ([Erickson et al., 2016](#)), `BRCA` (`D11`) ([Bakas et al., 2017](#)), and `KIRP` (`D12`) ([Linehan et al., 2016](#)) by harmonizing patient barcodes and merging clinical, genomic, and epigenomic layers:

- **BRCA**[14] **(N=1222):** Clinical outcomes, 249 somatic-mutation features (MAF), 860 CNV segments, 604 RNA-Seq counts ($\log_2(x + 1)$), and 223 RPPA protein levels.
- **UCEC**[15] **(N=559):** Survival outcomes, methylation $\beta$-values, and transcript counts.
- **GBMLGG**[16] **(N=1,153):** Combined glioblastoma and lower-grade glioma cases with clinical labels, methylation, and expression data.
- **KIRP**[17] **(N=291):** Survival times, genome-wide CNV (3,812 segments), 450 K methylation $\beta$-values, and $\log_2$-normalized transcript counts.

All cohorts were first randomly split into 80% training, 10% validation, and 10% test sets. To harmonize sample sizes across other modality datasets, we applied random upsampling on each training set to match the largest dataset's training size. Continuous features in all splits were standardized to zero mean and unit variance using statistics computed on the training data. Each

---

[10] https://www.cancerimagingarchive.net/analysis-result/rsna-asnr-miccai-brats-2021/
[11] https://www.kaggle.com/datasets/plameneduardo/sarscov2-ctscan-dataset
[12] https://www.cancerimagingarchive.net/collection/c-nmc-2019/
[13] https://www.kaggle.com/datasets/paultimothymooney/chest-xray-pneumonia
[14] https://www.cancerimagingarchive.net/collection/tcga-brca/
[15] https://www.cancerimagingarchive.net/collection/tcga-ucec/
[16] https://www.cancerimagingarchive.net/collection/tcga-gbm/
[17] https://www.cancerimagingarchive.net/collection/tcga-kirp/

feature vector was reshaped into a single-channel image of size $128 \times 128$ via bicubic interpolation and converted to three-channel RGB ($128 \times 128 \times 3$). Survival-prediction performance was evaluated on the test sets using the concordance index.

**MIMIC-III Dataset (Johnson et al., 2016).** The MIMIC-III v1.4 database[18] comprises 46 520 unique patients and 58 976 ICU admissions recorded between 2001 and 2012 at Beth Israel Deaconess Medical Center. Data include anonymized demographics, vital signs, laboratory results and ICD-9 diagnosis codes. We restrict to adult ICU stays (age $\geq$ 16 years), yielding 38 597 unique adult stays. We formulate two prediction tasks: in-hospital mortality ("MORT," binary classification) and primary ICD-9 code ("ICD-9," multi-class classification). The data were first randomly split into 80% training, 10% validation, and 10% test sets. To harmonize sample sizes across all modalities, we then applied random upsampling on each training set. Continuous features in each split were standardized to a zero mean and unit variance using statistics computed on the training data. Each feature vector was reshaped into a $128 \times 128$ single-channel image via bicubic interpolation and converted to three-channel ($128 \times 128 \times 3$).

**MHEALTH Dataset (Banos et al., 2014).** The MHealth dataset[19] comprises inertial and physio-logical signals from ten volunteers performing 12 activities: standing, sitting, lying, walking, stair climbing, waist bending, arm elevation, knee bending, cycling, jogging, running, and jumping. Data were collected at 50 Hz using three IMU sensors on the left ankle, right wrist, and chest, capturing acceleration, orientation and angular velocity. We applied a sliding window of length 50 samples (1 s) with no overlap, generating 6 863 examples across all activities. The examples were first split 80%/10%/10% into training, validation and test sets. We then applied random upsampling on each training set to match the largest modality's sample size. Continuous features were standardized to zero mean and unit variance (training-set statistics), reshaped into ($128 \times 128$) single-channel images via bicubic interpolation, and converted to ($128 \times 128 \times 3$).

**UCI-HAR Dataset (Anguita et al., 2013).** The UCI-HAR dataset[20] contains 10 299 recordings of six activities (walking, upstairs, downstairs, sitting, standing, laying) from 30 volunteers wearing a waist-mounted Samsung Galaxy S II. Accelerometer and gyroscope sensors recorded triaxial acceleration and angular velocity at 50 Hz. Raw signals were noise-filtered and segmented using a fixed 128-sample sliding window. We randomly split the 10 299 samples into 80% training, 10% validation and 10% test sets, then applied random upsampling on the training split to harmonize with other modalities. Continuous features were standardized (zero mean, unit variance) using training statistics, reshaped via bicubic interpolation into ($128 \times 128$) grayscale images, and converted to three-channel ($128 \times 128 \times 3$).

## H  APPENDIX – DATA AUGMENTATION

To improve effectiveness, we apply three image-level augmentation techniques to each training sample, generating three additional variants per image. After augmentation, the expanded set of original and transformed images is shuffled to ensure a diverse training distribution.

- **Rotation:** We rotate each image by a fixed angle of 20° using affine transform. *Purpose:* Encourages the model to learn orientation-invariant features, reducing sensitivity to slight rotations in medical images.
- **Translation:** We shift each image horizontally and vertically by 5 pixels via an affine translation. *Purpose:* Promotes spatial invariance, so the network can recognize structures regardless of their exact location within the frame.
- **Gaussian Blur:** We apply a $3 \times 3$ Gaussian filter to slightly smooth each image. *Purpose:* Simulates imaging noise and low-pass filtering, helping the model focus on larger anatomical patterns rather than pixel-level artifacts.

**Implementation details:**

---

[18]https://physionet.org/content/mimiciii/1.4/

[19]https://archive.ics.uci.edu/dataset/319/mhealth+dataset

[20]https://archive.ics.uci.edu/dataset/240/human+activity+recognition+using+smartphones/

1. For each original image $x$ with label $y$, compute rotated, translated, and smooth images and assign each the same label $y$.
2. Concatenate the augmented images with the original training set, resulting in a 4× increase in sample count.
3. Apply a random permutation to the combined dataset to mix original and augmented samples before training.

These augmentations collectively improve model performance by enhancing invariance to rotation, translation, and noise—key factors in real-world medical imaging variability.

# I  APPENDIX – IMPLEMENTATION DETAILS

We ensure a fair comparison by implementing `EHPAL-Net` on four backbone architectures—`ResNet18`, `ResNet50`, `Inception-v3`, and `ShuffleNet`—and by evaluating each baseline with its original backbone: `ResNet18` and `ResNet50` for CNN-based methods, and `PVTv2-B2`, `Swin-Tiny`, and `Swin-B` for transformer-based methods. Specifically, we benchmark five single-modal models—`POTTER` (Zheng et al., 2023), `NAT` (Hassani et al., 2023), `DDA-Net` (Cui et al., 2023b), `MFMSA` (Nam et al., 2024), and `MSCAM` (Rahman et al., 2024)—and nine multimodal fusion methods—`Gloria` (Huang et al., 2021), `MTTU-Net` (Cheng et al., 2022), `HAMLET` (Islam & Iqbal, 2020), `M³Att` (Liu et al., 2023), `MuMu` (Islam & Iqbal, 2022), `Perceiver` (Jaegle et al., 2021), `MOTCAT` (Xu & Chen, 2023), `HEALNet` (Hemker et al., 2024), and `DRIFA-Net` (Dhar et al., 2025). Among these, two baselines employ transformer backbones, while the remaining methods are CNN-based.

In our experiments, every model—whether originally designed for segmentation or for a different vision task—is adapted to the same three objectives: multi-disease classification, patient survival estimation, and mortality prediction. To do this, we strip away segmentation decoders and task-specific heads from all encoder–decoder architectures, retaining only the encoder stacks. Those models that provide standalone attention modules (such as `MFMSA` and `MSCAM`) have had their attention blocks grafted into a standard backbone (`ResNet-50` for `MFMSA`, `PVTv2-B2` for `MSCAM`), again discarding any original decoder or segmentation loss. Each resulting encoder produces a feature vector that is fed into shared, task-appropriate heads: global average pooling followed by fully connected layers for classification and a linear output for survival regression. We evaluate every trained network on accuracy and AUC for the classification tasks, and Harrell's C-index for survival prediction, ensuring a fair apples-to-apples comparison across all baselines and our own method.

**EHPAL-Net (Our Method).**  `EHPAL-Net` fuses all modalities in one unified pipeline while keeping minimal computation cost. We introduce exactly one Efficient Hybrid Fusion (EHF) layer per modality input as follows:

1. The first `EHF` layer jointly processes Modality 1 (e.g., MRI) and Modality 2 (e.g., CT-scan).
2. Each subsequent `EHF` layer takes the new modality's inputs (e.g., genomics or clinical) along with the shared representations from the previous stage—so there is never more than one active `EHF` layer per modality.
3. This cascade of four `EHF` layers (one per modality) captures every cross-modal interaction without the naive 16-pair fusion.
4. Because our imaging modality is used to initialize the parallel and then revisited in its own fusion stage, we apply the `PCMFA` block twice to further enrich the shared features.
5. Dropout ($p = 0.25$) follows each fusion for regularization.

It is important to note that in this study, we leverage four diverse modality basis groups for training. For instance:

- **Group 1** includes:
  - Imaging modalities: `HAM10000` (modality 1) and `SIPAKMed` (modality 2)
  - Multi-omics: `TCGA-BRCA` (modality 3)
  - Clinical (EHR): `MIMIC-III` for *mortality prediction* (modality 4)
- **Group 2** includes:
  - Imaging modalities: `PATHMNIST` (modality 1) and `OrganAMNIST` (modality 2)

- Multi-omics: `TCGA-UCEC` (modality 3)
- Clinical (EHR): `MIMIC-III` for *multi-disease (ICD) prediction* (modality 4)

This strategy enables effective training using one group at a time. Nevertheless, our architecture is fully capable of processing and integrating all modalities simultaneously, as outlined in the proposed design.

This four-stage design captures every cross-modal relationship with only four `EHF` layers—far fewer than the multiple blocks of simple pairwise fusion or the sixteen of a full "all-pairs" scheme—dramatically cutting FLOPs and memory without sacrificing performance.

**Single-modal Baselines.** We treat each modality independently as follows:

**POTTER.** Pooling Attention Transformer replaces standard self-attention with a lightweight Pooling Attention Transformer (`PAT`) block for human mesh recovery. We feed each modality independently to `POTTER` to benchmark its classification and survival performance. The raw source code is available at `https://github.com/zczcwh/POTTER`.

**NAT.** Neighborhood Attention Transformer localizes self-attention to each pixel's nearest neighbors, reducing complexity from quadratic to linear. We apply `NAT` per modality as a strong attention-only baseline. The raw code is available at `https://github.com/SHI-Labs/Neighborhood-Attention-Transformer`.

**DDA-Net.** Dual-Domain Attention Network cascades spatial- and frequency-domain attention for robust image deblurring. We repurpose its attention module on each modality to compare against our fusion approach. The raw code is available at `https://github.com/c-yn/DDANet`.

**MFMSA.** From `MADGNet's` Multi-Frequency Multi-Scale Attention block, we integrate `MFMSA` into ResNet-50—dropping the segmentation decoder—to assess its efficacy on classification and survival tasks. The raw code is available at `https://github.com/Inha-CVAI/MADGNet`.

**MSCAM.** We extract the Multi-Scale Convolutional Attention Module from an efficient segmentation decoder and graft it onto `PVTv2-b2's` encoder for single-modal comparison. The raw code is available at `https://github.com/SLDGroup/EMCAD`.

**Multimodal-Fusion Baselines.** All baselines are re-implemented as dual- or tri-stream networks—one stream per modality—and their feature streams are merged according to each method's originally published fusion strategy, as detailed below:

Table 5: Performance comparison of `EHPAL-Net` integrated with the ResNet18 base network (referred to as `EHPAL-Net-18`) against state-of-the-art (`SOTA`) methods on heterogeneous medical datasets: `BraTS-2021`, `SARS-CoV-2 CT-Scan`, `TCGA-GBMLGG`, and `MHEALTH`, across multiple tasks. **Bold** indicates the best results.

| Datasets → | BraTS-2021 | | SARS-CoV-2 CT-Scan | | TCGA-GBMLGG | MHEALTH | |
|---|---|---|---|---|---|---|---|
| Models ↓ | ACC | AUC | ACC | AUC | C-Index | ACC | AUC |
| MFMSA | $99.45_{\pm0.21}$ | $99.30_{\pm0.56}$ | $98.82_{\pm0.81}$ | $99.0_{\pm0.73}$ | $85.70_{\pm2.69}$ | $99.24_{\pm0.31}$ | $99.10_{\pm0.45}$ |
| MSCAM | $98.95_{\pm0.71}$ | $98.90_{\pm0.64}$ | $98.50_{\pm0.36}$ | $98.50_{\pm0.95}$ | $81.69_{\pm6.27}$ | $98.73_{\pm0.39}$ | $98.85_{\pm0.52}$ |
| Perceiver | $98.60_{\pm1.01}$ | $98.80_{\pm0.89}$ | $98.36_{\pm1.24}$ | $98.40_{\pm1.15}$ | $82.85_{\pm6.27}$ | $97.90_{\pm1.39}$ | $98.25_{\pm1.23}$ |
| MuMu | $99.0_{\pm0.74}$ | $99.0_{\pm0.86}$ | $98.72_{\pm0.90}$ | $98.54_{\pm1.02}$ | $82.63_{\pm5.87}$ | $98.56_{\pm1.14}$ | $98.5_{\pm1.25}$ |
| $M^3Att$ | $99.33_{\pm0.24}$ | $99.30_{\pm0.37}$ | $99.15_{\pm0.54}$ | $99.0_{\pm0.37}$ | $84.24_{\pm4.08}$ | $98.10_{\pm1.13}$ | $98.25_{\pm1.38}$ |
| HEALNet | $99.54_{\pm0.08}$ | $99.60_{\pm0.1}$ | $98.90_{\pm0.93}$ | $99.0_{\pm0.42}$ | $86.75_{\pm2.27}$ | $98.63_{\pm1.07}$ | $98.80_{\pm0.90}$ |
| DRIFA-Net | $99.80_{\pm0.15}$ | $99.80_{\pm0.1}$ | $99.23_{\pm0.42}$ | $99.10_{\pm0.30}$ | $85.52_{\pm2.38}$ | $99.33_{\pm0.48}$ | $99.0_{\pm0.48}$ |
| EHPAL-Net-18 | $\mathbf{100}_{\pm0.0}$ | $\mathbf{100}_{\pm0.0}$ | $\mathbf{99.70}_{\pm0.15}$ | $\mathbf{99.85}_{\pm0.10}$ | $\mathbf{89.34}_{\pm0.53}$ | $\mathbf{99.60}_{\pm0.37}$ | $\mathbf{99.60}_{\pm0.25}$ |

**Gloria.** `Gloria` (Global-Local Representations for Images using Attention) contrasts image sub-regions with words in paired reports to learn context-aware local and global medical image representations. Rather than relying on pretrained object detectors, it learns attention weights that highlight relevant image regions for each word, enabling joint multimodal embedding through

Table 6: Performance comparison of `EHPAL-Net` integrated with the ResNet18 base network (referred to as `EHPAL-Net-18`) against `SOTA` methods on heterogeneous medical datasets: `CNMC-2019`, `Pneumonia CXR`, `UCI-HAR`, and `TCGA-KIRP`, across multiple tasks. **Bold** indicates the best results.

| Datasets → | CNMC-2019 | | Pneumonia | | UCI-HAR | | KIRP |
|---|---|---|---|---|---|---|---|
| Models ↓ | ACC | AUC | ACC | AUC | ACC | AUC | C-Index |
| MFMSA | $96.70_{\pm1.19}$ | $96.70_{\pm1.22}$ | $98.61_{\pm0.609}$ | $98.48_{\pm0.384}$ | $96.30_{\pm0.93}$ | $96.20_{\pm0.81}$ | $82.05_{\pm2.62}$ |
| MSCAM | $96.21_{\pm1.32}$ | $96.30_{\pm2.45}$ | $98.44_{\pm1.1}$ | $98.27_{\pm0.95}$ | $96.18_{\pm0.88}$ | $96.20_{\pm0.55}$ | $81.67_{\pm3.70}$ |
| Perceiver | $93.50_{\pm3.65}$ | $93.81_{\pm2.08}$ | $97.56_{\pm1.12}$ | $98.0_{\pm0.90}$ | $95.0_{\pm3.771}$ | $94.90_{\pm1.98}$ | $78.35_{\pm5.74}$ |
| MuMu | $95.30_{\pm1.85}$ | $95.30_{\pm2.15}$ | $98.10_{\pm0.65}$ | $98.10_{\pm0.91}$ | $95.42_{\pm1.62}$ | $95.60_{\pm1.98}$ | $80.20_{\pm4.97}$ |
| $M^3Att$ | $95.52_{\pm2.40}$ | $95.90_{\pm2.07}$ | $98.44_{\pm1.05}$ | $98.50_{\pm0.97}$ | $95.37_{\pm1.41}$ | $95.40_{\pm1.37}$ | $81.37_{\pm4.92}$ |
| HEALNet | $96.12_{\pm1.05}$ | $96.45_{\pm1.27}$ | $98.0_{\pm0.95}$ | $98.0_{\pm1.34}$ | $95.52_{\pm1.84}$ | $95.27_{\pm1.53}$ | $82.85_{\pm2.56}$ |
| DRIFA-Net | $96.85_{\pm1.12}$ | $97.10_{\pm1.17}$ | $98.76_{\pm0.59}$ | $98.44_{\pm0.61}$ | $96.30_{\pm0.85}$ | $96.0_{\pm1.15}$ | $80.92_{\pm3.43}$ |
| EHPAL-Net-18 | $\mathbf{97.54_{\pm0.51}}$ | $\mathbf{97.83_{\pm0.73}}$ | $\mathbf{99.12_{\pm0.31}}$ | $\mathbf{99.0_{\pm0.202}}$ | $\mathbf{96.95_{\pm0.64}}$ | $\mathbf{97.10_{\pm0.42}}$ | $\mathbf{85.05_{\pm1.03}}$ |

contrastive learning. The framework demonstrates label efficiency by excelling at image-text retrieval, classification (fine-tuned and zero-shot), and segmentation under limited annotation. We adapt Gloria's attention mechanism into our multimodal fusion paradigm by employing a late-fusion scheme: two ResNet-18 branches process distinct modalities concurrently, and their outputs are fused post-attention to serve as a comparative baseline against `EHPAL-Net`. The raw code is available at `https://github.com/marshuang80/gloria`.

**MuMu.** MuMu (Islam & Iqbal, 2022) is a cooperative multitask learning–based guided multimodal fusion method designed to learn robust representations for human activity recognition. It first employs auxiliary tasks to extract features tailored to each activity-group—sets of activities with shared characteristics—and then uses these group-specific features to steer its Guided Multimodal Fusion (`GM-Fusion`) module, which produces complementary multimodal embeddings for the primary recognition task. In our work, we adapt MuMu to heterogeneous medical data by integrating three distinct modalities into its architecture and leveraging all of its core functionalities as a comparative baseline against our proposed `EHPAL-Net`.

**HAMLET.** `HAMLET` (Hierarchical Multimodal Self-Attention for `HAR`) (Islam & Iqbal, 2020) first extracts spatio-temporal salient features from each unimodal input stream. It then applies its Multimodal Attention-based Feature Fusion (`MAT`) module to disentangle and fuse these unimodal representations into a unified embedding, which drives improved recognition performance. In our implementation, we adapt `HAMLET` to heterogeneous medical data by providing three distinct modality streams and retaining all original model components, using it as a comparative baseline against `EHPAL-Net`.

**Perceiver.** The Perceiver extends the Transformer paradigm to handle very large, high-dimensional inputs by leveraging an asymmetric cross-attention mechanism that iteratively distills inputs into a fixed-size latent bottleneck. It makes minimal architectural assumptions about input structure while scaling to hundreds of thousands of tokens. In our implementation, we integrate the `Perceiver's` latent-array attention modules into a `ResNet-50` backbone to process three heterogeneous medical modalities via an early fusion strategy (see Appendix B for its associated drawbacks). The raw code is available at `https://github.com/lucidrains/perceiver-pytorch`.

**MTTU-Net.** MTTU-Net is a multi-task architecture that jointly performs glioma segmentation and IDH genotyping end-to-end from multimodal MRI scans. It employs a hybrid CNN-Transformer encoder—combining convolutional layers for local spatial feature extraction with a Transformer for global context modeling—to learn shared representations. A segmentation decoder and a multi-scale classification head then leverage these features for tumor delineation and `IDH` status prediction, respectively. In our work, we retain `MTTU-Net's` full encoder design and extend it with an additional branch to process two medical modalities, using it as a comparative baseline against `EHPAL-Net`. The raw code is available at `https://github.com/miacsu/MTTU-Net`.

**M³Att.** Multi-Modal Mutual Attention (`M³Att`) employs cross-modal mutual attention modules to fuse heterogeneous feature maps for improved segmentation performance. In our adaptation, we embed the $M^3Att$ attention blocks within a `Swin-B` transformer backbone and extend the network to a dual-branch architecture—each branch processing one modality of heterogeneous medical data—while preserving all original module functionalities. This implementation serves as a comparative baseline against our `EHPAL-Net`.

**MOTCAT.** `MOTCAT` (Multimodal Optimal Transport–based Co-Attention Transformer) employs a dual-branch architecture with an intermediate fusion strategy: one branch processes imaging data while the other processes genomic data. Optimal transport–based co-attention layers align and fuse these modality-specific embeddings at intermediate depths, enabling joint representation learning across heterogeneous medical data. In our implementation, we use two modality streams—retaining its original dual-branch design—and preserve all core components as a comparative baseline against `EHPAL-Net`. The raw code is available at `https://github.com/Innse/MOTCat`.

**HEALNet.** `HEALNet` (Hybrid Early-fusion Attention Learning Network) is a flexible multimodal fusion architecture that combines early and intermediate fusion to process multiple input streams in parallel. It employs modality-specific encoders for specialized feature extraction before fusion, enabling nuanced handling of diverse data types. However, this two-stage fusion strategy results in high computational complexity when applied to heterogeneous medical datasets. The implementation is available at `https://github.com/konst-int-i/healnet`.

**DRIFA-Net.** `DRIFA-Net` (Dual Robust Information Fusion Attention Network) employs a dual-branch architecture with an intermediate fusion strategy: each branch processes a distinct imaging modality, and attention-based fusion modules align and merge the modality-specific features to enhance multimodal representation learning. In our implementation, we retain the original dual-branch design—using two imaging modalities—and preserve all core components as a comparative baseline against `EHPAL-Net`. The raw code is available at `https://github.com/misti1203/DRIFA-Net`.

Table 7: Performance comparison for evaluating model reliability through uncertainty quantification. We compare the proposed `EHPAL-Net` against top competitive baselines—`DRIFA-Net` and `HEALNet`—on four benchmark datasets: `HAM10000`, `SIPAKMeD`, `TCGA-BRCA`, and `MIMIC-III` (used only for mortality prediction). **Bold** indicates the best results.

| Datasets → | HAM10000 | SIPAKMeD | TCGA-BRCA | MIMIC III (MORT) |
|---|---|---|---|---|
| Models ↓ | AUC | AUC | C-Index | AUC |
| DRIFA-Net | 97.65 | 94.81 | 63.54 | 91.20 |
| HEALNet | 97.80 | 94.07 | 66.30 | 92.42 |
| EHPAL-Net-18 | **99.40** | **99.15** | **70.94** | **95.02** |

## J    APPENDIX – ADDITIONAL PERFORMANCE COMPARISON

**Extensive performance comparison.** The unimodal baselines (`MFMSA` and `MSCAM`), as reported in Tables 2 and 5-6, represent the best-performing models among all unimodal settings we evaluated. Compared to the stronger of these two, `EHPAL-Net` achieves up to an approximate 5% improvement across heterogeneous modality datasets. We term this gain as the multimodal uplift, quantitatively depicted in Figure 7, which shows the percentage improvement of various multimodal fusion models over their corresponding best unimodal baselines.

As shown in the Figure 7, `EHPAL-Net` variants (EHPAL-Net-18, EHPAL-Net-50, EHPAL-Net-IN, EHPAL-Net-SN) consistently yield positive uplift across all four datasets—`HAM10000`, `SIPaKMeD`, `TCGA-BRCA`, and `MIMIC-III (MORT)`—outperforming traditional early, intermediate, late, and hybrid fusion approaches such as `Perceiver`, `MuMu`, `M³Att`, and `HEALNet`. The consistent magnitude of improvement across datasets highlights the strength of `EHPAL-Net`'s efficient hybrid fusion strategy and its superior capability in leveraging heterogeneous modalities.

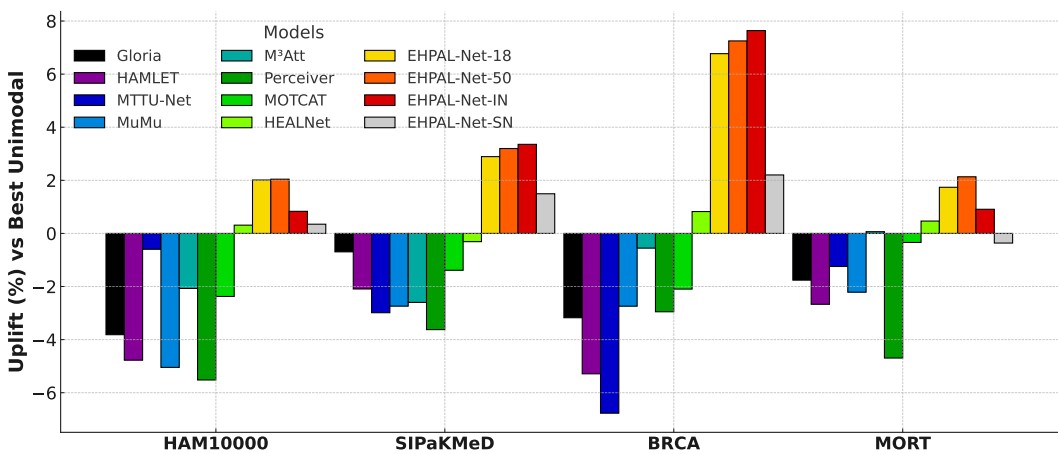

Figure 7: Mean percentage uplift of multimodal fusion models over the best unimodal baselines across four heterogeneous medical datasets. `EHPAL-Net`'s efficient hybrid fusion variants consistently outperform early, intermediate, late, and hybrid early fusion methods.

In contrast, models like `MOTCAT` and `MTTU-Net` show only limited or dataset-specific uplift, indicating less reliable performance across diverse tasks. This contrast further underscores the strong generalizability and robustness of `EHPAL-Net`.

**Uncertainty quantification and reliability assessment.** To make `EHPAL-Net` dependable in real-world deployment, we integrate *Monte Carlo Dropout* (`MCD`) into every EHF layer (dropout rate $p=0.25$). At inference, these layers stay active and we perform $T=30$ stochastic forward passes per sample, producing $T$ logits that are then averaged to yield the final prediction. This procedure supplies an empirical distribution whose spread serves as our uncertainty estimate, while the mean prediction is used to compute evaluation metrics (AUC or C-Index).

*Why it works.* `EHPAL-Net` couples *hyperbolic dual-geometry attention*, which preserves hierarchical and complex structural information (Nickel & Kiela, 2017a), with *quantum-inspired attention*, which excels at modeling long-range dependencies (Hirose, 2018). These complementary mechanisms help the network localize stochastic variations introduced by dropout and maintain stable representations.

*Empirical results.* As summarized in Table 7, `EHPAL-Net` experiences only a $\approx 0.59\%$ performance decline under stochastic inference. Competing models—`DRIFA-Net` and `HEALNet`—use heavier Euclidean fusion schemes and exhibit drops of up to $\approx 3.10\%$, highlighting their weaker ability to handle predictive uncertainty. Thus, `EHPAL-Net` not only offers state-of-the-art accuracy but also delivers robust and reliable predictions when its uncertainty is explicitly quantified.

## K   APPENDIX – ADDITIONAL ABLATION STUDY

Table 8: Ablation of different integrated components—`EMRC`, `PCMFA`, and `SIR`—within the `EHF` layer of `EHPAL-Net`, highlighting their individual contributions. Results are reported under two evaluation regimes: (i) *EHR-only* and (ii) a *mixed* setting with 50% medical-imaging data (HAM10000, SIPaKMeD) and 50% EHR data (UCI-HAR, MIMIC-III for the MORT task). See Section 4.2 for details.

| Integrated components of EHPAL-Net | | | | | | | | | |
|---|---|---|---|---|---|---|---|---|---|
| 100% EHR | | | | | 50% Medical Imaging + 50% EHR | | | | |
| EMRC | PCMFA | SIR | ACC | AUC | EMRC | PCMFA | SIR | ACC | AUC |
| ✓ | × | × | 81.0 | 86.8 | ✓ | × | × | 87.6 | 88.2 |
| ✓ | × | ✓ | 82.2 | 87.9 | ✓ | × | ✓ | 90.1 | 90.8 |
| ✓ | ✓ | × | 88.5 | 93.7 | ✓ | ✓ | × | 94.1 | 95.9 |
| × | ✓ | ✓ | 89.3 | 94.8 | × | ✓ | ✓ | 95.0 | 96.1 |
| ✓ | ✓ | ✓ | **90.2** | **96.9** | ✓ | ✓ | ✓ | **96.5** | **98.3** |

Table 9: Ablation of `PCMFA` components—`MHDGA`, `MQIA`, and `MAFG`—inside the `EHF` layer. Each experiment removes one or more of these components while keeping all other modules (e.g., `EMRC`) fixed. We compare two attention strategies—cascaded attention (`CA`) and parallel fusion attention (`PFA`)—on a heterogeneous cohort composed of 50% medical-imaging data, 25% multi-omics data, and 25% EHR data (see Section 4.2).

| Integrated components of PCMFA and CA vs. PFA | | | | | | | | | | | | | | | | | |
|---|---|---|---|---|---|---|---|---|---|---|---|---|---|---|---|---|---|
| 50% Imaging + 25% Multi-omics + 25% EHR | | | | | | | 50% Imaging + 25% Multi-omics + 25% EHR | | | | | | | | | |
| MHDGA | MQIA | MAFG | CF | PFA | ACC | AUC | C-Index | MHDGA | MQIA | MAFG | CF | PFA | ACC | AUC | C-Index | #Par | #FLOP |
| × | × | × | × | × | 91.34 | 93.83 | 64.50 | × | × | × | × | × | 82.20 | 98.03 | 65.35 | **5.02** | **0.33** |
| ✓ | ✓ | × | ✓ | × | 94.37 | 96.28 | 70.11 | ✓ | ✓ | × | ✓ | × | 86.07 | 99.20 | 69.48 | 7.63 | 0.57 |
| ✓ | ✓ | ✓ | × | ✓ | **96.02** | **98.40** | **71.27** | ✓ | ✓ | ✓ | × | ✓ | **88.54** | **99.39** | **71.74** | 7.71 | 0.59 |

**Ablations on Diverse Attentions.** Table 10 compares our `PCMFA` attention module against two prior fusion mechanisms—self-attention as used in `HEALNet` and the DRIFA block from `DRIFA-Net`—across seven heterogeneous medical datasets, organized into two-modality groups. The first group comprises two imaging datasets (HAM10000, SIPaKMeD), one multi-omics dataset (TCGA-BRCA), and one EHR dataset (MIMIC-III for mortality prediction). The second group comprises two imaging datasets (PathMNIST, OrganAMNIST), one multi-omics dataset (TCGA-UCEC), and one EHR dataset (MIMIC-III for ICD-9 prediction). For a fair comparison, all variants—including the self-attention and DRIFA baselines—are augmented with our EMRC and SIR modules, so that only the fusion mechanism differs.

Across both groups, `PCMFA` consistently outperforms the self-attention and DRIFA baselines by 0.29–3.54%, demonstrating its superior capacity to learn complementary shared representations with minimal computational cost.

Table 10: Ablation of `EHPAL-Net` (backbone: `ResNet18`) with three attention variants—self-attention (SA, as in `HEALNet`), DRIFA (from `DRIFA-Net`), and the proposed `PCMFA`. Performance is measured on a heterogeneous cohort comprising 50 % medical-imaging data, 25 % multi-omics data, and 25 % EHR data.

| Diverse Attentions | | | | | | | | | | | | | |
|---|---|---|---|---|---|---|---|---|---|---|---|---|---|
| 50% Imaging + 25% Multi-omics + 25% EHR | | | | | | 50% Imaging + 25% Multi-omics + 25% EHR | | | | | | | |
| SA | DRIFA | PCMFA | ACC | AUC | C-Index | SA | DRIFA | PCMFA | ACC | AUC | C-Index | #Parameters | #FLOPs |
| ✓ | × | × | 93.80 | 96.12 | 69.47 | × | ✓ | × | 85.73 | 98.9 | 68.20 | 9.62 | 0.65 |
| × | ✓ | × | 95.15 | 96.43 | 67.85 | × | × | × | 86.90 | 99.10 | 70.48 | 31.25 | 2.35 |
| × | × | ✓ | **96.02** | **98.40** | **71.27** | × | × | ✓ | **88.54** | **99.39** | **71.74** | **7.71** | **0.59** |

**Ablations on Integrated Components.** We ablated each attention component within `PCMFA`—PIL, LIL, their composite MHDGA, MQIA, and LF—to isolate its standalone impact in `EHPAL-Net`. Experiments on four benchmarks (`HAM10000`, `SIPaKMeD`, `TCGA-BRCA`, and `MIMIC-III`/mortality) show that the full `PCMFA` configuration attains the best performance on every dataset, surpassing any single-component variant by 3.07%–11.02% (Table 11). These gains stem from PCMFA's cooperative design: the sub-modules guide one another to capture richer, cross-modal representations, whereas isolated mechanisms (PIL, LIL, MHDGA, MQIA, LF) operate in silos and thus fail to exploit complementary cues.

Table 11: Ablation study of `EHPAL-Net` (`ResNet18` backbone). Each experiment disables exactly one component—PIL, LIL, MHDGA, MQIA, LF, or PCMFA—while keeping all other modules unchanged. Results are reported on four benchmarks: `HAM10000`, `SIPAKMeD`, `TCGA-BRCA`, and `MIMIC-III` (mortality). **Bold** marks the best score in each column.

| Datasets → | HAM10000 | SIPAKMeD | TCGA-BRCA | MIMIC III (MORT) |
|---|---|---|---|---|
| Approaches ↓ | AUC | AUC | C-Index | AUC |
| PIL | 90.81 | 92.72 | 60.25 | 87.94 |
| LIL | 92.37 | 94.85 | 62.60 | 89.10 |
| MHDGA | 95.52 | 96.60 | 67.08 | 92.46 |
| MQIA | 94.78 | 96.12 | 64.30 | 91.44 |
| LF | 90.54 | 87.90 | 60.23 | 85.71 |
| PCMFA | **99.99** | **99.67** | **71.27** | **95.55** |

Table 12: Ablation study comparing `PCMFA` with vanilla cross-attention (`CRA`). Metrics: Accuracy and C-Index.

| Approach | BRCA | MORT | HAM10000 | SIPAKMed | #Parameters | #FLOPs |
|----------|------|------|----------|----------|-------------|--------|
| CRA | 66.30 | 89.72 | 96.93 | 93.58 | 6.15 | 0.47 |
| PCMFA | 71.14 | 93.45 | 99.60 | 96.10 | 7.71 | 0.59 |

To further conduct an ablation study, we replaced the PCMFA block with a vanilla cross-attention (CRA) layer—keeping the rest of the EHF architecture unchanged—and evaluated the result (Table 12). This swap reduced performance by up to $\approx 4.9\%$, underscoring PCMFA's advantage over CRA.

**Ablations on Per-modality (Unimodal) Analysis.** To conduct experiments on each modality in multimodal learning, we trained single-modality variants of `EHPAL-Net` and compared them with unimodal versions of `HEALNet` and `DRIFA-Net` on identical data splits. As shown in Table 13, the image-only `EHPAL-Net` achieves **92.56**% accuracy on `HAM10000`, outperforming `HEALNet` by +3.33% and `DRIFA-Net` by +1.65%, respectively. On `SIPAKMED`, it achieves **88.31**% accuracy and exceeding the baselines by +1.44% and +0.86%). For the omics-only task on `TCGA-BRCA`, `EHPAL-Net` reaches a C-index of **65.74** and surpassing `HEALNet` by (+3.62% and `DRIFA-Net` by +5.11%). Its EHR-only counterpart yields **88.20**% accuracy on `MORT` and outperforming `HEALNet` by (+1.76% and `DRIFA-Net` by +2.09%). These findings underscore the strong standalone predictive power of each modality when modeled with `EHPAL-Net`.

Table 13: Per-modality (unimodal) performance (accuracy or C-index).

| Modality | Dataset | EHPAL-Net | HEALNet | DRIFA-Net |
|----------|---------|-----------|---------|-----------|
| Image only | HAM10000 | **92.56** | 89.23 | 90.91 |
| Image only | SIPAKMED | **88.31** | 86.87 | 87.45 |
| Multi-omics only | TCGA-BRCA | **65.74** | 62.12 | 60.63 |
| EHR only | MORT | **88.20** | 86.44 | 86.11 |

**Ablations on Modality-imbalance in Multimodal Fusion.** Because the three input modalities—images, electronic health records (`EHR`), and multi-omics—differ markedly in standalone accuracy, the fused model inherits modality *imbalance*: image features dominate, whereas EHR and multi-omics provide weaker yet complementary signals. Table 14 makes this explicit. For each dataset we first identify the strongest single-modality score from Table 13 ("Best unimodal"). We then compare it with a series of two- and three-modality fusions and report the absolute gain ($\Delta = Fusion - Best\ Unimodal$). Specifically, fusion is consistently beneficial, improving over the best unimodal baseline by +7.19% on `HAM10000`, +7.66% on `SIPAKMED`, +5.53% in C-index on `TCGA-BRCA`, and +4.14% on `MORT`. Two observations follow:

1. **Strength of images.** The largest gains occur whenever imaging is present, confirming that the image stream carries the most discriminative information (images $\gg$ `EHR` $\geq$ multi-omics).

2. **Complementarity of weak modalities.** Although individually weaker, `EHR` and multi-omics still add complementary information: the full three-modality configuration is best on every dataset, underscoring that balanced integration—not mere reliance on the dominant modality—yields the highest performance.

We further quantify this imbalance through a $\Delta$-gap analysis and systematic drop-one ablations ($-$Image, $-$`EHR`, $-$Multi-omics), which isolate each modality's contribution and demonstrate how our fusion module reweights them to mitigate performance skew.

**Ablations on Handling Modality Dominance.** We explicitly guard against modality dominance at the architecture level, and the empirical results confirm that no modality is hurt when others are added.

1. Architectural level:

   (a) Modality-specific gating in the MAFG block — Each modality has a learnable gate that scales its hyperbolic- and quantum-attention maps before fusion, allowing the network to down-weight an over-confident modality and up-weight a weaker one.

Table 14: Fusion results and modality-imbalance analysis. "Best unimodal" is the top single-modality score from Table 13. "Fusion" lists the score for each modality combination; $\Delta$ is the absolute gain over the best unimodal. Accuracy is reported for HAM10000, SIPAKMED, and MORT; C-index for TCGA-BRCA. Higher is better.

| Dataset | Best unimodal | Fusion setting | Fusion score | $\Delta$ |
|---|---|---|---|---|
| HAM10000 | 92.56 | Image + Multi-omics | 95.84 | +3.28 |
| | | Image + EHR | 97.27 | +4.71 |
| | | Image + Multi-omics + EHR | **99.75** | **+7.19** |
| SIPAKMED | 88.31 | Image + Multi-omics | 91.30 | +2.99 |
| | | Image + EHR | 94.41 | +6.10 |
| | | Image + Multi-omics + EHR | **95.97** | **+7.66** |
| TCGA-BRCA | 65.74 | Image + Multi-omics | 69.23 | +3.49 |
| | | Multi-omics + EHR | 68.18 | +2.44 |
| | | Image + Multi-omics + EHR | **71.27** | **+5.53** |
| MORT | 88.20 | Image + EHR | 91.03 | +2.83 |
| | | Multi-omics + EHR | 90.16 | +1.96 |
| | | Image + Multi-omics + EHR | **92.34** | **+4.14** |

(b) Residual cross-modal update with per-modality scale — Leverages two learnable gates: that rescales each modality's features before they interact, thereby the shared representation at layer is updated as $x_i^s = x_i' \odot A_{i-1} \odot \alpha_i$, so a high-performing modality cannot swamp a weaker one.

(c) Sequential EHF layers — Process modalities one-by-one (image $\rightarrow$ omics $\rightarrow$ EHR $\rightarrow$ sensor). Each new stream meets the shared representation from the previous stage instead of all modalities merging at once, which limits gradient competition and lets the weaker modality refine the shared representations later in the pipeline.

(d) Residual EMRC + SIR blocks — Preserve modality-specific paths alongside the shared path, so original modality information is never overwritten.

(e) Learned late-fusion weights inside every EHF layer — After the two PCMFA streams, we apply a trainable late-fusion so that the contribution of each modality to the final shared feature is learned, not fixed.

Together, these mechanisms give weaker modalities a dedicated, tunable influence rather than letting them be drowned out by a stronger counterpart.

2. Empirical evidence that no modality is harmed:

Across 15 heterogeneous datasets, every EHPAL-Net variant shows a positive uplift over the best single-modal baseline (ref. Fig. 7); we never observe a performance drop when additional modalities are fused (see Tables 2–7).

To isolate the effect of the gating mechanism, we compared fusion with versus without gates on four representative datasets (ref. Table 15).

Table 15: Impact of modality-specific gating on fusion performance. Accuracy (%) is reported for HAM10000, SIPaKMeD, and MORT; C-index for BRCA. $\Delta$ denotes the absolute gain achieved by enabling gating.

| Setting | HAM10000 (Acc ↑) | SIPaKMeD (Acc ↑) | BRCA (C-Index ↑) | MORT (Acc ↑) |
|---|---|---|---|---|
| Fusion + gating | 99.75 | 95.97 | 71.27 | 92.34 |
| Fusion w/o gating | 95.81 | 91.43 | 64.95 | 87.62 |
| $\Delta$ (gain) | +3.94 % | +4.54 % | +6.32 % | +4.72 % |

Disabling $c_i$ and $\alpha_i$ consistently reduces performance, confirming that the gates prevent any strong modality from degrading a weaker one.

Mixed-modality runs (Imaging + Omics + EHR) and single-domain tests likewise show only positive gains, demonstrating that weaker modality benefits from multimodal fusion.

*In summary, the performance gains of our components stem from the following key innovations: (1) **EMRC** enhances multi-scale spatial details at low computational cost. (2) **PCMFA** employs **MHDGA***

*and MQIA within a Hyperbolic-Quantum Mutual Guidance block, leveraging mutual guidance to learn richer shared representations. (3) **MHDGA** computes Poincaré and Lorentz attention weights in parallel, capturing hierarchical non-Euclidean structures. (4) **Parallel Fusion** of **MHDGA** and **MQIA** balances hyperbolic and quantum-inspired cues and avoids information loss inherent to cascaded design. (5) **SIR** increases representational diversity, further boosting complementary shared information.*

## L    APPENDIX – EXTENSIVE EXPERIMENTS AND RESULTS

**Extensive EHPAL-Net Evaluation.**    We further evaluate the performance of our proposed `EHPAL-Net` against top competitive baselines—`MSCAM`, `HEALNet`, and `DRIFA-Net`—on two additional datasets of different modalities: the endoscopy-based `KVASIR` dataset (Pogorelov et al., 2017) and the ECG-based MIT-BIH Arrhythmia dataset (Moody & Mark, 1982; 1990), using an input image size of $224 \times 224 \times 3$. Our results demonstrate that `EHPAL-Net` consistently outperforms existing methods, achieving improvements of up to 5.22% (ref. Table 16).

*It is important to note that, in this experiment, we leverage only two modalities to demonstrate that our proposed `EHPAL-Net` remains effective under limited modality support—similar to most existing works—and still achieves superior performance.*

**Extensive Reliability Assessment (Label noise adaptation).**    We further evaluate the reliability of our proposed `EHPAL-Net` on the `HAM10000`, `SIPAKMeD`, and `MIT-BIH` datasets under varying levels of symmetric label noise (Han et al., 2018), with noise rates set at 10%, 20%, 30%, and 40%. This setup simulates annotation inaccuracies by uniformly corrupting class labels across all categories. By systematically increasing the noise rate, we assess the robustness of our framework to label noise, observing a performance degradation of up to 42% at the highest noise level. Nevertheless, `EHPAL-Net` consistently outperforms competitive baselines—`DRIFA-Net` and `HEALNet`—achieving improvements of up to 4.7% (see Table 17), thereby highlighting its resilience and robust decision-making capabilities under noisy conditions.

Table 16: Performance comparison of `EHPAL-Net` integrated with the ResNet18 base network (referred to as `EHPAL-Net-18`) against SOTA methods on two medical datasets: `KVASIR` and `MIT-BIH`, across multiple tasks. **Bold** indicates the best results.

| Datasets → | KVASIR | | | MIT-BIH | | |
|---|---|---|---|---|---|---|
| Models ↓ | ACC | F1 | AUC | ACC | F1 | AUC |
| MSCAM | $89.52_{\pm 3.58}$ | $89.30_{\pm 2.72}$ | $89.61_{\pm 5.08}$ | $99.27_{\pm 0.25}$ | $99.18_{\pm 0.18}$ | $99.50_{\pm 0.05}$ |
| HEALNet | $89.70_{\pm 5.18}$ | $89.45_{\pm 4.71}$ | $90.0_{\pm 6.90}$ | $99.10_{\pm 0.34}$ | $99.10_{\pm 0.25}$ | $99.20_{\pm 0.53}$ |
| DRIFA-Net | $91.47_{\pm 4.12}$ | $91.10_{\pm 3.17}$ | $91.60_{\pm 4.59}$ | $99.60_{\pm 0.15}$ | $99.50_{\pm 0.20}$ | $99.60_{\pm 0.15}$ |
| EHPAL-Net-18 | $\mathbf{94.74}_{\pm 1.51}$ | $\mathbf{94.30}_{\pm 1.31}$ | $\mathbf{94.52}_{\pm 1.25}$ | $\mathbf{100}_{\pm 0.0}$ | $\mathbf{100}_{\pm 0.0}$ | $\mathbf{100}_{\pm 0.0}$ |

Table 17: Performance comparison in terms of accuracy for evaluating model reliability under different levels of label noise. The proposed `EHPAL-Net` is compared against top competitive baselines—`DRIFA-Net` and `HEALNet`—on three benchmark datasets: `HAM10000`, `SIPAKMeD`, and `MIT-BIH`. **Bold** indicates the best results.

| Datasets → | HAM10000 | | | SIPAKMeD | | | MIT-BIH | | |
|---|---|---|---|---|---|---|---|---|---|
| Label Noise Rate ↓ | DRIFA-Net | HEALNet | EHPAL-Net | DRIFA-Net | HEALNet | EHPAL-Net | DRIFA-Net | HEALNet | EHPAL-Net |
| 0.1 | 84.1 | 83.7 | **85.4** | 81.5 | 82.8 | **86.2** | 87.7 | 87.1 | **89.9** |
| 0.2 | 73.7 | 73.2 | **75.9** | 71.3 | 71.8 | **76.1** | 78.6 | 79.4 | **80.0** |
| 0.3 | 63.5 | 63.5 | **66.8** | 62.9 | 61.5 | **67.4** | 68.7 | 68.4 | **70.0** |
| 0.4 | 51.3 | 53.0 | **58.2** | 50.7 | 52.5 | **57.8** | 58.4 | 57.9 | **60.9** |

**Handling Missing Modalities.**    `EHPAL-Net` is jointly trained on four modalities (`HAM10000` (dermoscopy), `SIPAKMED` (cytology), `BRCA` (multi-omics), and `MORT` (electronic health records))—and assesses robustness when modalities are absent at inference. We consider two complementary test settings:

1. *Non-imaging-only* (imaging omitted): `HAM10000` and `SIPAKMED` held out; the model sees `BRCA` and `MORT`.

2. *Imaging-only* (non-imaging omitted): `BRCA` and `MORT` held out; the model sees `HAM10000` and `SIPAKMED`.

We evaluate `MOTCAT`, `DRIFA-Net`, `HEALNet`, and our `EHPAL-Net` under both settings (Table 18), and also when all four modalities are available (e.g., Table 2). Across the two missing-modality scenarios, `EHPAL-Net` consistently achieves the best performance among the fusion methods, with at most an $\approx 13.64\%$ drop relative to the all-modalities condition, whereas the strongest baseline (`HEALNet`) drops considerably more. These results confirm that `EHPAL-Net` delivers superior robustness to missing inputs.

This robustness arises from the interplay of the `PCMFA` and `LF` modules. The `PCMFA` module (i) adaptively reweights the observed modalities, (ii) suppresses incoherent information, and (iii) preserves fine-grained structure, yielding graceful degradation when inputs are missing. The `LF` layer (§3.1.3) further strengthens robustness through a mask-aware attention mechanism that nullifies missing-modality contributions during both training and inference. Together, these mechanisms allow `EHPAL-Net` to sustain state-of-the-art accuracy even when half of the input modalities are withheld.

Table 18: Robustness to missing modalities. Comparison of `MOTCAT`, `DRIFA-Net`, and `EHPAL-Net`—each pretrained on four modalities (`HAM10000` dermoscopy, `SIPAKMED` cytology, `BRCA` multi-omics, `MORT` EHR)—under two inference settings: *imaging-only* (`BRCA` + `MORT` held out) and *non-imaging-only* (`HAM10000` + `SIPAKMED` held out). Metrics are accuracy (%) for `HAM10000`, `SIPAKMED`, and `MORT`, and `C-Index` (%) for `BRCA`.

| Missing Modalities | Test Modality | MOTCAT | DRIFA-Net | HEALNet | EHPAL-Net |
|---|---|---|---|---|---|
| HAM10000 + SIPAKMED | BRCA | 51.88 | 52.23 | 57.80 | 60.74 |
| | MORT | 76.46 | 77.15 | 82.62 | 85.54 |
| BRCA + MORT | HAM10000 | 81.12 | 82.70 | 86.58 | 89.41 |
| | SIPAKMED | 72.37 | 73.42 | 80.81 | 82.33 |

Table 19: Performance when swapping the modality order (higher is better). Best per column in **bold**.

| Models | BRCA | MORT | HAM10000 | SIPAKMED |
|---|---|---|---|---|
| MOTCAT | 65.05 | 90.38 | 95.87 | 93.24 |
| DRIFA-Net | 66.63 | 91.20 | 98.41 | 95.58 |
| EHPAL-Net | **71.14** | **93.45** | **99.60** | **96.35** |

**Input Order-Robustness in Fusion.** `EHPAL-Net` is *permutation-invariant*: re-ordering the input modalities—medical images (e.g., dermoscopy (`HAM10000`), cytology (`SIPAKMED`), multi-omics (e.g., `BRCA`), and clinical (`EHR`)—does not affect its predictions. Although the data stream sequentially passes through cascaded EHF layers, each layer's `PCMFA` and `LF`-gating blocks dynamically re-weight cross-modal dependencies, keeping both intermediate representations and final outputs stable. Table 19 empirically confirms this invariance.

The first `EHF` layer fuses modalities $m_1$ and $m_2$—for example, dermoscopy (`HAM10000`) and cytology (`SIPAKMED`), producing a joint representation $\{x_i^{s'}\}_{i=1}^m$. The next layer fuses $\{x_i^{s'}\}_{i=1}^m$ with $m_3$ (multi-omics; `BRCA`), and the process continues until all $m$ modalities are integrated. Swapping the order (e.g., feeding omics before imaging) only adjusts internal attention paths; `PCMFA` and `LF`-gating promptly reweight these paths, preserving the robust shared representation. Table 19 empirically confirms `EHPAL-Net`'s performance invariance across all permutations.

**Handling Misaligned or Asynchronous Modalities** In clinical practice, multimodal data are rarely synchronized. `EHPAL-Net` employs a sequential, pair-wise fusion strategy in its `MSIL` phase: This design allows heterogeneous inputs—imaging, multi-omics, and clinical—to be processed in any order (ref. Table 19), without requiring modality synchronization or explicit alignment. Concretely, given $n$ heterogeneous mods, the `MSIL` phase processes inputs sequentially as follows:

1. The first `EHF` layer jointly integrates modality 1 and modality 2—for example, dermoscopy (`HAM10000`) and cytology (`SIPAKMED`)—to capture shared representations $x_i^{s'}$, where $i = 1$.

2. For step $i$ ($3 \le i \le m$), the $(i-1)$-th `EHF` layer fuses $\{x_i^{s'}\}_{i=1}^{m}$ with the next $i$-th modality (e.g., multi-omics; `BRCA`), yielding updated richer shared representations $x_i^{s'}$.

3. This process repeats until all $i = [1:m]$ modalities have been incorporated, with no need for explicit alignment.

Within each `EHF` layer (Fig. 2 and Algo. 2), we perform the below steps:

1. `EMRC` captures multi-scale spatial cues.

2. `PCMFA` learns shared representations in hyperbolic-quantum space without external alignment and sample-adaptively attenuates missing inputs.

3. `LF` gating adaptively modulates the shared features, further mitigating modality missing.

4. `SIR` promotes diverse robust shared representations.

*Notably, Each modality is first reshaped and augmented to a common spatial size before being fed into EHF.* Hence, `EHPAL-Net` is capable of processing any modalities, in any order, and without requiring spatial or structural alignment. This makes it highly suitable for real-world clinical settings, where imaging, omics, and `EHR` data are often collected independently and may not be perfectly matched (Tab. 19).

**Comparison with Recent Baselines.** We also compare our `EHPAL-Net` against more recent multimodal fusion baselines, such as `LegoFuse` (Hemker et al., 2025b) and `CA-MLIF` (An et al., 2025), where we have seen our `EHPAL-Net` still outperformed the existing more recent multimodal fusion baselines to achieve optimal performance improvements by up to $\approx 3.11\%$ while maintaining minimal computational cost in terms of parameters by up to $\approx 72.1\%$ and `FLOPs` up to $\approx 85.18\%$ than these existing multimodal fusion baselines (ref. Table 20). Specifically, compared with `LegoFuse` (Hemker et al., 2025b) (27.6 M params, 2.74 GFLOPs) and `CA-MLIF` (An et al., 2025) (4.1 M params, 3.98 GFLOPs), our `EHPAL-Net` (7.71 M params, 0.59 GFLOPs) achieves richer cross–modal interactions via the proposed `PCMFA` at low cost. Within `PCMFA`, two mutually guided streams—`MHDGA` (hyperbolic dual–geometry attention) and `MQIA` (quantum-inspired attention)—capture hierarchical structural details and long-range dependencies, respectively. A learnable gating module then adaptively fuses these priors, preserving structure and suppressing noise.

Our baseline multimodal fusion model (backbone: ResNet-18 without `EMRC`/`PCMFA`/`SIR`) attains $\approx 77.9\%$ mean performance with 12.1 M parameters and 0.65 GFLOPs. Introducing `EMRC` reduces parameters by $\approx 54\%$ (to 5.53 M) and `FLOPs` by $\approx 42\%$ (to 0.38 GFLOPs), while providing a modest performance lift by capturing multi-scale spatial information. Adding `PCMFA` introduces only $\approx 2$ M parameters and 0.17 GFLOPs, yet boosts performance by up to $\approx 10\%$ through complementary shared-representation learning. Finally, `SIR` adds merely $\approx 0.18$ M parameters and 0.04 GFLOPs, yielding a further $\approx 2.1\%$ performance gain by refining the shared representation.

These design choices enable `EHPAL-Net` to exceed recent baselines by up to $\approx 3.11\%$ while using $\approx 72.1\%$ fewer parameters than `LegoFuse` and up to $\approx 85.2\%$ fewer `FLOPs` than both competitors—demonstrating state-of-the-art performance with exceptional computational efficiency.

Table 20: Performance comparison with recent multimodal fusion baselines.

| Datasets → | HAM10000 | | SIPAKMeD | | TCGA-BRCA | MIMIC III (MORT) | | | |
|---|---|---|---|---|---|---|---|---|---|
| Models ↓ | Acc | AUC | Acc | AUC | C-Index | Acc | AUC | #Parameters | #FLOPs |
| LegoFuse | 98.73 | 98.84 | 95.60 | 96.25 | 69.12 | 91.51 | 94.70 | 27.6 | 2.74 |
| CA-MLIF | 96.64 | 96.90 | 94.23 | 94.20 | 66.38 | 90.77 | 94.15 | 4.1 | 3.98 |
| EHPAL-Net-18 | **99.75** | **99.99** | 95.57 | **99.67** | **71.27** | **92.34** | **95.55** | 7.71 | 0.59 |

## M   APPENDIX – DISCUSSION

**Effective Structure-preservation with efficient multimodal fusion.** Effective multimodal fusion requires balancing two crucial objectives: (1) preserving modality-specific structural properties to facilitate enhanced complementary representation learning and (2) maintaining low computational cost. Existing fusion paradigms in multimodal fusion learning methods highlight this trade-off:

- Early fusion: Directly concatenates low-level features, which can overwhelm downstream layers with noisy or misaligned low-level signals, resulting in overparameterization and attenuated modality-specific cues.

- Intermediate fusion: Projects all modalities into a shared latent space but often fails to separate shared semantics from modality-specific information.

- Late fusion: Preserves discriminative features but forfeits opportunities for cross-modal interaction during representation learning.

HEALNet (Hemker et al., 2024) employs a hybrid early–intermediate fusion approach, utilizing iterative cross-attention, and attempts to mitigate these issues by preserving low-level modality structures and facilitating cross-modal interactions. However, its reliance on large, high-dimensional attention matrices early-to-intermediate fusion incurs substantial computational and memory overhead—particularly limiting its applicability in resource-constrained medical-AI settings. Furthermore, its architecture lacks a dedicated late fusion mechanism to explicitly reintegrate high-level, modality-specific semantic cues after iterative intermediate fusion. This may limit the model's ability to fully exploit complementary modality-specific discriminative information.

EHPAL-Net addresses these limitations through a hybrid intermediate-late fusion strategy, offering a more effective and scalable alternative. By independently learning multi-scale spatial details for each modality via the EMRC module, EHPAL-Net preserves high-level semantic structures. The PCMFA module captures fine-grained cross-modal dependencies in the frequency domain through intermediate information fusion, facilitating the learning of enhanced complementary shared representations for each modality input. Subsequent late fusion integrates these complementary cues, ensuring the retention of both shared cross-modal patterns and thereby capturing more expressive multimodal representations, without the reliance on early fusion or the intensive iterative process used in HEALNet (Hemker et al., 2024). This modular fusion design enhances flexibility, reduces computational costs, and improves the model's ability to capture complementary shared patterns, which is particularly crucial for effective understanding of heterogeneous medical data.

Additionally, leveraging the SIR module further enriches representations by capturing multi-scale spatial patterns after each PCMFA module for each modality input. This process enhances the diversity of representational learning in the dual domain while maintaining minimal computational costs.

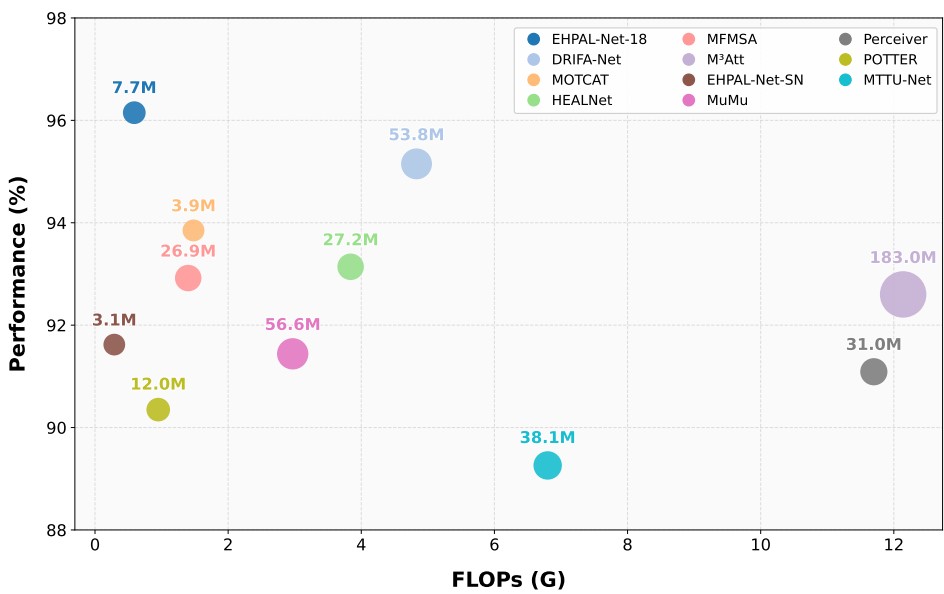

Figure 8: Average performance comparison among single-modality learning models (e.g., MFMSA), computationally intensive multimodal fusion learning (MFL) methods (e.g., DRIFA-Net (Dhar et al., 2024), MuMu (Islam & Iqbal, 2022)), HEALNet (Hemker et al., 2024), and our efficient and effective EHPAL-Net across eight benchmark datasets: HAM10000, SIPaKMeD, TCGA-BRCA, MIMIC-III (MORT and ICD9 tasks), PATHMNIST, OrganAMNIST, and TCGA-UCEC.

Empirical results (Table 2 and Fig. 8) demonstrate that `EHPAL-Net` consistently outperforms existing Multimodal Fusion Learning (`MFL`) baselines—including `HEALNet`—across all employed heterogeneous medical datasets, achieving performance improvements of up to 3.97% while significantly reducing model parameters by up to 85.7% and `FLOPs` by up to 87.8% compared to leading competitors. These findings confirm that fusing at multiple semantic levels results in more effective and efficient multimodal integration, making `EHPAL-Net` especially suitable for real-world medical AI applications where **effectiveness, efficiency, and generalizability** are essential.

**Effective and Efficient Cross-Modal Learning.** Effective cross-modal learning must reconcile two objectives: preserving modality-specific semantics and capturing rich inter-modal interactions without incurring prohibitive computational cost. Late-fusion strategies—where each modality is processed independently and only combined at the end—fail to expose the model to rich cross-modal relationships during representation learning, as evidenced by the relatively modest uplift of late-fusion baselines in Fig. 7. Intermediate-fusion methods (for example, `DRIFA`'s dual-stage attention or `HEALNet`'s multimodal self-attention) do unlock richer interactions, but they do so at the expense of drastically increased parameter counts and `FLOPs`—and still achieve only modest improvements in complementary shared representations, as reflected again by the moderate uplift seen in Fig. 7.

To bridge these gaps, we introduce a Physics-inspired attention paradigm within our `PCMFA` module, which jointly learns two specialized attention blocks in parallel with mutually guiding each other:

**M**ultimodal **H**yperbolic **D**ual-**G**eometry **A**ttention (`MHDGA`): **H**yperbolic **n**eural **n**etworks (`HNNs`) leverage the constant negative curvature of hyperbolic spaces—most commonly instantiated as the Poincaré ball or Lorentz hyperboloid models—to embed hierarchical (e.g., tree-like) structures in low-dimensional manifolds more compactly than Euclidean approaches, often outperforming high-dimensional CNNs and Transformers on complex tasks. These deep learning models embed features in Euclidean space, which distorts many real-world modalities—such as anatomical structures in medical imaging—that exhibit intrinsically non-Euclidean geometries. Recent work introduces gyrovector operations and Möbius transformations to enable principled learning in constant-curvature spaces, improving stability and expressivity. Building on these advances, our hyperbolic attention stream projects each modality's multi-scale spatial details into both the Poincaré ball and the Lorentz hyperboloid in parallel, generating dual attention weights that faithfully capture complex hierarchical relationships across modalities.

**M**ultimodal **Q**uantum-**I**nspired **A**ttention (`MQIA`): The recent surge in quantum-inspired self-attention highlights the benefits of simulating quantum mechanics concepts—such as superposition and entanglement—to enhance representational expressivity with fewer parameters. Classical self-attention computes full pairwise correlations, which is costly; by contrast, quantum self-attention frameworks map features into a simulated Hilbert space and use the Born rule to derive compact, amplitude-based affinities. Inspired by the Quantum Complex-Valued Self-Attention Model of Chen et al., `MQIA` encodes each modality's spectral components as complex quantum states and applies the Born rule to compute amplitude bias vectors. These channel-wise biases are then injected into a learnable-curvature Lorentz manifold via Minkowski inner-product modulation, guiding the hyperbolic attention stream to fuse non-local spectral cues with hierarchical context.

This dual-stream fusion eschews the need for heavy convolutional blocks and large dense attention matrices. Instead, it relies on compact hyperbolic mappings and amplitude computations, thus drastically reducing parameter counts and `FLOPs` compared to existing intermediate-fusion based attentions (e.g., `DRIFA` in `DRIFA-Net` (Dhar et al., 2025) or self-attention in `HEAL-Net` (Hemker et al., 2024)).

When integrated into `EHPAL-Net`'s `PCMFA` module, the synergistic fusion of `MQIA` and `MHDGA` enables learning of richer and more discriminative shared representations by capturing complex frequency-domain hierarchical structures. This results in significant performance improvements across heterogeneous medical imaging benchmarks, demonstrating the promise of quantum-hyperbolic fusion for scalable and effective cross-modal learning in real-world applications.

**Computational Complexity.** A key advantage of `EHPAL-Net`'s sequential fusion—akin to `HEALNet`'s (Hemker et al., 2024)—is its linear scaling in both the number of modalities $m$ (here $m = 4$) and the number of samples $n$. We replace each of `HEALNet`'s per-modality hybrid early-

Table 21: Module-level complexity.

| Module | Per-layer Cost |
|---|---|
| `EMRC` or `PIL` or `LIL` or `MQIA` or `SIR` or LF gating | $O(n)$ |
| Total per `EHF` layer | $6 \times O(n) = O(n)$ |
| Entire `MSIL` phase ($m$ layers) | $m \times O(n) = O(mn)$ |

fusion layers with one efficient hybrid fusion (`EHF`) module per modality (so for both `HEALNet` and `EHPAL-Net`, $e = m$).

In `HEALNet`, there are $e$ hybrid early-fusion layers—one per modality—so $e = m$. Each layer performs $m$ cross-attention operations and $m$ self-normalising updates—each costing $\mathcal{O}(n)$ per modality—giving

$$m\,\mathcal{O}(n) + m\,\mathcal{O}(n) = \mathcal{O}(m\,n)$$

per layer, and across $e$ layers

$$\mathcal{O}(e\,m\,n).$$

In `EHPAL-Net`, each `EHF` module (one per modality) executes:

- One `EMRC`→`PCMFA`→`SIR` sequence: $\mathcal{O}(n)$,
- Two `PCMFA` streams (dual hyperbolic + quantum): $3\,\mathcal{O}(n)$,
- One `LF` module: $\mathcal{O}(n)$,
- One `SIR` module: $\mathcal{O}(n)$.

Thus, each module costs

$$1 \cdot \mathcal{O}(n) + 3 \cdot \mathcal{O}(n) + 1 \cdot \mathcal{O}(n) + 1 \cdot \mathcal{O}(n) = 6\,\mathcal{O}(n) = \mathcal{O}(n).$$

With $e = m$ modules in total, the overall complexity is

$$m \times \mathcal{O}(n) = \mathcal{O}(m\,n),$$

dropping constant factors. Under $m = 4$, this is $\mathcal{O}(24\,n)$, i.e. still $\mathcal{O}(n)$ for fixed $m$. By comparison, cascaded-attention based `MFL` baselines (e.g., `DRIFA-Net`) and Kronecker-fusion based `MFL` methods (e.g., `MOTCAT`) scale as $\mathcal{O}(m^2\,n)$.

**Variable definitions:**

- $m$: number of modalities, here $m = 4$.
- $n$: number of training samples.
- $e$: number of fusion layers (for both `HEALNet` and `EHPAL-Net`), here $e = m$.

To show computational complexity analysis based on each module-level breakdown, explicitly detailing each component's computational cost. Each core module processes every sample exactly once per `EHF` layer; therefore, the added functionality is linear-time, i.e., $O(n)$ cost. Thus, an `EHF` layer—comprising `EMRC`, the three `PCMFA` streams (`PIL`, `LIL`, `MQIA`), `SIR`, & `LF`—runs in $O(n)$; with $m$ mods, the full pipeline is $O(mn)$ (ref. Table 21). Empirically, adding `EHF` layers reduces EN's footprint from 12.1 M→7.71 M params & from 0.65→0.59 `GFLOPs`, confirming that the added functionality comes at minimal computational cost—all linear in $n$.

To better quantify the computational benefits of our design, Table 22 compares the complexity of `EHPAL-Net-18` with that of `DRIFA-Net` under our heterogeneous multimodal setting, using the same `ResNet-18` backbone and input resolution. `EHPAL-Net-18` requires only 7.71M parameters and 0.59 `GFLOPs` per multimodal forward pass, whereas `DRIFA-Net` uses 53.8M parameters and 4.83 `GFLOPs`. This corresponds to roughly 85.7% fewer parameters and 87.8% fewer `FLOPs`. In practice, the model size on disk decreases from 205.55 MB to 29.43 MB (about $7\times$ smaller), and the wall-clock inference time is reduced from 6.53 s to 0.683 s per forward pass (about $9.6\times$ speed-up), measured on the same hardware.

Crucially, these efficiency gains do not come at the cost of accuracy. As reported in Tables 2, 5, and 6, `EHPAL-Net-18` consistently outperforms `DRIFA-Net` across all fifteen datasets in terms

of `ACC`, `AUC`, and `C-Index`. This confirms that replacing `DRIFA-Net`'s quadratic-cost dual-attention fusion with our linear-cost `EHF` layer—which combines `EMRC`, the physics-inspired `PCMFA`, and `SIR`—yields richer complementary shared representations at substantially lower computational cost. The resulting performance–efficiency trade-off makes `EHPAL-Net` particularly suitable for deployment in resource-constrained healthcare settings.

Table 22: Complexity cost analysis of `EHPAL-Net-18` when conducting on heterogeneous medical datasets against prior `DRIFA-Net`.

| Method | # Parameters (M) | # FLOPs (G) | # Model Size (MB) | Inference Time (sec.) |
|---|---|---|---|---|
| DRIFA-Net | 53.8 | 4.83 | 205.55 | 6.53 |
| EHPAL-Net-18 | 7.71 | 0.59 | 29.43 | 0.683 |

# N   APPENDIX – QUALITATIVE ANALYSIS

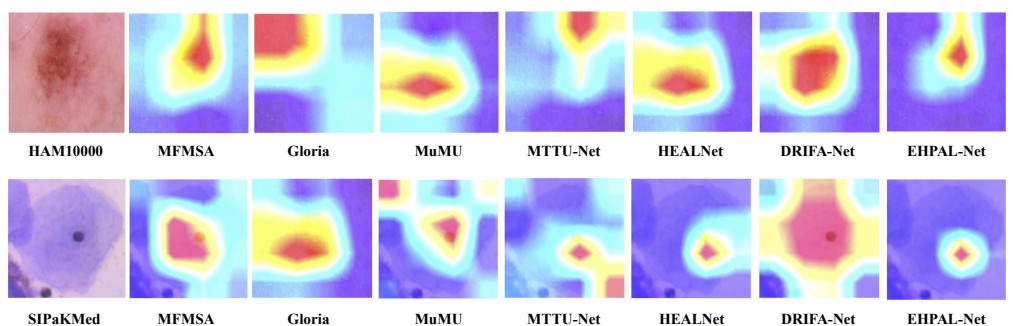

Figure 9: Visual comparison of discriminative regions highlighted by our proposed `EHPAL-Net` variants (e.g., `EHPAL-Net-18`) and seven top-performing state-of-the-art methods using the `Grad-CAM` technique on two benchmark datasets: `HAM10000` (top row) and `SIPaKMeD` (bottom row).

In the `Grad-CAM` visualizations shown in Fig. 9, POTTER lights up broad, unfocused areas—much of which is irrelevant background noise—while MuMu reduces this noise but still highlights non-informative regions. Both `MFMSA` and `HEALNet` predominantly focus on the target structure, though they still produce occasional spurious highlights in surrounding regions. `DRIFA-Net` achieves sharper boundary localization; however, subtle background artifacts may still persist around object edges. In contrast, `EHPAL-Net` demonstrates superior focus by effectively suppressing irrelevant cues and precisely attending to the underlying anatomical or structural regions—achieving the cleanest and most distinct separation between relevant and irrelevant features.

**Grad-CAM Comparison of Attention Mechanisms.**   Figure 10 presents a qualitative analysis using Grad-CAM visualizations, highlighting the lesion-focused areas that most influence the model's classification decisions. In the case of the HAM10000 dataset, these correspond to visible skin lesions, while in SIPaKMeD, they reflect abnormal cervical cell regions. The visualizations demonstrate how each attention mechanism—PA, LA, MHDGA, and MQIA—focuses on relevant feature regions, in comparison to the sharply concentrated and noise-suppressive focus achieved by our proposed Physics-inspire Cross-modal Fusion Attention (PCMFA) module within `EHPAL-Net`.

Poincaré Attention (PA) exhibits broadly dispersed visualizations, frequently highlighting broad and non-discriminative image regions. This widespread focus introduces substantial visual noise and reduces interpretability by masking the class-informative features necessary for robust classification.

Lorentz Attention (LA) demonstrates improved localization, concentrating attention more effectively around targeted structures. However, it still highlights peripheral areas that are less informative, indicating limited suppression of irrelevant cues.

MHDGA offers a clearer and more structured focus on significant regions compared to PA and LA. It refines boundaries around lesion areas more effectively. Nevertheless, minor residual highlights remain at the lesion periphery, suggesting incomplete filtering of irrelevant information.

MQIA shows greater refinement and specificity, significantly minimizing irrelevant heatmap noise compared to hyperbolic-only methods. Despite this improvement, MQIA occasionally retains faint, non-diagnostic highlights, particularly visible as mild peripheral artifacts.

In contrast, the proposed PCMFA module generates the most refined and interpretable heatmaps, distinctly highlighting crucial areas with exceptional clarity. It effectively suppresses nearly all irrelevant visual noise and peripheral distractions, thereby facilitating a clearer understanding of the model's decision-making behavior.

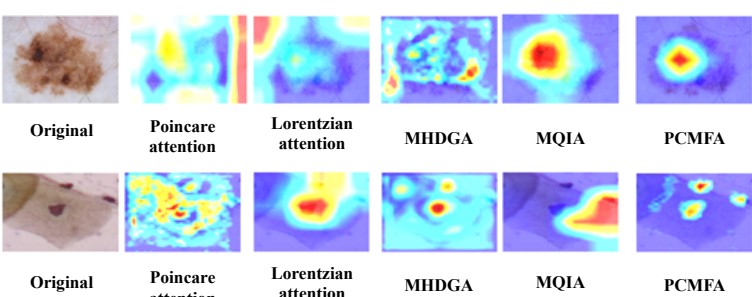

Figure 10: Grad-CAM-based visual comparison of discriminative regions highlighted by our proposed `PCMFA` module and its constituent attention components—PA, LA, MHDGA, and MQIA—on two benchmark datasets: `HAM10000` (top row) and `SIPaKMeD` (bottom row). The visualizations demonstrate the enhanced focus and noise suppression achieved by `PCMFA` relative to its individual attention mechanisms.

**t-SNE Plot Comparison of Attention Mechanisms.** In this study, Figure 11 presents t-SNE visualizations of feature embeddings produced by each attention mechanism (ref. Table 23), when individually integrated into our `EHPAL-Net`, as follows:

**PA:** • **Observation:** Clusters are poorly formed and heavily overlapping. There is poor separation between different classes.
- **Justification:** This indicates that the learned feature representations under Poincaré Attention lack discriminative power, likely due to broad and diffuse attention focus, as seen in qualitative visualizations. It struggles to isolate class-specific structures in latent space.

**LA:** • **Observation:** Slightly tighter clusters than Poincaré, but still considerable overlap.
- **Justification:** Lorentzian geometry provides better local structure preservation, aiding in marginally better inter-class separation. However, irrelevant cues are still not fully filtered, limiting cluster compactness.

**MHDGA:** • **Observation:** More distinct and tighter clusters than both PA and LA. There is better inter-cluster margin and reduced overlap.
- **Justification:** MHDGA combines Lorentz and Poincaré representations, enabling complementary hierarchical encoding. This fusion improves the ability to learn non-Euclidean structures relevant to class distinctions.

**MQIA:** • **Observation:** Clusters are more compact than those from PA and LA but remain less distinct and more overlapping than MHDGA.
- **Justification:** Quantum-inspired embeddings enrich feature expressiveness and reduce some noise, but they do not achieve the same inter-cluster margins as MHDGA, resulting in only moderate class separability.

**PCMFA:** • **Observation:** This module shows the most distinct, compact, and well-separated clusters. Minimal overlap is observed.
- **Justification:** Integrating MHDGA and MQIA within a mutual-guidance framework under physics-based constraints yields highly discriminative, modality-aligned representations.

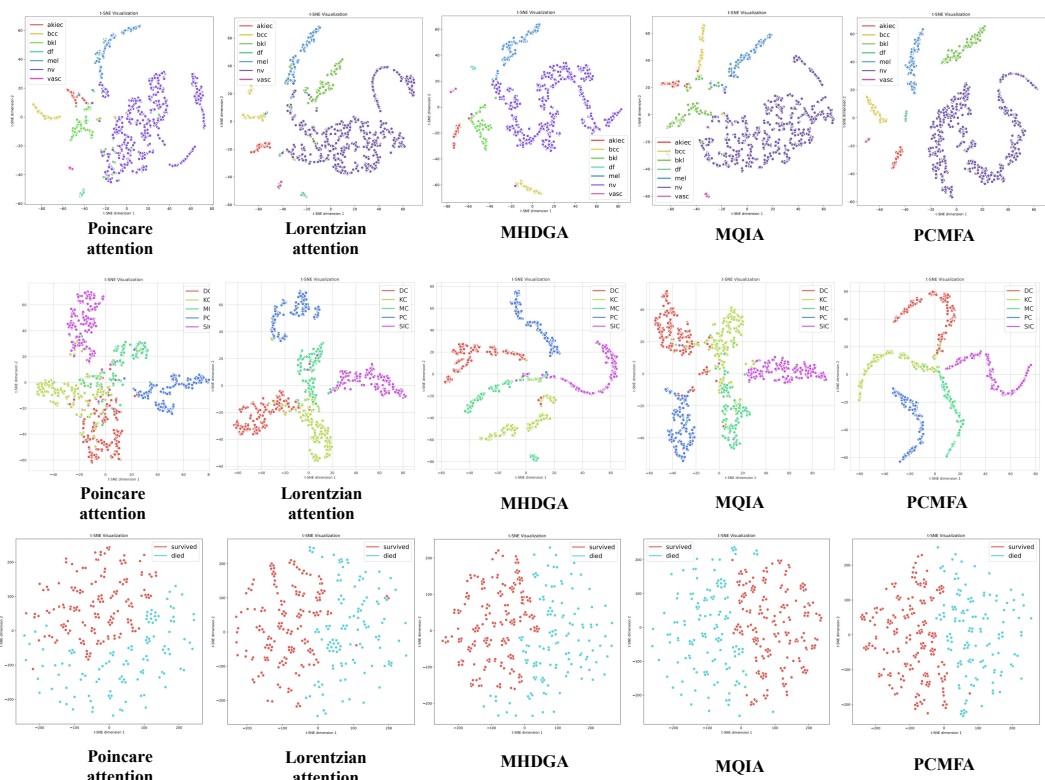

Figure 11: t-SNE visualization of feature embeddings produced by Poincaré Attention (PA), Lorentzian Attention (LA), MHDGA, MQIA, and our proposed PCMFA within `EHPAL-Net`. Rows correspond to benchmark datasets: `HAM10000` (top), `SIPaKMeD` (middle), and `MIMIC III` (bottom). Columns, from left to right, depict embeddings from PA, LA, MHDGA, MQIA, and PCMFA-enabled `EHPAL-Net`. The PCMFA-based `EHPAL-Net` embeddings form the most compact and well-separated clusters, demonstrating superior feature discriminability and modality alignment.

Table 23: Comparison of cluster compactness, class separation, and interpretability across attention mechanisms.

| Module | Compactness | Separation | Interpretability | Reasons |
|---|---|---|---|---|
| Poincaré | Poor | Low | Low | Broad, noisy embeddings |
| Lorentzian | Fair | Moderate | Fair | Better than Poincaré but still captures irrelevant cues |
| MHDGA | Good | Good | High | Dual-geometry fusion enhances structure |
| MQIA | Moderate | Moderate | High | Quantum cues help, but less separable than MHDGA |
| PCMFA | Excellent | Excellent | Excellent | Physics-inspire mutual guidance maximizes clarity |

SUMMARY OF VISUALIZATION

**Note:** Notably, MHDGA outperforms MQIA on most metrics—yielding tighter clusters, clearer class boundaries, and higher overall interpretability—while PCMFA achieves the best overall separability.

## O  APPENDIX – FUTURE SCOPE

Although `EHPAL-Net` achieves robust performance across heterogeneous modalities, future work will focus on the following enhancements:

- *Integrating domain generalization techniques*, such as adversarial domain alignment, meta-learning, and self-supervised adaptation, to improve cross-domain robustness.
- *Incorporating adversarial defenses*, including randomized smoothing, certified perturbation bounds, and noise-injected attention mechanisms, to strengthen resilience against both single-modal and cross-modal adversarial attacks.

## P   APPENDIX – BROADER IMPACTS

EHPAL-Net is a novel, effective, and efficient multimodal fusion framework that not only pushes the frontiers of heterogeneous data integration but also carries important broader impacts:

- **Advancing multimodal biomedical AI.** `EHPAL-Net` introduces a single-pass, non-iterative fusion pipeline enriched with hyperbolic dual-geometry and quantum-inspired attention, enabling seamless integration of imaging, multi-omics, and clinical data. It outperforms state-of-the-art methods for diverse tasks—including multi-disease classification, cancer prognosis, mortality prediction, and human activity recognition—while keeping computational costs low.
- **Resource-efficient design.** With its lightweight hybrid fusion layer and the novel `PCMFA` module, `EHPAL-Net` preserves hierarchical structural details to learn robust complementary representations, striking a balance between optimal performance and minimal computational cost.
- **Reliable, uncertainty-aware predictions.** We employ Monte Carlo Dropout to quantify predictive uncertainty and evaluate reliability under symmetric label noise. Even in these challenging conditions, `EHPAL-Net` consistently outperforms competitive baselines.

Although validated in a research environment, real-world deployment—especially in clinical practice—requires rigorous privacy safeguards, bias mitigation, and extensive validation (e.g., clinical trials, regulatory approval, adversarial testing) to ensure safety and trustworthiness in resource-constrained healthcare settings.

## Q   APPENDIX – ETHICS STATEMENT

This work develops and evaluates a multimodal fusion framework (`EHPAL-Net`) for integrating heterogeneous biomedical data. All experiments were conducted exclusively on publicly available, de-identified datasets (e.g., `HAM10000`, `SIPaKMeD`, `TCGA`, `MIMIC-III`), ensuring no private or personally identifiable patient information was used. Our research aims to improve the efficiency and generalizability of AI systems for healthcare applications, particularly in resource-constrained settings. We recognize that algorithmic decisions in medical contexts carry ethical implications, including potential biases due to dataset imbalance, overfitting to specific populations, or limitations in clinical generalizability. To mitigate these risks, we emphasize transparent reporting, reproducibility, and the need for further clinical validation before deployment in real-world healthcare practice.

