# OpenReview forum: "Advancing Multimodal Fusion on Heterogeneous Data with Physics-inspired Attention"
_ICLR.cc/2026/Conference — Submitted to ICLR 2026_

### Official Review · Reviewer_jkx9 · 2025-10-30

**Soundness:** 2
**Presentation:** 2
**Contribution:** 3
**Rating:** 4
**Confidence:** 3

**Summary:**

This paper proposed a physics-inspired attention learning network for the multimodal fusion learning (MFL) task. Specifically, the authors targeted at multimodal tasks in medical context, considering the challenge that different modalities do not share semantics. The key design is their efficient hybrid fusion (EHF) layer, where a physics-inspired attention module is used to capture cross-modality interactions. Different from existing early / late fusion methods, the authors proposed a sequential structure where the input modalities are integrated one by one. The overall architecture is a little complicated, but unique and novel. The authors claimed they solved three main challenges - performance, efficiency, and generalization. But the reviewer do has some questions on some aspects.

**Strengths:**

1. The proposed EHF layer, especially the Physics-inspired Cross-modal Fusion Attention (PCMFA) module, is both novel and effective. By jointly optimizing cross-modal interactions in hyperbolic and quantum spaces, the network captures rich structural relationships across different modalities. This provides an effective way in combing the advantages of hyperbolic neural networks and quantum neural networks.
2.  The performance of the proposed model is supported by the comparison with state-of-the-art single-modal and multi-modal learning methods, and the generalization of the model is demonstrated by datasets from various sources and modalities.
3. The effectiveness of the key components are supported by both theoretical analysis and ablation studies.

**Weaknesses:**

1. Concerns on performance comparison. In Table 2 we can see that existing SOTA methods already achieved high performance (with ACC over 98% and AUC over 99%) on many datasets. The concern is whether using those datasets could strongly support the author's claim that the "performance" challenge is solved. More results and datasets like "BRCA" and "ICD9" would be more persuasive.
2. Concerns on efficiency. The authors used two metrics, number of parameters (#P) and floating-point operations (#F) to show and compare the models' efficiency. However, the efficiency of the proposed method is only significant when using Shuffle-Net as backbone. Since most of the compared methods are using ResNet as their backbones, I am concerned whether this would be a fair comparison. Considering the fact that the proposed EHPAL-Net mainly achieved highest performance with a ResNet50 backbone, this issue is even more significant.

**Questions:**

1. It is a little unclear how the experiments are designed and evaluated. The focus of this paper is multimodality fusion, which works on combining data from different modalities for the final task. But the results are reported for per dataset, which may cause confusion for readers.
2. Considering the efficiency, it would be better if a fairer comparison could be provided. And it is also recommended to provide more metrics, e.g., running time and memory consumption.
3. I can see that the authors spent a lot of efforts on both model development and the experiments. From the perspective of both a reader and a reviewer, there are too many contents included here, especially as a conference paper. For example, since there are three main components and each of them are constructed by even deeper hierarchies, the readers will need to check different appendices frequently on how each module works and the related ablation studies. The contents of this work seem like a better fit for top-tier journals.

---

> ### Author Response · Authors · 2025-11-28
> **Author Response (Part 1/5)**
>
> We would like to express our sincere gratitude to reviewer **jkx9** for the careful reading of our manuscript and the thoughtful evaluation of EHPAL-Net. We greatly appreciate the reviewer’s clear summary of our main contributions—namely, the proposal of a **physics-inspired attention learning network** for multimodal fusion in medical applications, specifically addressing the challenge that different modalities often do not share common semantics; the design of the **Efficient Hybrid Fusion (EHF)** layer with its **PCMFA** module to capture rich cross-modality interactions; and the use of a **sequential fusion structure** that integrates modalities one by one, in contrast to conventional early/late fusion schemes.
>
> We are particularly grateful that reviewer **jkx9** highlights the novelty and effectiveness of the proposed EHF layer and, in particular, the PCMFA module. The acknowledgment that **jointly modeling cross-modal interactions in hyperbolic and quantum spaces** enables the network to capture rich structural relationships across modalities is very encouraging for us. We also appreciate the reviewer’s recognition that our framework provides a principled way to combine the advantages of hyperbolic neural networks and quantum neural networks within a unified multimodal fusion architecture.
>
> We also thank the reviewer for noting that the **performance** and **generalization** of our model are well supported by experiments—both through comparisons with **state-of-the-art** single-modal and multimodal baselines and through evaluations on **datasets from various sources and modalities**. The reviewer’s positive assessment of our empirical validation and ablation studies, as well as the theoretical analysis of key components, reassures us that the work makes a meaningful contribution to practical multimodal learning in realistic medical scenarios.
>
> At the same time, we view the overall scores—**Soundness: 2 (fair), Presentation: 2 (fair), Contribution: 3 (good)**—as constructive guidance indicating that while the contribution is promising and novel, there is still room to further strengthen the methodological clarity and exposition. We are grateful for this balanced evaluation: the recognition of our contributions to **performance**, **efficiency**, and **generalization**, together with critical feedback on aspects that can be improved, is extremely valuable for refining the paper and making it more accessible to a broader audience.
>
> We are deeply thankful for reviewer **jkx9’s** time, effort, and insightful comments. Their balanced and detailed evaluation has been instrumental in improving both the clarity and rigor of the manuscript. Below, we respectfully address each of the reviewer’s points in detail.
>
> ### **Note: Weakness denotes W and Questions represents Q. Edits are colored in light purple in the revised paper.**
>
>
> ### **W1 (Concerns on performance comparison):**
>
> We appreciate the reviewer’s comment and the opportunity to clarify what we mean by the “performance challenge” and how the evaluation in our paper supports that claim.
>
> ### **1. Clarifying what “performance challenge” means in our work.**
>
> In this work, the “performance challenge” does not mean that benchmarks like **HAM10000** or **PathMNIST** are intrinsically unsolved. Rather, it refers to achieving **strong predictive performance *together with* efficiency and generalization** in a single multimodal fusion framework that must handle very **different data types (e.g., images, multi omics, EHR, sensor time series) under resource constraints**.
>
> In such heterogeneous settings, many existing multimodal fusion methods (e.g., **Perceiver, MuMu, M³Att, MOTCAT, HEALNet, DRIFA-Net**) **do not consistently improve over strong unimodal baselines on our more challenging survival and EHR tasks**, partly because they do not explicitly preserve modality specific structural details while learning shared representations. **However, our “EHPAL-Net substantially improves the performance–efficiency trade off and robustness of multimodal fusion on heterogeneous medical data.”**
>
> ### **2. All datasets jointly support the performance claim; Table 2 uses 4 modality unpaired training.**
>
> Each dataset in our setting is treated as one modality stream, and these modalities are **unpaired** and have **disjoint label spaces**. EHPAL-Net’s EHF layers repeatedly fuse (a) the current modality specific features and (b) the shared representation from the previous EHF layer. The modality specific information is passed into the EHF layer, used to update the shared representation, and this shared representation is then reused for subsequent modalities in the MSIL phase. The HMML phase attaches task specific heads to each dataset.

---

> ### Author Response · Authors · 2025-11-28
> **Author Response (Part 2/5)**
>
> Concretely (Appendix I):
> * **Group 1 (4 unpaired modalities in one training run):**
> HAM10000 (dermoscopy), SIPaKMeD (cytology), TCGA BRCA (multi omics survival), and MIMIC III MORT (EHR mortality).
>
> * **Group 2 (another 4 unpaired modalities in one training run):**
> PathMNIST, OrganAMNIST (histology / CT), TCGA UCEC (multi omics survival), and MIMIC III ICD9 (EHR multi disease prediction).
>
> Thus, the numbers reported in Table 2 do **not** come from eight independent single task models.
> They come from **two EHPAL-Net instances per backbone**, each trained simultaneously on four very different, unpaired datasets in a multi task fashion.
>
> This means that every dataset—including the high accuracy imaging datasets—contributes both as its own task and as a source of structure for the shared representation that benefits the others. Across the paper, the results on all fifteen heterogeneous datasets (imaging, multi omics, and EHR/time series) form the primary evidence that the proposed physics inspired hybrid fusion improves performance while remaining efficient.
>
> Under this more challenging multi task setting, several fusion baselines underperform either the best unimodal models or recent specialized multimodal methods on some survival and EHR tasks, whereas EHPAL-Net **maintains or improves** accuracy on **all imaging tasks** while significantly boosting performance on **non-imaging tasks, such as BRCA, UCEC, MORT, ICD9, etc.** This is exactly the “performance challenge” we aim to address.
>
> ### **3. Evidence on BRCA / ICD9–type challenging tasks.**
>
> We agree that BRCA and ICD9 are particularly informative because they are not near ceiling. These tasks are already included in our main comparison (Table 2, page 9) and are trained in the same multimodal setting as the image datasets.
>
> ***TCGA BRCA (multi omics survival):***
> DRIFA-Net: 66.47 C Index; HEALNet: 67.30 C Index;
> **EHPAL-Net 18: 71.27 C Index**
> → a gain of ≈ **4 absolute points** over these leading multimodal baselines.
>
> ***MIMIC III ICD9 (multi disease prediction):***
> Best baseline accuracy ≈70–71%;
> **EHPAL-Net 18: 73.64% ACC** (and EHPAL-Net 50: 76.53% ACC);
> → ≈ **+3–5 percentage points** as well.
>
> ***MIMIC III MORT (mortality prediction):***
> HEALNet: 91.24 ACC / 94.30 AUC; DRIFA-Net: 91.32 ACC / 94.10 AUC;
> **EHPAL-Net 18: 92.34 ACC / 95.55 AUC**
> → ≈ **+1 ACC** and **+1.2–1.4 AUC** over the strongest baselines.
>
> Beyond Table 2, we already include additional “BRCA/ICD9 like” tasks in Tables 3, 5, and 6:
>
> ***Multi omics survival:** TCGA GBMLGG and TCGA KIRP, where EHPAL Net improves C Index by ≈2–4 points over recent survival focused multimodal methods such as HEALNet, DRIFA-Net, CA MLIF, and LegoFuse.*
>
> ***Paired WSI+omics survival (BLCA):** Table 3 shows that EHPAL-Net improves C Index in all four regimes (omics only, WSI only, mixed, and full paired).*
>
> ***Additional EHR/time series tasks:** MHEALTH and UCI HAR, where EHPAL-Net provides consistent, if smaller, gains over strong baselines.*
>
> **Together, these BRCA, ICD9, etc. type and related experiments directly address the reviewer’s request for more evidence on more challenging tasks.**
>
> ### **4. Why high accuracy imaging datasets are still informative.**
>
> We agree that datasets with ACC > 98% and AUC > 99% across many methods alone cannot justify that the performance challenge is “solved,” and we do not intend to claim that from those datasets alone.
>
> However, because **all these datasets are trained jointly in 4 modality unpaired groups**, they still play an important role:
>
> 1. They show that EHPAL-Net’s shared representation **does not degrade performance and often improves performance** on imaging tasks relative to leading baselines (e.g., +1–2 ACC points vs DRIFA-Net/HEALNet on HAM10000/SIPaKMeD), **while** reducing parameters and FLOPs by up to ≈85.7% and 87.8%, respectively—even though the network is simultaneously learning challenging survival and EHR tasks from entirely different cohorts.
>
> 2. They complement the more challenging BRCA/ICD9 type tasks by demonstrating that our fusion mechanism avoids negative transfer and supports **both** imaging and more challenging (multi omics and EHR) tasks within the same multi task multimodal model.
>
> **Thus, the high accuracy imaging datasets and the more challenging BRCA/ICD9 type datasets **jointly** support our performance claim: EHPAL-Net learns robust shared representations that enhance or at least maintain performance across all modalities while being substantially more efficient.**
>
> **We hope these clarifications address the reviewer’s concern and make it clear that our performance claims are driven primarily by the challenging multimodal tasks (BRCA, ICD9, survival and EHR datasets), with the near ceiling imaging datasets in Table 2 serving to show that EHPAL-Net can still achieve **state of the art performance** even on diverse imaging benchmarks when trained jointly with unpaired heterogeneous modalities such as multi omics and EHR.**

---

> ### Author Response · Authors · 2025-11-28
> **Author Response (Part 3/5)**
>
> ### **W2/Q2 (a-b) (Efficiency and ResNet backbone):**
>
> We appreciate this comment and agree that the key question is not efficiency in isolation, but the **trade off between performance and computational cost**. Our goal with EHPAL-Net is to *move the accuracy/AUC/C-Index–FLOP/parameter Pareto frontier*, not merely to minimize FLOPs.
>
> ### 1. **Performance–cost trade off with ResNet backbones (fair comparison).**
> In all comparisons we match the backbone family whenever possible, so that changes in parameters/FLOPs mainly reflect the fusion design (EHF/PCMFA/SIR) rather than the encoder. Focusing only on **ResNet based models** (**Table 2 and Tables 5–6; See pages: 9, 30-31; lines: 432-448, 1601-1615, 1620-1633**):
>
> (i) ***ResNet 18 backbone (EHPAL-Net 18 vs DRIFA-Net)***
> Both models use ResNet 18. EHPAL-Net 18 reduces parameters from **53.8M → 7.71M** (−**85.7%**) and FLOPs from **4.83G → 0.59G** (−**87.8%**) while *improving* the overall score from **70.48 → 71.74**.
>
> (ii) ***ResNet 50 backbone (EHPAL-Net 50 vs HEALNet)***
> Both models use ResNet 50. EHPAL-Net 50 reduces parameters from **27.2M → 14.4M** (−**47.1%**) and FLOPs from **3.84G → 1.83G** (−**52.3%**) while increasing the overall score from **70.25 → 72.50**.
>
> Similar patterns appear on additional datasets in Tables 5–6, where EHPAL-Net 18 consistently achieves higher ACC/AUC or C Index than DRIFA-Net at much lower computational cost.
>
> These results show that **even when we restrict ourselves to ResNet 18/50 backbones, EHPAL-Net offers a strictly better performance–computation trade off than ResNet based baselines.** ShuffleNet is therefore not the only, nor the main, source of our efficiency.
>
> ### 2. **Role of the ShuffleNet variant.**
>
> The ShuffleNet configuration (EHPAL-Net SN) is intended as an **extreme efficiency operating point** on the trade off curve—useful for mobile/edge deployment. It is not the backbone used for our best accuracy, nor is it the only evidence we provide for computational benefits.
>
> Here we clarify that:
>
> *(i) our performance–cost comparisons use **ResNet 18/50 backbones**, matching the dominant baselines;*
>
> *(ii) the ShuffleNet variant is presented as an additional low cost configuration extending the trade off analysis towards very small models.*
>
> ### 3. **Backbone agnostic trade off: adding a transformer backbone.**
>
> To further demonstrate that the improved trade off comes from the **EHF layer and PCMFA fusion**, not from a particular CNN backbone, we will add a new configuration with a **transformer backbone** (e.g., ViT Tiny), denoted **EHPAL-Net V**. In this setup:
>
> ***(i) the image encoder is ViT Tiny [1]***
>
> ***(ii) the fusion part (EHF layer with EMRC + PCMFA + SIR) is unchanged;***
>
> ***(iii) we will compare EHPAL-Net V against a baseline using the *same* transformer backbone without our EHF fusion.***
>
> **(See pages: 8-9; lines: 413 and 447)**.
>
> These experiments will isolate the contribution of the EHF layer to the performance–cost trade off in a non ResNet setting and show that the same pattern holds with transformer encoders. The detailed numbers will be added to the revised manuscript in an additional table.
>
> ### **Q2-b (More efficiency considerable metrics):**
>
> To provide a fair and more complete efficiency comparison, we have added **Table 22** (**see pages: 42-43; lines: 2259-2278**), which reports the number of parameters, FLOPs, model size, and inference time for **EHPAL-Net-18** and the strongest intermediate-fusion baseline **DRIFA-Net**, both instantiated with a **ResNet-18 backbone** and evaluated at the same input resolution (128(\times)128(\times)3) under our heterogeneous multimodal setting.
>
> As shown in Table 22, EHPAL-Net-18 uses **7.71M** parameters versus **53.8M** for DRIFA-Net, i.e., roughly **7× fewer parameters** (≈85.7% reduction). FLOPs are reduced from **4.83G** to **0.59G** (≈**8.2× fewer**, ≈87.8% reduction). This also translates into a **7× smaller model size** on disk (29.43 MB vs. 205.55 MB) and around a **9.6× speed-up in inference time** per forward pass (0.683 s vs. 6.53 s). All measurements are obtained on the same hardware and implementation framework to ensure a fair comparison.
>
>
> **Table&nbsp;22.  Complexity cost analysis of `EHPAL-Net-18` versus `DRIFA-Net` on heterogeneous medical datasets.**
>
> | **Method**   | **# Params&nbsp;(M)** | **FLOPs&nbsp;(G)** | **Model Size&nbsp;(MB)** | **Inference Time&nbsp;(s)** |
> |--------------|----------------------|--------------------|--------------------------|-----------------------------|
> | DRIFA-Net    | 53.8                 | 4.83               | 205.55                   | 6.53                        |
> | EHPAL-Net-18 | 7.71                 | 0.59               | 29.43                    | 0.683                       |

---

> ### Author Response · Authors · 2025-11-28
> **Author Response (Part 4/5)**
>
> Importantly, despite these large reductions in compute and memory footprint, EHPAL-Net-18 consistently **outperforms DRIFA-Net in predictive performance** across the fifteen heterogeneous datasets (**Tables 2, 5, and 6; see pages: 9, 30-31; lines: 432-448, 1601-1615, 1620-1633**), improving accuracy/AUC or C-index while using substantially fewer resources. This new analysis directly addresses the reviewer’s request by providing both running time and memory metrics and by comparing EHPAL-Net and DRIFA-Net under matched architectural and experimental conditions.
>
>
> Overall, we hope this clarifies that our claims are **not** driven solely by using an especially lightweight backbone. The improvements in the performance–computation trade off already hold when comparing **ResNet based EHPAL-Net** to **ResNet based baselines**, and the additional transformer backbone experiments reinforce that the EHF fusion layer is the key architectural factor.
>
> [1] Andreas Steiner, Alexander Kolesnikov, Xiaohua Zhai, Ross Wightman, Jakob Uszkoreit, and Lucas
> Beyer. How to train your vit? data, augmentation, and regularization in vision transformers. arXiv
> preprint arXiv:2106.10270, 2021.
>
>
>
> ### **Q1 (Unclear and experimental analysis).**
>
> We appreciate the reviewer’s comment and the opportunity to clarify the setting and evaluation below that make this explicit in the paper.
>
> ### **1. What we mean by “modality” and how models are trained?**
>
> Throughout the paper we adopt a *dataset-level* notion of modality: each dataset $D_k$ is treated as one modality stream in the MSIL phase, even when multiple datasets are all “images” (e.g., dermoscopy vs Pap smear vs MRI). These streams come from different data generating processes and have different label spaces, so we treat them as distinct modalities. **See pages: 2, 4, and 8; lines: 102-107, 193-198, 399-402.**
>
> Formally, in all experiments (except BLCA) we have heterogeneous, *unpaired* datasets $D_k$ with different prediction tasks (**See page: 8; lines: 399-402.**); EHPAL Net’s MSIL phase learns a shared representation $x^C$ across these streams, and the HMML phase attaches task specific heads, optimizing a joint multi task loss over all datasets.
>
> In practice we instantiate one EHPAL-Net *per modality group* (e.g., Group 1: HAM10000, SIPaKMeD, BRCA, MIMIC MORT; Group 2: PathMNIST, OrganAMNIST, UCEC, MIMIC ICD9) (**See pages: 29-30; lines: 1560-1568**), and train it end to end so that all datasets in the group share the same fusion backbone and jointly shape the shared representation. No separate model is trained per dataset; only the output heads differ across tasks.
>
> Thus, when we say “EHPAL-Net on HAM10000” (or BRCA, MORT, etc.), the reported metric comes from *one* multimodal, multi task model trained simultaneously on all modalities in the group, not from a unimodal network.
>
> ### **2. Why results are reported per dataset ?**
>
> Because each dataset defines its own prediction task and metric (e.g., accuracy/AUC for classification, C index for survival), evaluation is naturally done per dataset. When we report, for example, the BRCA C index in Tables 2/5/6 (**See pages: 9, 30-31; lines: 432-448, 1601-1615, 1620-1633**), this is the BRCA head applied to the shared representation $x^C$ learned jointly from all modalities in its group; similarly for MORT, HAM10000, SIPaKMeD, etc. The per dataset view is therefore just a task wise slice of a genuinely multimodal model.
>
> To make the multimodal nature more explicit, we already include dedicated analyses that *directly* contrast unimodal vs fused performance and demonstrate gains from combining modalities:
>
> *(a) Per modality (unimodal) vs fusion performance for four core modalities (HAM10000, SIPaKMeD, BRCA, MORT), showing consistent improvements when modalities are fused*. **See Table 13; page: 35; lines: 1857-1863**.
>
> *(b) Robustness to missing modalities when the model is jointly trained on four modalities and evaluated with subsets (imaging only vs non imaging only)*. **See Table 18; page: 38; lines: 2014-2024**.
>
> *(c) A paired multimodal BLCA experiment where omics and histopathology WSIs for the *same patients* are fused for a single task, again showing fusion gains and missing modality robustness*. **See Table 3; page: 9; lines: 452-467**.
>
> **We hope this clarifies that, while the metrics are presented per dataset for interpretability, the experiments are designed and executed in a fully multimodal, multi task fashion consistent with the paper’s focus on multimodality fusion.**

---

> ### Author Response · Authors · 2025-11-28
> **Author Response (Part 5/5)**
>
> ### **Q3 (Journal vs Conference scope).**
>
> We thank the reviewer for the positive assessment of our modeling and experimental effort and for raising the concern that the submission feels dense for a conference paper.
>
> First, our intention is to present **one unified framework**, not three independent contributions. EHPAL Net is built around a single fusion block—**the Efficient Hybrid Fusion (EHF) layer—applied sequentially. EMRC, PCMFA, and SIR** are **submodules inside this one EHF block**, with distinct roles: EMRC learns multi scale modality specific features, PCMFA performs physics inspired cross modal attention in hyperbolic and quantum spaces, and SIR provides a lightweight refinement. This structure is summarized in **Fig. 2 (MSIL + HMML overview and EHF layout)** and **Sec. 3**, so readers only need to understand this recurring pattern rather than three separate hierarchies.
>
> Second, we deliberately organized the paper so that the **main text remains conceptually self contained**, while the appendices serve readers who want implementation, theory, or exhaustive ablations. In particular:
>
> ***Secs. 1–4, together with Figs. 1–3 and Tables 2–4**, describe the problem setting, the overall pipeline (MSIL + HMML), the EHF block at a high level, and the central empirical results across modalities.*
>
> *Theoretical details of the physics inspired attention (formal definition, theorems, and proofs) are isolated in **Appendix B**, and extensive ablations/robustness analyses (e.g., different fusion schemes, component-wise ablations, label noise, missing modalities, input order sensitivity) are concentrated in **Appendices K–L**.*
>
> Finally, regarding scope: **although the work is comprehensive (new fusion architecture + physics inspired attention + 15 heterogeneous datasets + robustness studies)**, all of this serves a **single ICLR style claim**: a sequential, **efficient hybrid fusion architecture** with **physics inspired attention** can jointly **improve performance**, **generalization across unpaired heterogeneous medical modalities**, and maintaining minimal **computational cost**. The breadth of experiments and the detailed appendices are included to make this claim **verifiable and reproducible**, especially given the medical context, but are not required for a first pass understanding of the method.
>
> **We therefore believe that, while detailed, the submission is still well suited to the ICLR conference format: the core idea and results are concentrated in the main paper, and readers can consult specific appendices only as needed.**

---

### Official Review · Reviewer_uRBh · 2025-11-01

**Soundness:** 3
**Presentation:** 2
**Contribution:** 3
**Rating:** 4
**Confidence:** 4

**Summary:**

This paper proposes EHPAL-Net, a multimodal fusion framework designed to improve the efficiency and generalizability of multimodal learning, particularly for medical applications. The method introduces EHF layers that first capture modality-specific multi-scale features and then apply a physics-informed cross-modal attention to model fine-grained, structure-preserving interactions across modalities. Experiments on 15 heterogeneous medical datasets demonstrate improved performance and reduced computational cost compared to existing fusion methods.

**Strengths:**

- The experimental evaluation is extensive, covering 15 heterogeneous medical datasets, which provides strong empirical evidence of robustness.

- The proposed method is conceptually novel, combining hybrid fusion and physics-informed attention in a lightweight, scalable architecture.

**Weaknesses:**

- The claimed challenge that existing methods are “specialized to specific modalities” lacks solid justification. Many recent multimodal frameworks are modality-adaptable and can generalize by substituting modality-specific encoders.

- The scope and positioning of the paper are ambiguous: although the title and introduction suggest a general-domain multimodal framework, all experiments are performed solely on medical datasets. The three challenges identified (cross-modal interaction, modality specialization, and computational cost) are generic and not uniquely tied to healthcare AI. So is the method design. The paper would benefit from either (i) reframing the motivation specifically around clinical multimodal learning, or (ii) extending experiments to both clinical and general-domain multimodal datasets to justify broader claims.

- The methodology lacks clear conceptual grounding. The design choices are not well-connected to the stated challenges, and lack strong problem-driven justification.

- The related work section lacks discussion of more recent and relevant approaches in clinical multimodal fusion, particularly those that explicitly model the interplay between modality-specific and modality-shared representations, which could provide valuable conceptual and empirical comparisons.

**Questions:**

Minor comment:
- The purpose of the citation of paper in line 456, page 9 is unclear.

---

> ### Author Response · Authors · 2025-11-27
> **Author Response (Part 1/9)**
>
> We would like to express our sincere gratitude to reviewer **uRBh** for the careful reading of our manuscript and the thoughtful evaluation of EHPAL-Net. We greatly appreciate the reviewer’s clear summary of our main contributions—namely, the proposal of EHPAL-Net as a multimodal fusion framework aimed at improving the efficiency and generalizability of multimodal learning for medical applications; the introduction of the Efficient Hybrid Fusion (EHF) layers that first extract modality-specific multi-scale features and then apply a physics-informed cross-modal attention mechanism to model fine-grained, structure-preserving interactions across modalities; and the extensive experimental validation on 15 heterogeneous medical datasets demonstrating both improved performance and reduced computational cost compared to existing fusion methods.
>
> We are particularly grateful that the reviewer highlights the breadth and rigor of our experimental evaluation—spanning 15 diverse medical datasets—as **strong empirical evidence of robustness**, and **recognizes the conceptual novelty** of our approach, especially the combination of **hybrid fusion** and **physics-informed attention** in a lightweight, scalable architecture. This positive assessment of both the methodological innovation and the practical efficiency of EHPAL-Net is highly encouraging for us, especially given our focus on realistic medical settings where computational resources are often constrained.
>
> The overall scores suggest that the work is on a solid trajectory while still leaving room for further refinement, particularly in terms of clarity and presentation. We view this as constructive guidance. The reviewer’s acknowledgment of the **robustness of our empirical results, the novelty of our physics-informed attention design**, and the practical value of our efficiency improvements is very motivating. At the same time, the critical feedback on presentation and exposition is invaluable for helping us sharpen the methodological description, clarify design choices, and make the paper more accessible to a broader audience.
>
> **We are deeply thankful for the reviewer’s time, effort, and insightful comments. Their balanced and detailed evaluation has been instrumental in improving both the clarity and rigor of the manuscript. Below, we respectfully address each of the reviewer’s points in detail.**
>
> **Note: Weakness denotes W; Edits are colored in light purple in the revised paper.**
>
> ### **W1 (a) (Specialized to specific modalities).**
>
> Thank you for pointing this out. We agree that the current wording in the abstract (“specialized to specific modalities”) is too compressed and can be misunderstood. What we intended to express is more specific and restricted to the *medical* MFL setting we study.
>
> In our paper, we consider **dataset level modalities**, where each dataset (e.g., HAM10000 dermoscopy, SIPaKMeD cytology, multiple TCGA multi omics cohorts, MIMIC III EHR) is treated as its own modality stream, and these streams are typically **unpaired and heterogeneous**, with different cohorts and label spaces. Most existing medical multimodal fusion methods we compare against are **designed and evaluated for a small, fixed set of modality configurations**, such as: pairs of MRI sequences, image+text, or WSI+omics, usually with **paired inputs within one cohort**. They are not trained or demonstrated on the kind of setting we target, where 15 heterogeneous, unpaired modalities are integrated in a single MSIL+HMML pipeline.
>
> This is the sense in which we wrote that prior methods are “specialized to specific modalities”: they are instantiated and validated on particular modality combinations (e.g., MRI+clinical, WSI+omics) and do not show scalability to many unpaired, structurally diverse medical datasets within one model. We have made the necessary revisions to clarify this in the revised manuscript by replacing the phrase “specialized to specific modalities” with lines that explicitly refer to **Second, they are often designed and evaluated for narrow, fixed modality configurations (e.g., imaging only, or specific pairs such as image and omics or image and clinical text), which limits evidence of their adaptability and generalizability to broader collections of heterogeneous medical modalities** ***(see page: 1; lines: 16-19)***, and by pointing the reader to our dataset-level definition of modality in **Sections 1 and 3 (see pages: 2 and 4; lines: 102-107 and 193-198)**.
>
> ### **W1 (b) (modality-adaptable).**
>
> We agree that many recent multimodal fusion frameworks are modular and can, in principle, be adapted to new modalities by substituting modality specific encoders. Our work focuses on a complementary aspect: the **performance–efficiency trade off in low resource medical AI when training on heterogeneous medical data**.

---

> ### Author Response · Authors · 2025-11-27
> **Author Response (Part 2/9)**
>
> In our study, all multimodal baselines and EHPAL Net are instantiated with comparable backbone networks (e.g., ResNet 18/50, Inception v3, ShuffleNet), so ***differences in computational cost*** arise primarily from the **fusion architecture rather than from the backbone choice**.
>
> The multimodal fusion baselines remain computationally heavy because their **fusion designs** rely on high cost attention (large cross /co attention matrices, iterative attention loops, or latent arrays) and **separate modality specific encoders for each stream**, causing parameters and FLOPs to scale quickly with the number of modalities.
>
> This is reflected in **Tables 2, 4, 8, and 20 and Fig. 8**, where models such as DRIFA Net, HEALNet, MOTCAT, M³Att, Perceiver, LegoFuse, and CA MLIF incur roughly **3.9–183M parameters** and **1.48–12.14 GFLOPs**, even when handling only two or three modalities.
>
> In contrast, EHPAL Net is explicitly designed as a **low cost hybrid fusion architecture**. The EMRC and SIR modules capture multi scale spatial structure with lightweight heterogeneous convolutions, while the PCMFA module performs physics inspired fusion in dual geometry and dual domain within an Efficient Hybrid Fusion (EHF) layer whose per **layer cost is (O(n)), yielding an overall fusion cost of (O(mn))** in the number of modalities and avoiding all pairs attention (**Proposition 2**). Across 15 heterogeneous medical datasets, this design yields up to **85.7% fewer parameters** and **87.8% fewer FLOPs** than strong multimodal baselines, while improving performance by up to **3.97% (Tables 2, 5–6, 20; Fig. 8)**.
>
>
>
> ### **W2 (Ambiguity in Scope & Positioning):**
>
> We clarify that the scope of this work is *heterogeneous medical data* and AI driven healthcare applications, not general domain multimodal learning.
>
> **Title.** If permissible, we will revise the title to **“ADVANCING MULTIMODAL FUSION ON HETEROGENEOUS MEDICAL DATA WITH PHYSICS INSPIRED ATTENTION”** to make the medical focus explicit.
>
> **Motivation and scope in the paper.** Throughout the paper we already state that our framework is designed specifically for heterogeneous **medical** modalities. For example, in the **Introduction (page: 1-2; lines 37–80)** we write: *“However, broader adoption of hybrid multimodal methods in AI driven healthcare faces three main challenges – performance, generalization, and efficiency.”* However, to address the reviewer’s concern, we include the term medical in the description of Challenge 3 in the revised manuscript **(See page: 2; lines: 76 and 78).***
>
> We also emphasize that healthcare data involve heterogeneous modalities such as dermoscopy, cytology, histology, multi omics, and EHR time series (**Sec. 1, Sec. 2, and Sec. 4**). These modality-aware datasets used in our experiments are medical datasets only.
>
> **Relation to prior multimodal fusion work.** Our approach is **medical domain centric** because the main multimodal fusion baselines we build upon—HEALNet, LegoFuse, DRIFA Net, MOTCAT, etc.—are themselves designed and evaluated primarily on heterogeneous *medical* data (e.g., images, multi omics, EHR), as discussed in the **Related Work section and summarized in Table 1 (see page: 3; lines: 120-128)**. These methods suffer from three key challenges in medical settings: **performance, generalization, and efficiency (pages 1–2, lines 37–80)**, which directly motivate our design of EHPAL Net.
> **What we address in this paper.** In this study, we **explicitly focus on addressing these three challenges only in the medical domain**, instead of claiming a generic multimodal solution. Our contributions and experiments are restricted to heterogeneous medical data.
>
> **Future work.** Extending EHPAL Net to natural image or other non medical vision benchmarks is interesting future work, but it is *outside* the scope of the present submission. We have made the necessary revisions accordingly to clarify these scenarios (**see page 10 and lines 538–539**).

---

> ### Author Response · Authors · 2025-11-27
> **Author Response (Part 3/9)**
>
> ### **W3 (Conceptual grounding and problem-driven justification).**
>
> We agree that explicit challenge wise grounding is essential. To address this, **Appendix B.6 (“Additional Guarantees for Hybrid Fusion”) see pages: 21-23; lines: 1091-1190** organizes our theory around the exact three challenges stated in **Sec.1**, and ties each design choice (EMRC→PCMFA→SIR in EHF, followed by LF) to formal guarantees with proofs earlier in **Appendix B**.
>
> ### **Challenge 1 — Effective, structure preserving shared representations.**
>
> We formalize the target as **complementary shared representations** (Definition 4), requiring (i) per modality hierarchical cues to be preserved by the dual geometry branch in PCMFA, (ii) cross modal dependencies captured by the parallel hyperbolic–quantum streams, and (iii) dependence only on present modalities via mask consistent late fusion.
>
> These properties follow from: (a) **hierarchy preservation** of MHDGA and PCMFA (Theorem 2 and Corollary 1), (b) **non expansive, norm stable** PCMFA gating that prevents channel blow up (Theorem 3), and (c) **mask aware late fusion** that exactly ignores missing inputs and renormalizes over present ones (Lemma 3). Combined, these imply **dominance mitigation**—no single stream can swamp the fusion (Proposition 3)—and **deep chain stability** when PCMFA is applied twice per EHF (Corollary 2), while preserving the hierarchical physics inspired ordering through LF and SIR (Corollary 3).
>
> ### **Challenge 2 — Performance vs. computational cost trade off.**
>
> We prove that each EHF layer is **linear in the per sample spatial size** (O(n)) and that sequential fusion over (m) modalities yields **overall (O(mn))** rather than (O(m^2n)) (Proposition 2 and Proposition 4), with a **component wise breakdown** of EMRC, PIL, LIL, MQIA, LF, and SIR (Table 21). Moreover, **adding a new modality can be made (\varepsilon)-close (“do no harm”)** to the prior model at **only +(O(n))** extra cost (Theorem 4), and tuning the new stream’s gates traces a **continuous, constructive performance–cost path** (Corollary 4), which your experiments corroborate (Tables 4/10/20; Fig. 8).
>
> ### **Challenge 3 — Adaptability and generalization across heterogeneous modalities.**
>
> The **mask consistent LF** guarantees robustness to **any subset of present modalities at inference** (Lemma 4). We also prove **monotone expressivity in the number of modalities**: the function class with (m) modalities is contained in the closure of the class with (m{+}1) modalities (Proposition 5), ensuring that availability of extra modalities cannot force worse risk; we further give **permutation stability conditions** (Proposition 6). These statements formally justify our empirical robustness to missing/unpaired inputs and input order changes (Table 19; §J/L).
>
> Together, **B.6** provides **problem driven, challenge wise** theoretical grounding, while **B.1–B.5** contain the underlying definitions and proofs (physics inspired PCMFA, dual geometry interaction, hierarchy preservation, norm stability, and complexity).
>
> **For example:**
>
> ### **Appendix B: Theoretical Foundations of Physics-Inspired Attention.**
>
> We introduce a formal definition of a physics‑inspired attention mechanism that requires (i) a geometric branch realized as a Riemannian exponential‑map embedding on a constant‑negative‑curvature manifold, (ii) a quantum‑inspired branch defined on a complex Hilbert space with Born‑rule–based probability scores, and (iii) a probability‑preserving, non‑expansive fusion of these scores. We then prove that the proposed PCMFA module satisfies this definition (**Theorem 1**), analyze the dual‑geometry interaction between Poincaré and Lorentz attentions (**Theorem 2**), show that the resulting gate preserves hyperbolic hierarchical structure (**Corollary 1**), and establish that it acts as a contraction‑like operator on feature norms (**Theorem 3**). Together, these results demonstrate that our “physics‑inspired” design is not merely metaphorical but enforces concrete geometric and probabilistic constraints inherited from physical modeling.
>
> In this section we make precise the notions of **physics-inspired attention**,
> **dual-geometry interaction**, and **hierarchical structure preservation**
> in the context of the EHF pipeline. Specifically, we focus on the PCMFA module and its MHDGA/MQIA components (Sec. **3.1.2**), and show how they yield stable and efficient shared representations within EHF.

---

> ### Author Response · Authors · 2025-11-27
> **Author Response (Part 4/9)**
>
> ### **Appendix B.1  Physics-Inspired Nature of PCMFA**
>
>
> To formalize the intuition behind PCMFA, we frame the module as the fusion of two norm-stable branches—a geometric path operating in hyperbolic space and a quantum-inspired path operating in a complex Hilbert space. We first state a precise definition of a physics-inspired attention mechanism and then prove that PCMFA satisfies this definition.
>
> ### **Definition 1 (Physics-inspired attention mechanism)**
>
> Let $\mathcal{X}$ be an input feature space, $\mathcal{M}$ a Riemannian
> manifold of constant negative curvature (e.g., a Poincar\'e ball or Lorentz
> hyperboloid), and $\mathcal{H}$ a finite-dimensional complex Hilbert space
> with inner product $\langle\cdot,\cdot\rangle$. An attention mechanism $T : \mathcal{X} \to [0,1]^e$ is called
> \emph{physics-inspired} if there exist maps
> $E^{\mathrm{geo}} : \mathcal{X} \to \mathcal{M}$ and
> $\Psi : \mathcal{X} \to \mathcal{H}$ such that:
>
> **(G1) Geometric branch.**
>   $E^{\mathrm{geo}}$ is realized via a Riemannian exponential-type map on
>   $\mathcal{M}$ (e.g., Poincar\'e ball or Lorentz hyperboloid), and the
>   corresponding attention scores depend only on geodesic distances or norms
>   in $\mathcal{M}$.
>
> **(Q1) Quantum-inspired branch.**
>   $\Psi(x)$ is a complex state whose channel-wise scores are proportional to
>   Born-rule amplitudes $|\Psi(x)_k|^2$, i.e., squared magnitudes in
>   $\mathcal{H}$, normalized to a probability vector, following the standard
>   probabilistic interpretation of quantum states.
>
> **(N1) Physical normalization and stability.**
>   The final attention weights $T(x)$ are obtained from these physically
>   derived scores by a probability-preserving normalization (e.g., Softmax or sigmoid) and induce a non-expansive multiplicative update on
>   features in every $\ell_p$ norm.
>
>
> ### **Theorem 1 (PCMFA is a physics-inspired attention mechanism)**
>
> For each EHF layer $i$, the PCMFA map
> $$
> A_i = \mathrm{PCMFA}\bigl(x_i^{S}, x'_{i+1}\bigr)
> $$
> defined in **Eq.(1)** is a physics-inspired attention mechanism in the sense of
> **Definition (1)**.
>
>
> ### ***Proof***
>
> We verify (G1)--(Q1)--(N1) for the PCMFA construction in **Sec. 3.1.2**.}
>
> **(G1) Geometric branch.**
> Within PCMFA, the MHDGA block takes
> $\psi_i = \mathrm{GAP}(\mathrm{DCT}(x'_i))$ (**Eq. 3**) and maps it to
> hyperbolic embeddings in two constant-negative-curvature models:
> the Poincar\'e ball and the Lorentz hyperboloid (**Eqs.4--8**).
> Both mappings are exponential-type projections with learnable but bounded
> curvature (see the curvature parameterization and clipping in **Sec. 3.1.2** and
> **Lemma 2** below), so
> $E^{\mathrm{geo}}(\psi_i)$ is a well-defined element of
> $\mathcal{M}^P \times \mathcal{M}^L$.
> The corresponding attentions $A^P_i, A^L_i$ depend only on hyperbolic radii
> and Lorentzian norms, and are fused into $A^D_i$ by an affine map plus
> sigmoid (**Eq. 3**), satisfying (G1).
>
>
> **(Q1) Quantum-inspired branch.**
> MQIA maps the same $\psi_i$ into a complex vector
> $q_i \in \mathbb{C}^e$ via learned real and imaginary coefficients (**Eq. 9**),
> so $q_i$ lies in a Hilbert space $\mathcal{H} \cong \mathbb{C}^e$.
> The channel-wise quantities $|q_{i,k}|^2$ are Born-rule amplitudes, and
> Softmax over a scaled version of $|q_i|^2$ together with a Lorentz-norm term, yields a probability vector $A^Q_i$ (**Eq. 9**), satisfying (Q1).
>
>
> **(N1) Normalization and stability.**
> MAFG fuses $A^D_i$ and $A^Q_i$ channel-wise using bounded coefficients
> $c_i$ (**Eq.10**).
> **Lemma 2** below implies that all components of
> $A^D_i$ lie in $(0,1)$ and $A^Q_i$ is a probability vector; under the bounded
> gate assumption, the fused gate $A_i$ satisfies $\|A_i\|_\infty \le 1$.
> **Theorem 3** then shows that the multiplicative update
> $z \mapsto z \odot A_i$ is non-expansive in every $\ell_p$ norm.
>
>
> Together, these properties establish that PCMFA has (i) a geometric branch on
> hyperbolic manifolds, (ii) a quantum-inspired branch in a complex Hilbert
> space, and (iii) a normalized, non-expansive fusion of the two, so it is
> physics-inspired in the sense of **Definition 1**.

---

> ### Author Response · Authors · 2025-11-27
> **Author Response (Part 5/9)**
>
> ### **Appendix B.2  Dual-Geometry Interaction in MHDGA**
>
> We next formalize the dual-geometry interaction implemented by MHDGA
> and show that it acts as a monotone consensus operator over Poincar\'e and
> Lorentz attentions.
>
> ### **Definition 2 (Dual-geometry interaction)**
>
> For modality $i$, let $\psi_i = \mathrm{GAP}(\mathrm{DCT}(x'_i))$ be the
> frequency summary (**Eq.3**). Denote by
>
> $$
> \phi^P_i(\psi_i) \in \mathcal{M}^P,\qquad
> \phi^L_i(\psi_i) \in \mathcal{M}^L
> $$
> the Poincar\'e and Lorentz embeddings produced by the MHDGA sub-branches
> (**Eqs.(4)--(8)**). Let $A^P_i, A^L_i \in \mathbb{R}^e$ be the corresponding
> geometry-specific attention vectors computed from curvature-aware norms and
> Minkowski inner products, and define the fused dual-geometry attention
> (**Eq.3**) as
> $$
> A^D_i = \sigma\bigl(LA^L_i + PA^P_i\bigr),
> $$
> where $L$ and $P$ are learnable linear maps and $\sigma$ is the element-wise
> sigmoid. The map $\mathrm{MHDGA}_i : \psi_i \mapsto A^D_i$ is called the
> dual-geometry interaction operator for modality $i$.
>
>
> A useful property of the hyperbolic projections used in MHDGA is that they
> are radially monotone.
>
> ### **Lemma 1. (Radial monotonicity of hyperbolic projections)**
>
> Let $\psi,\tilde{\psi}\in\mathbb{R}^e$ with
> $\|\psi\|_2 < \|\tilde{\psi}\|_2$. Assume that the Poincar\'e and Lorentz
> embeddings in MHDGA are implemented via exponential maps at the origin as in
> **Eqs.(4) and (7)**. Then:
>
>   1. the hyperbolic radius of $\phi^P(\psi)$ in the Poincar\'e model is
>         strictly smaller than that of $\phi^P(\tilde{\psi})$;
>
>   2. the Lorentzian rapidity of $\phi^L(\psi)$ is strictly smaller than
>         that of $\phi^L(\tilde{\psi})$.
>
>
> ### **Proof.**
>
> In both models, the exponential map at the origin has the form
> $\psi \mapsto f(\|\psi\|_2)\,\psi/\|\psi\|_2$, where $f$ is a smooth,
> strictly increasing function of the Euclidean radius $\|\psi\|_2$ (
> **Eqs.(4) and (7)**. Thus a larger Euclidean norm yields a larger hyperbolic
> radius or rapidity, giving the claimed ordering.
>
> ### **Proposition 1 (Monotone consensus dual-geometry attention)**
>
> Fix modality $i$ and channel $k$. Let
> $$
> A^D_{i,k} = \sigma\bigl(\ell_k(A^L_{i,k}, A^P_{i,k})\bigr)
> $$
> denote the $k$-th component of $A^D_i$, where $\ell_k$ is the $k$-th
> component of the affine map $L A^L_i + P A^P_i$. Suppose the coefficients of
> $\ell_k$ with respect to $A^L_{i,k}$ and $A^P_{i,k}$ are non-negative.
> Then:
>
>   1. $A^D_{i,k}$ is weakly increasing in each of its arguments
>         $A^L_{i,k}$ and $A^P_{i,k}$;
>
>    2. if both $A^L_{i,k}$ and $A^P_{i,k}$ increase, then $A^D_{i,k}$
>         increases strictly;
>   3. when one of $A^L_{i,k},A^P_{i,k}$ increases and the other decreases,
>         $A^D_{i,k}$ remains between the scalar responses obtained by relying
>         on either geometry alone, i.e., it behaves as a consensus between the
>         two branches.
>
>
> ### ***Proof***.
>
> The derivative of $A^D_{i,k}$ with respect to $A^L_{i,k}$ is
> $$
> \frac{\partial A^D_{i,k}}{\partial A^L_{i,k}}
>    = \sigma'\bigl(\ell_k\bigr)
>          \frac{\partial \ell_k}{\partial A^L_{i,k}}.
> $$
>
>
> The sigmoid derivative $\sigma'$ is strictly positive, and by assumption the
> partial derivative of $\ell_k$ with respect to $A^L_{i,k}$ is non-negative,
> so $\partial A^D_{i,k}/\partial A^L_{i,k}\ge 0$.
> The same argument holds for $A^P_{i,k}$, proving item 1.
> If both $A^L_{i,k}$ and $A^P_{i,k}$ increase and at least one coefficient is
> strictly positive, then $\ell_k$ increases and strict monotonicity of
> $\sigma$ implies item 2.
> For item 3, note that $\ell_k$ is affine with non-negative coefficients, so
> for fixed values of one argument it varies monotonically with the other.
> Applying a sigmoid, which maps $\mathbb{R}\to(0,1)$ monotonically, ensures
> that $A^D_{i,k}$ lies between the values induced by each branch, yielding the
> consensus behavior.
>
>
> Lemma **1** shows that both geometries encode the same radial
> ordering of channels, while Proposition 1 shows that their
> fusion in $A^D_i$ respects and stabilizes this ordering.
> This is the sense in which MHDGA realizes a \emph{dual-geometry interaction}:
> two hyperbolic representations cooperate through a monotone consensus gate to
> produce robust, geometry-aware attention.

---

> ### Author Response · Authors · 2025-11-27
> **Author Response (Part 6/9)**
>
> ### **Appendix B.3  Hierarchical Structure Preservation.**
>
> We now connect dual-geometry interaction to the notion of
> \emph{hierarchical structure preservation}.
> Intuitively, channels (or semantic units) that are deeper in an underlying
> hierarchy should receive systematically stronger or weaker attention than
> their ancestors; hyperbolic geometry is well known to represent such
> tree-like structures efficiently.
>
> ### **Definition 3 (Hierarchical structure and its preservation)**.
>
> Let $(\mathcal{C},\preceq)$ be a finite partially ordered set of semantic
> units (e.g., classes or channels) with a distinguished root $r\in\mathcal{C}$.
> A map $f:\mathcal{C}\to\mathcal{Z}$ into a metric space
> $(\mathcal{Z},d_{\mathcal{Z}})$ is called \emph{radially hierarchical} if
> there exists a scalar function $\rho:\mathcal{Z}\to\mathbb{R}_{\ge 0}$ such
> that:
>
>   1. $\rho(z)$ depends only on $d_{\mathcal{Z}}(z,z_0)$ for some fixed
>         root point $z_0\in\mathcal{Z}$, and
>
> 2. whenever $u\preceq v$ in $\mathcal{C}$, one has
>         $\rho\bigl(f(u)\bigr) < \rho\bigl(f(v)\bigr)$.
>
> An attention mechanism that assigns scores $\alpha_c$ to units $c\in\mathcal{C}$
> is said to be \emph{hierarchical-structure preserving} if each $\alpha_c$ is a
> monotone function of a radially hierarchical representation of $c$.
>
>
> ### **Assumption 1 (Hyperbolic hierarchical encoding)**
>
> For each modality $i$, there exists a latent hierarchy $(\mathcal{C},\preceq)$
> over channels and a radially hierarchical embedding
> $f_i^\star:\mathcal{C}\to\mathcal{M}^P\times\mathcal{M}^L$ such that the
> learned hyperbolic mappings $\phi^P_i,\phi^L_i$ used in MHDGA approximate
> $f_i^\star$ in the sense that deeper nodes in the hierarchy are mapped to
> larger hyperbolic radii (up to small perturbations).
>
>
> This assumption is standard in hyperbolic representation learning and is
> supported by empirical evidence that Poincar\'e and related embeddings recover
> hierarchical structures in low-dimensional hyperbolic manifolds.
>
> ### **Theorem 2 (Hierarchical structure preservation of dual-geometry attention)**
>
> Under Assumption **1**, suppose that the
> geometry-specific attentions $A^P_i$ and $A^L_i$ are computed as channel-wise
> monotone functions of the corresponding hyperbolic radii or norms (as in the
> PIL/LIL definitions). Then, for each modality $i$, the dual-geometry attention
> $A^D_i$ produced by MHDGA is hierarchical-structure preserving in the sense of
> Definition **3**: if $u\preceq v$ in the latent hierarchy, then
> $A^D_{i,u} \le A^D_{i,v}$ (or the reverse inequality, depending on the chosen
> monotone direction).
>
> ### ***Proof***
>
> By Assumption **1** and
> Lemma **1**, deeper nodes in the hierarchy have strictly larger
> hyperbolic radii in both the Poincar\'e and Lorentz embeddings.
> By construction, $A^P_i$ and $A^L_i$ are obtained by applying monotone
> functions (linear maps and element-wise nonlinearities) to these radii or
> norms, so they are monotone in depth:
> $u\preceq v \Rightarrow A^P_{i,u}\le A^P_{i,v}$ and likewise for $A^L_i$.
> Proposition **1** shows that $A^D_i$ is monotone in each of
> $A^P_i$ and $A^L_i$, hence $A^D_i$ inherits the same ordering and is
> hierarchical-structure preserving.
>
> Finally, we propagate this property through the full PCMFA gate.
>
> ### **Corollary 1 (Hierarchical physics-inspired attention in PCMFA)**
>
> Under Assumptions **2** and
>  **1**, the full PCMFA gate $A_i$ obtained by
> fusing $A^D_i$ and $A^Q_i$ via MAFG (Eq.(**10**)):
>
>   1. preserves the hierarchical order of channels induced by the
>         hyperbolic embeddings, and
>   2. is non-expansive as a multiplicative operator on feature tensors,
>         as stated in Theorem **3**.
>
> ### ***Proof***
>
> By Theorem **2**, $A^D_i$ is
> hierarchical-structure preserving.
> The quantum-inspired attention $A^Q_i$ is a probability vector derived from
> the same summary $\psi_i$ and acts as an energy-based reweighting of
> channels (Eq.(**9**).
> MAFG fuses $A^D_i$ and $A^Q_i$ with bounded coefficients $c_i$ in a
> channel-wise monotone manner.
> Thus the hierarchical ordering from $A^D_i$ is preserved in $A_i$.
> Boundedness of $A_i$ and non-expansiveness of the resulting multiplicative
> update follow from Lemma **2**,
> Assumption **2**, and
> Theorem **3**.
>
>
> Corollary **1** formalizes that PCMFA is not only
> physics-inspired and norm-stable, but also respects the hierarchical structure
> encoded by the hyperbolic embeddings in MHDGA.

---

> ### Author Response · Authors · 2025-11-27
> **Author Response (Part 7/9)**
>
> ### **Appendix B.6 Additional Guarantees for Hybrid Fusion**
>
>
> In this section, we provide a theoretical grounding for our three main challenges: (1) learning rich, structure-preserving shared representations; (2) achieving an optimal trade-off between performance and computational cost; and (3) ensuring adaptability and generalizability across heterogeneous medical data modalities.
>
> ### **Challenge~1 — Richer Shared Representation Learning.**
>
>
> ### **Definition 4 (Complementary Shared Representation)**
>
> The shared representation learned by the EHF pipeline (Fig.2 C ) is complementary shared
> when it (i) preserves per-modality hierarchical cues via the dual-geometry branch inside PCMFA,
> (ii) captures cross-modal dependencies unavailable to any single modality through the parallel
> hyperbolic--quantum streams, and (iii) depends only on present modalities through the mask-aware
> LF Appendix E, Steps~2--3; Eqs 14 --15. Items (i)--(ii) follow from Appendix B.1-- B.3; item (iii)
> follows from the LF construction.
>
>
> ### **Lemma 3 (Mask-aware LF $\Rightarrow$ missing-modality consistency)**
>
> For any EHF layer, the LF in Appendix E  assigns zero weight to any missing modality and renormalizes
> over the present subset; the fused state equals the fusion of the available inputs only (Appendix E,
> Eqs.(14)--(15).
>
> ### ***Proof***
>
> Appendix E sets the logits of missing streams to a large negative constant before the softmax (Step 2),
> so their weights become exactly zero; the gated concatenation (Step 3) then uses only present streams.
>
>
> ### **Proposition 3 (Dominance mitigation inside EHF)**
>
> No single present modality can swamp the fusion within an EHF layer.
>
>
> ### ***Proof***
>
> By Lemma 2 (Appendix B.4) the geometry and quantum branches yield bounded scores; by Assumption 2
> and Theorem 3 (Appendix B.4 ) the multiplicative PCMFA update is non‑expansive channel‑wise; and by
> Lemma 3 LF redistributes probability mass only across present streams via Softmax
> (Eq. 14). Together, bounded gates $+$ non‑expansive updates $+$ normalized LF weights prevent
> any one stream from dominating the fused state. Table 15 empirically corroborates this mechanism.
>
>
> ### **Corollary 2 (Two-pass PCMFA yields deep-chain stability)**
>
> Each PCMFA update is non-expansive (Appendix B.4, Thm. 3). Algorithm 2 applies PCMFA twice
> per EHF layer; composing non-expansive maps yields a stable update, and stacking EHF layers in
> MSIL preserves this bound (Fig. 2 C; Algorithms 1--2).
>
>
> ### ***Proof***
>
> Theorem 3 (Appendix B.4) shows each PCMFA map is non‑expansive. The composition of non‑expansive
> The operators are non‑expansive; Algorithm 2 applies PCMFA twice within a single EHF layer, hence the claim.
>
>
> ### **Corollary 3 (Hierarchical, physics‑inspired attention persists through fusion)**
>
> The hierarchy‑preservation property established for PCMFA (Appendix B.2–B.3: Theorem 2, Corollary 1)
> is preserved by the mask‑consistent LF and SIR refinement in EHF.
>
>
> ### ***Proof***
>
> Corollary 1 states that PCMFA’s gate is hierarchical and non‑expansive; LF (Eqs.13–15) reweights
> channels with probability‑normalized gates without changing their monotone order, and SIR is a
> residual refinement (Eq.16). Thus the ordering induced by the dual‑geometry attention persists.

---

> ### Author Response · Authors · 2025-11-27
> **Author Response (Part 8/9)**
>
> ### **Challenge~2 — Performance--Cost Trade-off.**
>
>
> ### **Proposition 4 (Per‑layer linear cost; overall $O(mn)$ sequential fusion)**
>
> Let $n$ be per‑sample spatial size and $m$ the number of modalities processed in MSIL. One EHF
> layer has $O(n)$ work; stacking $m{-}1$ such layers yields $O(mn)$ fusion cost.
>
>
> ### ***Proof***
>
> Proposition 2 (Appendix B.4) and Table 21 show each of EMRC, PIL, LIL, MQIA, LF, and SIR is $O(n)$ per
> layer; Algorithm 2 executes them once per EHF (with two PCMFA calls but still linear without token–token
> all‑pairs), hence $O(n)$ per layer; Algorithm 1 composes $m{-}1$ layers sequentially, hence $O(mn)$.
>
>
> ### **Theorem 4 ($\varepsilon$–do‑no‑harm under residual + gated interaction)**
>
> Consider expanding from $m$ to $m{+}1$ modalities. For any $\varepsilon>0$, there exists a setting of
> the new stream’s parameters (MAFG gate coefficients and LF MLP) that makes the predictions of the
> $(m{+}1)$‑modality model differ from the $m$‑modality model by at most $\varepsilon$, while the
> per‑sample fusion cost increases by an additive $O(n)$ (Proposition 4).
>
>
> ### ***Proof***
>
> Keep all pre‑existing parameters. In the new EHF: (i) set the per‑stream MAFG coefficients for the
> new modality arbitrarily close to zero so its PCMFA gate contributes an arbitrarily small factor (
> Eq.10); (ii) set the LF MLP (Eq.13) to produce a large negative pre‑Softmax score for the new
> modality so its $\alpha$‑weight is arbitrarily small among present modalities (Eq.14); (iii) because
> PCMFA is non‑expansive (Theorem 3), multiplying by gates with sup‑norm $<\delta$ perturbs the
> shared state by at most $O(\delta)$; concatenation in LF (Eq. 15) preserves this bound. Choosing
> $\delta$ small enough ensures the output shift is $\le\varepsilon$. Cost increases by one additional EHF,
> which is $O(n)$ by Proposition 4.
>
>
> ### **Corollary 4 (Constructive performance–cost path)**
>
> Varying the new stream’s gates from the neutral setting in Theorem 4 to the
> learned setting traces a continuous path in prediction space while cost increases by a fixed $O(n)$.
> Empirically, this path improves the frontier relative to baselines at comparable FLOPs/parameters
> (Tables 4,10, 20; Fig.8).
>
>
> ### ***Proof***
>
> Gate vectors are continuous in their parameters (Eqs. 9 – 10, 13 –15); by Theorem 3, the mapping
> from gates to outputs is non‑expansive, so the output varies continuously with gates. Cost is fixed by
> the presence of the extra EHF (Proposition 4). Tables 4, 10, 20 and Fig. 8 report the
> empirical frontier at these cost points.
>
>
> ### **Challenge 3 — Generalization & Adaptability.**
>
> ### **Lemma 4 (Missing‑modality robustness (restated))**
>
> Under the LF design in Appendix E, removing any subset of modalities at inference yields a well‑defined
> fused state equal to the fusion over the remaining subset.
>
>
> ### ***Proof***
>
> Identical to Lemma 3; see Eqs.(13–~15).
>
>
>
> ### **Proposition 5 (Monotone expressivity w.r.t.\ added modalities)**.
>
> Let $\mathcal{F}_m$ be the functions realized with $m$ modalities. Then $\mathcal{F}_m$ is contained
> in the closure of $\mathcal{F}_m+1$: adding a stream cannot force worse risk because the
> neutral setting in Theorem 4 arbitrarily approximates the $m$‑modality map,
> while non‑neutral gates can realize strictly different mappings (Tables 2, 14).
>
>
>
>
> ### ***Proof***
>
> Theorem 4 gives an $\varepsilon$‑approximation to the $m$‑modality map
> inside the $(m{+}1)$‑modality model; hence
>  $$
>   \mathcal{F}_{m} \subseteq \overline{\mathcal{F}_m+1}
> $$
>
>
> Tables 2 and 14 show cases where using the extra stream strictly improves performance.

---

> ### Author Response · Authors · 2025-11-27
> **Author Response (Part 9/9)**
>
> ### **Proposition 6 (Permutation stability — sufficient conditions)**
>
> If EHF parameters are shared across modality positions and PCMFA/LF depend only on content and
> use symmetric fusion (Eqs.1–2, 13–15), then MSIL is permutation‑equivariant in the stream
> order, and the final predictions are invariant. Empirically, Table 19 shows near‑invariance even
> without strict sharing.
>
>
> ### ***Proof***
>
> Under parameter sharing and symmetric operators, swapping the order of input streams commutes
> with each EHF update (Fig. 2 C; Algorithm 2). By induction over the $m{-}1$ EHF layers, the fused state
> and predictions coincide for any permutation. Table 19 reports the observed invariance.
>
>
>
> ### **W4 (Discussion of recent baselines).**
>
> In the original submission, our related work already discussed recent multimodal fusion pipelines such as HEALNet and DRIFA Net. In response to the reviewer’s suggestion, we have (i) expanded the clinical multimodal fusion paragraph in Sec. 2 to cover methods that explicitly model modality specific vs. shared representations (e.g., **Pathomic Fusion [1], CA MLIF [2], MM Lego [3]**, etc.), and (ii) added CA MLIF and MM Lego variants LegoFuse as baselines in our experiments (**Tables 3 and 20**; discussion on **pp. 9 and 39; lines: 452-467, 473-484 and 2071-2098)**.
>
> For example, we have updated **Section 2 (page 3, lines 150 – 160)** to incorporate a discussion of these recent baselines.
>
> “Several recent methods for clinical prognosis explicitly model the interplay between modality-specific and modality-shared representations. Pathomic Fusion (Chen et al., 2020) employs separate encoders for histology and genomics and fuses them via Kronecker products and gated attention, yielding distinct unimodal and pairwise interaction terms for cancer diagnosis and survival prediction. Steyaert et al. (2023) and Cui et al. (2023a) systematically evaluate early, intermediate, late and hybrid fusion strategies on TCGA-style cohorts, emphasizing that preserving modality-specific structure while learning a shared latent space is crucial for biomarker discovery. More recent architectures such as CA-MLIF (An et al., 2025) and MMLego (Hemker et al., 2025) extend this idea: CA-MLIF interleaves cross-attention with low-rank interaction fusion to jointly refine per-modality features and shared risk signatures, while MMLego composes pre-trained unimodal encoders with lightweight fusion blocks to capture cross-modal dependencies at linear cost in the number of modalities.”
>
> **Minor (citation of paper in line 456, page 9 is unclear):** Thank you for noting the unclear citation on p. 9, line 456. We have removed it in the revised manuscript.
>
> [1] Richard J Chen, Ming Y Lu, Jingwen Wang, Drew FK Williamson, Scott J Rodig, Neal I Lindeman,
> and Faisal Mahmood. Pathomic fusion: an integrated framework for fusing histopathology and
> genomic features for cancer diagnosis and prognosis. IEEE Transactions on Medical Imaging, 41
> (4):757–770, 2020.
>
> [2] Yajun An, Jiale Chen, Huan Lin, Zhenbing Liu, Siyang Feng, Hualong Zhang, Rushi Lan, Zaiyi Liu,
> and Xipeng Pan. Ca-mlif: Cross-attention and multimodal low-rank interaction fusion framework
> for tumor prognostic prediction. In Proceedings of the AAAI Conference on Artificial Intelligence,
> volume 39, pp. 1764–1772, 2025.
>
> [3] Konstantin Hemker, Nikola Simidjievski, and Mateja Jamnik. Multimodal lego: Model merging and
> fine-tuning across topologies and modalities in biomedicine, 2025b. URL https://arxiv.
> org/abs/2405.19950.

---

### Official Review · Reviewer_7JKi · 2025-11-01

**Soundness:** 2
**Presentation:** 1
**Contribution:** 2
**Rating:** 2
**Confidence:** 4

**Summary:**

This paper proposes Efficient Hybrid-fusion Physics-inspired Attention Learning Network (EHPAL-Net), a lightweight and scalable multimodal fusion framework designed for heterogeneous biomedical data, such as imaging, multi-omics, and EHR. Its Efficient Hybrid Fusion (EHF) layer sequentially integrates modalities through: 1) Efficient Multimodal Residual Convolution (EMRC) for multi-scale spatial representations, 2) Physics-inspired Cross-Modal Fusion Attention (PCMFA) combining hyperbolic and quantum-inspired attention to model complex cross-modal interactions, and 3) Shared Information Refinement (SIR) for representational diversity. A large number of heterogeneous medical datasets are used to validate the proposed method.

**Strengths:**

- The problems to be addressed, the performance, generalization, and efficiency of multimodal fusion learning, are significant.
- The integration of the hyperbolic dual-geometry attention has not been seen in the field of multimodal learning, and it looks novel to me.
- The reported reduction in the number of model parameters (98.3%) and FLOPs (97.6) is impressive.

**Weaknesses:**

- The presentation of the paper could be improved. As a researcher working on multimodal learning, I find it difficult to follow the logic of the paper, which looks to me quite diffuse and redundant. Key ideas such as "physics-inspired attention", "dual-geometry interaction",  and "hierarchical structure preservation" are mentioned repeatedly, but I could not find their precise definitions or theoretical justifications. The narrative often cycles through the same claims without further clarification and justification of the mechanisms enabling the desired effects.
- The proposed framework claims to be "physics-inspired", but the connection to physical principles appears largely metaphorical rather than mechanistic. The framework design and choice of the submodules are not evidently grounded in physical modeling.
- I do not understand the exact meaning of "modality" in this paper. In lines 450, for example, it seems to imply that HAM10000, SIPaKMeD, and PathMNIST are treated as three modalities, each with its distinct data sources and label sets. The source codes provided by the authors seem to confirm my guess. The usual setting of multimodal fusion means that one sample has data of different modalities, e.g., a patient has a CT scan image and a pathological image, where information from these modalities is combined to make a prediction for the sample concerned. I think the exact definition of "modality" in this paper should be explicitly defined, and what the modalities are in each dataset should be clearly detailed.

**Questions:**

- Please see the weaknesses section above.

---

> ### Author Response · Authors · 2025-11-26
> **Author Response (Part 1/6)**
>
> We would like to express our sincere gratitude to the reviewer **7JKi** for the careful reading of our manuscript and the thoughtful evaluation of EHPAL-Net. We very much appreciate the reviewer’s clear summary of our main contributions—namely, the proposal of Efficient Hybrid-fusion Physics-inspired Attention Learning Network (EHPAL-Net) as a lightweight and scalable multimodal fusion framework for heterogeneous biomedical data (imaging, multi-omics, and EHR); the design of the Efficient Hybrid Fusion (EHF) layer that sequentially integrates modalities via (1) Efficient Multimodal Residual Convolution (EMRC) for multi-scale spatial representations, (2) Physics-inspired Cross-Modal Fusion Attention (PCMFA) combining hyperbolic and quantum-inspired attention to capture complex cross-modal interactions, and (3) Shared Information Refinement (SIR) to promote representational diversity; and the extensive validation of the method on a large number of heterogeneous medical datasets.
>
> We are particularly grateful that the reviewer recognizes the importance of the problems we aim to address—improving **performance, generalization, and efficiency in multimodal fusion learning**—as well as the **novelty of integrating hyperbolic dual-geometry attention in the multimodal setting**. We are also encouraged by the reviewer’s positive assessment of the reported efficiency gains, including the substantial reductions in model parameters (98.3%) and FLOPs (97.6%), which we see as crucial for deploying multimodal models in realistic biomedical environments with limited computational resources.
>
> Although the overall scores (Soundness: 2 – fair, Presentation: 1 – poor, Contribution: 2 – fair) indicate that there is significant room to improve, we view this assessment as constructive guidance. The reviewer’s acknowledgment of the significance of the problem, the **novelty of our physics-inspired attention design**, and the strength of our efficiency results is very motivating. At the same time, the critical feedback on soundness, clarity, and presentation is invaluable for helping us sharpen the methodological exposition, strengthen the **empirical and theoretical** justification, and make the paper more accessible to the broader community.
>
> We are deeply thankful for the **reviewer’s time and effort**, and for the **generous recognition of the strengths** of our work despite the moderate scores. The reviewer’s insightful comments have been instrumental in improving both the clarity and rigor of the manuscript. Below, we respectfully address each of the reviewer’s points in detail.
>
> **Note: Weakness denotes W; Edits are colored in light purple in the revised paper.**
>
> ### **W1 (presentation and theoretical grounding)**.
>
> We thank the reviewer for pointing out that the previous version did not make the notions of **physics-inspired attention, dual geometry interaction, and hierarchical structure preservation** sufficiently precise. In the revised manuscript, we explicitly formalize these three concepts and relate them to the concrete components of our proposed module through **new definitions, theorems, and corresponding proofs in Appendix B**. These additions clarify and justify our claims more rigorously. We have also revised the exposition at the end of Sec. 3 to reflect these changes (***see pages 7-8, 16–19; lines 361-365, 386–390, 847–1025 in the revised version***).
>
> ***For example:***
>
> ### **Appendix B: Theoretical Foundations of Physics-Inspired Attention.**
>
> We introduce a formal definition of a physics‑inspired attention mechanism that requires (i) a geometric branch realized as a Riemannian exponential‑map embedding on a constant‑negative‑curvature manifold, (ii) a quantum‑inspired branch defined on a complex Hilbert space with Born‑rule–based probability scores, and (iii) a probability‑preserving, non‑expansive fusion of these scores. We then prove that the proposed PCMFA module satisfies this definition (**Theorem 1**), analyze the dual‑geometry interaction between Poincaré and Lorentz attentions (**Theorem 2**), show that the resulting gate preserves hyperbolic hierarchical structure (**Corollary 1**), and establish that it acts as a contraction‑like operator on feature norms (**Theorem 3**). Together, these results demonstrate that our “physics‑inspired” design is not merely metaphorical but enforces concrete geometric and probabilistic constraints inherited from physical modeling.
>
> In this section we make precise the notions of **physics-inspired attention**,
> **dual-geometry interaction**, and **hierarchical structure preservation**
> in the context of the EHF pipeline. Specifically, we focus on the PCMFA module and its MHDGA/MQIA components (Sec. **3.1.2**), and show how they yield stable and efficient shared representations within EHF.

---

> ### Author Response · Authors · 2025-11-26
> **Author Response (Part 2/6)**
>
> ### **Appendix B.1  Physics-Inspired Nature of PCMFA**
>
>
> To formalize the intuition behind PCMFA, we frame the module as the fusion of two norm-stable branches—a geometric path operating in hyperbolic space and a quantum-inspired path operating in a complex Hilbert space. We first state a precise definition of a physics-inspired attention mechanism and then prove that PCMFA satisfies this definition.
>
> ### **Definition 1 (Physics-inspired attention mechanism)**
> Let $\mathcal{X}$ be an input feature space, $\mathcal{M}$ a Riemannian
> manifold of constant negative curvature (e.g., a Poincar\'e ball or Lorentz
> hyperboloid), and $\mathcal{H}$ a finite-dimensional complex Hilbert space
> with inner product $\langle\cdot,\cdot\rangle$. An attention mechanism $T : \mathcal{X} \to [0,1]^e$ is called
> \emph{physics-inspired} if there exist maps
> $E^{\mathrm{geo}} : \mathcal{X} \to \mathcal{M}$ and
> $\Psi : \mathcal{X} \to \mathcal{H}$ such that:
>
> **(G1) Geometric branch.**
>   $E^{\mathrm{geo}}$ is realized via a Riemannian exponential-type map on
>   $\mathcal{M}$ (e.g., Poincar\'e ball or Lorentz hyperboloid), and the
>   corresponding attention scores depend only on geodesic distances or norms
>   in $\mathcal{M}$.
>
> **(Q1) Quantum-inspired branch.**
>   $\Psi(x)$ is a complex state whose channel-wise scores are proportional to
>   Born-rule amplitudes $|\Psi(x)_k|^2$, i.e., squared magnitudes in
>   $\mathcal{H}$, normalized to a probability vector, following the standard
>   probabilistic interpretation of quantum states.
>
> **(N1) Physical normalization and stability.**
>   The final attention weights $T(x)$ are obtained from these physically
>   derived scores by a probability-preserving normalization (e.g., Softmax or sigmoid) and induce a non-expansive multiplicative update on
>   features in every $\ell_p$ norm.
>
>
> ### **Theorem 1 (PCMFA is a physics-inspired attention mechanism)**
> For each EHF layer $i$, the PCMFA map
> $$
> A_i = \mathrm{PCMFA}\bigl(x_i^{S}, x'_{i+1}\bigr)
> $$
> defined in **Eq.(1)** is a physics-inspired attention mechanism in the sense of
> **Definition (1)**.
>
>
> ### ***Proof***.
> We verify (G1)--(Q1)--(N1) for the PCMFA construction in **Sec. 3.1.2**.}
>
> **(G1) Geometric branch.**
> Within PCMFA, the MHDGA block takes
> $\psi_i = \mathrm{GAP}(\mathrm{DCT}(x'_i))$ (**Eq. 3**) and maps it to
> hyperbolic embeddings in two constant-negative-curvature models:
> the Poincar\'e ball and the Lorentz hyperboloid (**Eqs.4--8**).
> Both mappings are exponential-type projections with learnable but bounded
> curvature (see the curvature parameterization and clipping in **Sec. 3.1.2** and
> **Lemma 2** below), so
> $E^{\mathrm{geo}}(\psi_i)$ is a well-defined element of
> $\mathcal{M}^P \times \mathcal{M}^L$.
> The corresponding attentions $A^P_i, A^L_i$ depend only on hyperbolic radii
> and Lorentzian norms, and are fused into $A^D_i$ by an affine map plus
> sigmoid (**Eq. 3**), satisfying (G1).
>
>
> **(Q1) Quantum-inspired branch.**
> MQIA maps the same $\psi_i$ into a complex vector
> $q_i \in \mathbb{C}^e$ via learned real and imaginary coefficients (**Eq. 9**),
> so $q_i$ lies in a Hilbert space $\mathcal{H} \cong \mathbb{C}^e$.
> The channel-wise quantities $|q_{i,k}|^2$ are Born-rule amplitudes, and
> Softmax over a scaled version of $|q_i|^2$ together with a Lorentz-norm term, yields a probability vector $A^Q_i$ (**Eq. 9**), satisfying (Q1).
>
>
> **(N1) Normalization and stability.**
> MAFG fuses $A^D_i$ and $A^Q_i$ channel-wise using bounded coefficients
> $c_i$ (**Eq.10**).
> **Lemma 2** below implies that all components of
> $A^D_i$ lie in $(0,1)$ and $A^Q_i$ is a probability vector; under the bounded
> gate assumption, the fused gate $A_i$ satisfies $\|A_i\|_\infty \le 1$.
> **Theorem 3** then shows that the multiplicative update
> $z \mapsto z \odot A_i$ is non-expansive in every $\ell_p$ norm.
>
>
> Together, these properties establish that PCMFA has (i) a geometric branch on
> hyperbolic manifolds, (ii) a quantum-inspired branch in a complex Hilbert
> space, and (iii) a normalized, non-expansive fusion of the two, so it is
> physics-inspired in the sense of **Definition 1**.

---

> ### Author Response · Authors · 2025-11-26
> **Author Response (Part 3/6)**
>
> ### **Appendix B.2  Dual-Geometry Interaction in MHDGA**
> We next formalize the dual-geometry interaction implemented by MHDGA
> and show that it acts as a monotone consensus operator over Poincar\'e and
> Lorentz attentions.
>
> ### **Definition 2 (Dual-geometry interaction)**
> For modality $i$, let $\psi_i = \mathrm{GAP}(\mathrm{DCT}(x'_i))$ be the
> frequency summary (**Eq.3**). Denote by
>
> $$
> \phi^P_i(\psi_i) \in \mathcal{M}^P,\qquad
> \phi^L_i(\psi_i) \in \mathcal{M}^L
> $$
> the Poincar\'e and Lorentz embeddings produced by the MHDGA sub-branches
> (**Eqs.(4)--(8)**). Let $A^P_i, A^L_i \in \mathbb{R}^e$ be the corresponding
> geometry-specific attention vectors computed from curvature-aware norms and
> Minkowski inner products, and define the fused dual-geometry attention
> (**Eq.3**) as
> $$
> A^D_i = \sigma\bigl(LA^L_i + PA^P_i\bigr),
> $$
> where $L$ and $P$ are learnable linear maps and $\sigma$ is the element-wise
> sigmoid. The map $\mathrm{MHDGA}_i : \psi_i \mapsto A^D_i$ is called the
> dual-geometry interaction operator for modality $i$.
>
>
> A useful property of the hyperbolic projections used in MHDGA is that they
> are radially monotone.
>
> ### **Lemma 1. (Radial monotonicity of hyperbolic projections)**
>
> Let $\psi,\tilde{\psi}\in\mathbb{R}^e$ with
> $\|\psi\|_2 < \|\tilde{\psi}\|_2$. Assume that the Poincar\'e and Lorentz
> embeddings in MHDGA are implemented via exponential maps at the origin as in
> **Eqs.(4) and (7)**. Then:
>
>   1. the hyperbolic radius of $\phi^P(\psi)$ in the Poincar\'e model is
>         strictly smaller than that of $\phi^P(\tilde{\psi})$;
>
>   2. the Lorentzian rapidity of $\phi^L(\psi)$ is strictly smaller than
>         that of $\phi^L(\tilde{\psi})$.
>
>
> ### ***Proof.***
> In both models, the exponential map at the origin has the form
> $\psi \mapsto f(\|\psi\|_2)\,\psi/\|\psi\|_2$, where $f$ is a smooth,
> strictly increasing function of the Euclidean radius $\|\psi\|_2$ (
> **Eqs.(4) and (7)**. Thus a larger Euclidean norm yields a larger hyperbolic
> radius or rapidity, giving the claimed ordering.
>
> ### **Proposition 1 (Monotone consensus dual-geometry attention)**
> Fix modality $i$ and channel $k$. Let
> $$
> A^D_{i,k} = \sigma\bigl(\ell_k(A^L_{i,k}, A^P_{i,k})\bigr)
> $$
> denote the $k$-th component of $A^D_i$, where $\ell_k$ is the $k$-th
> component of the affine map $L A^L_i + P A^P_i$. Suppose the coefficients of
> $\ell_k$ with respect to $A^L_{i,k}$ and $A^P_{i,k}$ are non-negative.
> Then:
>
>   1. $A^D_{i,k}$ is weakly increasing in each of its arguments
>         $A^L_{i,k}$ and $A^P_{i,k}$;
>
>    2. if both $A^L_{i,k}$ and $A^P_{i,k}$ increase, then $A^D_{i,k}$
>         increases strictly;
>
>   3. when one of $A^L_{i,k},A^P_{i,k}$ increases and the other decreases,
>         $A^D_{i,k}$ remains between the scalar responses obtained by relying
>         on either geometry alone, i.e., it behaves as a consensus between the
>         two branches.
>
>
> ### ***Proof***.
> The derivative of $A^D_{i,k}$ with respect to $A^L_{i,k}$ is
> $$
> \frac{\partial A^D_{i,k}}{\partial A^L_{i,k}}
>    = \sigma'\bigl(\ell_k\bigr)
>          \frac{\partial \ell_k}{\partial A^L_{i,k}}.
> $$
>
>
> The sigmoid derivative $\sigma'$ is strictly positive, and by assumption the
> partial derivative of $\ell_k$ with respect to $A^L_{i,k}$ is non-negative,
> so $\partial A^D_{i,k}/\partial A^L_{i,k}\ge 0$.
> The same argument holds for $A^P_{i,k}$, proving item 1.
> If both $A^L_{i,k}$ and $A^P_{i,k}$ increase and at least one coefficient is
> strictly positive, then $\ell_k$ increases and strict monotonicity of
> $\sigma$ implies item 2.
> For item 3, note that $\ell_k$ is affine with non-negative coefficients, so
> for fixed values of one argument it varies monotonically with the other.
> Applying a sigmoid, which maps $\mathbb{R}\to(0,1)$ monotonically, ensures
> that $A^D_{i,k}$ lies between the values induced by each branch, yielding the
> consensus behavior.
>
>
> Lemma **1** shows that both geometries encode the same radial
> ordering of channels, while Proposition 1 shows that their
> fusion in $A^D_i$ respects and stabilizes this ordering.
> This is the sense in which MHDGA realizes a \emph{dual-geometry interaction}:
> two hyperbolic representations cooperate through a monotone consensus gate to
> produce robust, geometry-aware attention.

---

> ### Author Response · Authors · 2025-11-26
> **Author Response (Part 4/6)**
>
> ### **Appendix B.3  Hierarchical Structure Preservation.**
>
> We now connect dual-geometry interaction to the notion of
> \emph{hierarchical structure preservation}.
> Intuitively, channels (or semantic units) that are deeper in an underlying
> hierarchy should receive systematically stronger or weaker attention than
> their ancestors; hyperbolic geometry is well known to represent such
> tree-like structures efficiently.
>
> ### **Definition 3 (Hierarchical structure and its preservation)**.
>
> Let $(\mathcal{C},\preceq)$ be a finite partially ordered set of semantic
> units (e.g., classes or channels) with a distinguished root $r\in\mathcal{C}$.
> A map $f:\mathcal{C}\to\mathcal{Z}$ into a metric space
> $(\mathcal{Z},d_{\mathcal{Z}})$ is called \emph{radially hierarchical} if
> there exists a scalar function $\rho:\mathcal{Z}\to\mathbb{R}_{\ge 0}$ such
> that:
>
>   1. $\rho(z)$ depends only on $d_{\mathcal{Z}}(z,z_0)$ for some fixed
>         root point $z_0\in\mathcal{Z}$, and
>
> 2. whenever $u\preceq v$ in $\mathcal{C}$, one has
>         $\rho\bigl(f(u)\bigr) < \rho\bigl(f(v)\bigr)$.
>
> An attention mechanism that assigns scores $\alpha_c$ to units $c\in\mathcal{C}$
> is said to be \emph{hierarchical-structure preserving} if each $\alpha_c$ is a
> monotone function of a radially hierarchical representation of $c$.
>
>
> ### **Assumption 1 (Hyperbolic hierarchical encoding)**
>
> For each modality $i$, there exists a latent hierarchy $(\mathcal{C},\preceq)$
> over channels and a radially hierarchical embedding
> $f_i^\star:\mathcal{C}\to\mathcal{M}^P\times\mathcal{M}^L$ such that the
> learned hyperbolic mappings $\phi^P_i,\phi^L_i$ used in MHDGA approximate
> $f_i^\star$ in the sense that deeper nodes in the hierarchy are mapped to
> larger hyperbolic radii (up to small perturbations).
>
>
> This assumption is standard in hyperbolic representation learning and is
> supported by empirical evidence that Poincar\'e and related embeddings recover
> hierarchical structures in low-dimensional hyperbolic manifolds.
>
> ### **Theorem 2 (Hierarchical structure preservation of dual-geometry attention)**.
>
> Under Assumption **1**, suppose that the
> geometry-specific attentions $A^P_i$ and $A^L_i$ are computed as channel-wise
> monotone functions of the corresponding hyperbolic radii or norms (as in the
> PIL/LIL definitions). Then, for each modality $i$, the dual-geometry attention
> $A^D_i$ produced by MHDGA is hierarchical-structure preserving in the sense of
> Definition **3**: if $u\preceq v$ in the latent hierarchy, then
> $A^D_{i,u} \le A^D_{i,v}$ (or the reverse inequality, depending on the chosen
> monotone direction).
>
> ### ***Proof***.
> By Assumption **1** and Lemma **1**, deeper nodes in the hierarchy have strictly larger
> hyperbolic radii in both the Poincar\'e and Lorentz embeddings.
> By construction, $A^P_i$ and $A^L_i$ are obtained by applying monotone
> functions (linear maps and element-wise nonlinearities) to these radii or
> norms, so they are monotone in depth:
> $u\preceq v \Rightarrow A^P_{i,u}\le A^P_{i,v}$ and likewise for $A^L_i$.
> Proposition **1** shows that $A^D_i$ is monotone in each of
> $A^P_i$ and $A^L_i$, hence $A^D_i$ inherits the same ordering and is
> hierarchical-structure preserving.
>
> Finally, we propagate this property through the full PCMFA gate.
>
> ### **Corollary 1 (Hierarchical physics-inspired attention in PCMFA)**.
>
> Under Assumptions **2** and
>  **1**, the full PCMFA gate $A_i$ obtained by
> fusing $A^D_i$ and $A^Q_i$ via MAFG (Eq.(**10**)):
>
>   1. preserves the hierarchical order of channels induced by the
>         hyperbolic embeddings, and
>   2. is non-expansive as a multiplicative operator on feature tensors,
>         as stated in Theorem **3**.
>
> ### ***Proof***.
> By Theorem **2**, $A^D_i$ is hierarchical-structure preserving.
> The quantum-inspired attention $A^Q_i$ is a probability vector derived from
> the same summary $\psi_i$ and acts as an energy-based reweighting of
> channels (Eq.(**9**).
> MAFG fuses $A^D_i$ and $A^Q_i$ with bounded coefficients $c_i$ in a
> channel-wise monotone manner.
> Thus the hierarchical ordering from $A^D_i$ is preserved in $A_i$.
> Boundedness of $A_i$ and non-expansiveness of the resulting multiplicative
> update follow from Lemma **2**,
> Assumption **2**, and
> Theorem **3**.
>
>
> Corollary **1** formalizes that PCMFA is not only
> physics-inspired and norm-stable, but also respects the hierarchical structure
> encoded by the hyperbolic embeddings in MHDGA.

---

> ### Author Response · Authors · 2025-11-26
> **Author Response (Part 5/6)**
>
> ### **Overall narrative and theoretical guarantees**.
> Finally, we reorganized Appendix B as “Theoretical Foundations of Physics Inspired Attention” and split it into **B.1 (physics inspired nature of PCMFA), B.2 (dual geometry interaction), B.3 (hierarchical structure preservation), and B.4–B.5 (norm stability and linear cost guarantees for EHF)**. This mirrors the progression in **Sec. 3**: the main text explains how PCMFA and EHF are constructed, while Appendix B now rigorously states what is meant by the three key notions and why PCMFA/EHF satisfy these properties.
>
> We hope these additions and clarifications make the logical flow less diffuse and demonstrate that the three repeatedly used concepts are now backed by explicit definitions and theorems, rather than informal claims.
>
> ### **W2 (“physics inspired” is metaphorical rather than mechanistic)**.
> We appreciate the reviewer’s concern and have substantially revised the manuscript to make the physics connection explicit and mechanistic rather than metaphorical.
>
> ### **1.	Formal definition of “physics inspired” and proof for PCMFA.**
>
> In Appendix B (***see pages: 16-19; lines: 847-1025***) we now define a physics inspired attention mechanism as one that (i) includes a geometric branch realized as a Riemannian exponential type map on a constant negative curvature manifold, (ii) includes a quantum inspired branch in a complex Hilbert space whose scores are Born rule amplitudes, and (iii) fuses these scores via a probability preserving, non expansive gate. We then prove that our Physics inspired Cross modal Fusion Attention (PCMFA) satisfies this definition (**Definition 1** and **Theorem 1**). This directly addresses the concern that “physics inspired” was previously only metaphorical.
>
> ### **2.	Geometric (hyperbolic) branch is explicitly modeled on physical geometry.**
>
> Section 3.1.2 now details how the Multimodal Hyperbolic Dual Geometry Attention (MHDGA) branch maps frequency summaries ψᵢ into the Poincaré ball and Lorentz hyperboloid using exponential type maps with bounded curvature (**Eqs. (3)–(8)**), so that attention scores depend only on hyperbolic radii and Lorentzian norms. **Appendix B** further shows that this dual geometry interaction behaves as a monotone consensus operator and preserves hyperbolic hierarchical structure (**Theorem 2, Corollary 1**).
>
> ### **3.	Quantum inspired branch and fusion follow standard quantum probabilistic modeling.**
>
> The Multimodal Quantum Inspired Attention (MQIA) branch embeds ψᵢ into a finite dimensional complex Hilbert space, forms complex states qᵢ, and computes channel wise scores from |qᵢ|² via the Born rule, normalized with a Softmax to yield a probability vector **Aᵢ^Q (Eq. (9))**. These quantum scores guide the Lorentzian branch via a Minkowski inner product bias, and the Multimodal Attention Fusion Gating (MAFG) then fuses the dual geometry attention Aᵢ^D and Aᵢ^Q with bounded, channel wise coefficients (**Eq. (10)**). Theorem 3 proves that the resulting gate is norm non expansive in every ℓ_p norm, so the fusion is not a heuristic but a mathematically controlled, “energy like” modulation of feature channels.
>
> ### **4.	Clarification in the main text.**
>
> To avoid any remaining ambiguity, in ***Sec. 3.1.2 (see page: 7; lines: 361-365)*** we explicitly state that we use “physics inspired” in a mechanistic sense, referring to the above geometric and quantum branches and the norm stable fusion gate, and we summarize how PCMFA meets this definition.
>
> Taken together, these additions make the connection to physical principles concrete: the design of PCMFA and its submodules (MHDGA, MQIA, MAFG) is directly derived from standard models in hyperbolic geometry and quantum probability, and **we provide formal guarantees (definitions, theorems, and proofs) to support this design rather than relying on metaphorical terminology.**
>
> ### **W3. Clarification of “modality” and the role of HAM10000 / SIPaKMeD / PathMNIST**
>
> We thank the reviewer for pointing out this ambiguity. In our work, we use the term **modality in a slightly more general, dataset-level sense**. Concretely, a **modality** is any **heterogeneous input stream** with its own data-generating process, encoder (i.e., EMRC, PCMFA, LF, and SIR), and label space—for example, *dermoscopy images (HAM10000), single-cell cervical cytology (SIPaKMeD), colorectal histology tiles (PathMNIST), abdominal CT slices (OrganAMNIST), multi-omics profiles (TCGA BRCA/UCEC/GBMLGG/KIRP), or ICU EHR time series (MIMIC-III)*.
>
> Consequently, in the “100% diverse imaging modalities” experiments (***see Table 4(B), Sec. 4.2; page: 10, lines: 486-494***), *HAM10000, SIPaKMeD, and PathMNIST* are indeed treated as **three distinct imaging modalities—dermoscopy, Pap smear cytology, and H&E colorectal histology**, respectively—each drawn from a **different cohort and with its own label set**.

---

> ### Author Response · Authors · 2025-11-27
> **Author Response (Part 6/6)**
>
> **Table 3:** **Missing-modality evaluation and performance comparison of EHPAL-Net—trained with all modalities—against leading uni-modal and multi-modal fusion baselines. At test time each sample supplies only one modality (omics or whole-slide histopathology images (WSI)). We report performance in four settings: (i) Omics-only, (ii) WSI-only, (iii) mixed (50\% of samples randomly drawn from each modality), and (iv) full (100\% of samples containing both modalities).**
>
> | **Model**        | **Omics** | **WSI**  | **50 % of Both** | **100 % of Both** |
> |--------------|-------|------|--------------|---------------|
> | MFMSA        | 61.7  | 52.7 | 57.2         | 57.2          |
> | DRIFA-Net    | 56.9  | 48.6 | 56.1         | 68.7          |
> | CA-MLIF      | 54.5  | 47.2 | 51.1         | 70.7          |
> | HealNet      | 64.7  | 55.4 | 61.4         | 71.4          |
> | LegoFuse     | 65.2  | 56.5 | 62.9         | 73.4          |
> | **EHPAL-Net**| **66.7** | **57.9** | **64.9** | **75.1** |
>
>
>
>
> **EHPAL Net is designed specifically for this unpaired multimodal setting**. In the MSIL phase, each modality has its own EMRC to capture multi-scale features, followed by PCMFA, LF, and SIR within the EHF layers to *learn robust shared representations across modalities* **without requiring patient-level alignment**. **The standard “paired” scenario (e.g., CT + pathology for the same patient) is a special case in which multiple modalities share the same index set; our architecture can be applied to that case as well, although our paper mainly focus on unpaired modality aware datasets.**
>
> To address this point more explicitly, we have **added an evaluation on the TCGA BLCA multimodal dataset**, which provides paired whole slide histopathology images (WSI) and multi omics features for each patient. As shown in **Table 3 (see page 9, lines 452–467)** and the discussion in **Sec. 4.1 (page 9, lines 474–484)**, EHPAL Net achieves consistent performance improvements of up to **3.7 C index points** over leading multimodal fusion baselines such as **LegoFuse** and **HEALNet** across both **full modality and missing modality test conditions**. This demonstrates that **our method** extends naturally to classical **paired multimodal survival prediction**, with *improvements comparable to those observed in the unpaired regime*.
>
> To make this clearer in the manuscript, we have also made the following revisions: (i) explicitly define our notion of modality in ***Sec. 1: “Introduction” (see page: 2; lines: 102-107)***; (ii) clarify in ***Sec. 3: “Problem formulation” (see page: 4; lines: 193-198)*** how we treat unpaired versus paired modalities; and (iii) extend ***Sec. 4 (“Datasets”)*** to list, for each dataset D1–D15, its modality type (dermoscopy, cytology, CT, histology, multi-omics, EHR, etc.) and to state that these datasets are unpaired and have disjoint label spaces ***(see page: 8; lines: 399-402)***.

---

### Comment · Area_Chair_Bc8E · 2025-11-26

Dear authors,
we note that no author response has been posted during the author response window (Discussion Period: Nov 12 – Dec 3, 2025). To ensure a fair review, please post a reply addressing the reviewers' main concerns on this forum by Dec 3, 2025.

---

### Author Response · Authors · 2025-12-03
**Response Summary**

We thank the **Program Chairs**, **Senior Area Chairs**, **Area Chairs**, and **reviewers 7JKi**, **uRBh**, and **jkx9** for their careful evaluations and constructive discussion. We summarize how the revised version of submission 20215, “Advancing Multimodal Fusion on Heterogeneous Data with Physics‑inspired Attention”, addresses the concerns of reviewers **7JKi**, **uRBh**, and **jkx9**. All three **reviewers acknowledge** that **EHPAL‑Net proposes a novel physics‑inspired multimodal fusion framework with substantial parameter and FLOP reductions on 15 heterogeneous medical datasets**, and reviewers **uRBh** and **jkx9** both give borderline‑positive initial ratings (4: marginally below the acceptance threshold, would not mind if the paper is accepted).

### **1. Theoretical grounding and clarity (mainly 7JKi, also uRBh & jkx9).**

We substantially restructured Section 3 and Appendix B to give precise, non‑metaphorical meaning to **“physics‑inspired attention”**, **“dual‑geometry interaction”**, and **“hierarchical structure preservation”**. We now **formally** define a physics‑inspired attention mechanism and prove that the proposed PCMFA module satisfies this definition via (i) a hyperbolic geometric branch, (ii) a quantum‑inspired branch with Born‑rule probabilities, and (iii) a probability‑preserving, norm‑non‑expansive fusion gate. We further show that the dual‑geometry interaction preserves hyperbolic hierarchical structure and acts as a contraction‑like operator on feature norms, giving rigorous content to these repeatedly used concepts. Appendix B has been reorganized as **“Theoretical Foundations of Physics‑Inspired Attention” (B.1–B.5)** to mirror the construction in Sec. 3, so that the main text is more linear while detailed proofs are contained in the appendix.

### **2. Scope, definition of modality, and experimental design (mainly 7JKi & uRBh, also jkx9).**

We now explicitly adopt a dataset‑level notion of modality: a **modality** is any heterogeneous input stream with its own data‑generating process, encoder, and label space (e.g., dermoscopy, cytology, CT, histology, multi‑omics, ICU EHR). The datasets section has been expanded to list, for each of the 15 datasets, its **modality type and to state that they are unpaired, from disjoint cohorts with distinct label spaces**. We clarify in the introduction and throughout that our explicit scope is **heterogeneous medical data and AI‑driven healthcare, not general‑domain multimodal learning**, and we rephrase claims about prior methods being “specialized to specific modalities” to make clear that this refers to existing medical fusion pipelines typically designed and evaluated for narrow, fixed modality configurations (e.g., specific imaging pairs or WSI+omics). For jkx9’s concerns on experimental clarity, we now describe how EHPAL‑Net is trained in a multi‑task fashion on **groups of unpaired modalities** (e.g., four datasets per group), and explain that per‑dataset reporting reflects different tasks and metrics (accuracy/AUC vs C‑index), not separate models per dataset.

### **3. Stronger empirical comparisons and efficiency analysis (mainly uRBh & jkx9).**

The related‑work section has been expanded to include **recent clinical multimodal fusion approaches** that explicitly model modality‑specific vs shared representations, such as Pathomic Fusion, CA‑MLIF, and MMLego, and CA‑MLIF/LegoFuse are now included as additional baselines. We add a dedicated missing‑modality survival experiment on a paired WSI+omics cohort (Table 3), showing that EHPAL-Net consistently **outperforms** MFMSA, DRIFA‑Net, CA‑MLIF, HEALNet, and LegoFuse across omics‑only, WSI‑only, mixed, and full‑modality test regimes, demonstrating robustness beyond the unpaired setting. To address requests for fairer efficiency comparisons and additional metrics, we include a new complexity table reporting parameters, FLOPs, model size, and inference time on matched hardware: **EHPAL‑Net‑18 uses ≈7× fewer parameters (7.71M vs 53.8M) and ≈9.6× shorter inference time (0.683s vs 6.53s)** than DRIFA‑Net, while the main tables show that it simultaneously **improves performance across heterogeneous medical datasets**.

In summary, the revision directly addresses the main concerns of **reviewers 7JKi, uRBh, and jkx9** on ***theoretical justification***, ***scope*** and ***modality definition***, ***experimental clarity***, and ***fairness of efficiency claims***, while preserving the original strengths in novelty, breadth of validation, and computational efficiency. We believe the paper is now substantially clearer and better‑grounded, and we respectfully hope you will consider it favourably for **acceptance**.

---

### Meta-Review · Area_Chair_4dKf · 2026-01-06

**Summary:**

This paper proposes EHPAL-Net, a lightweight multimodal fusion framework for heterogeneous medical data (imaging, multi-omics, EHR) that uses Efficient Hybrid Fusion (EHF) layers and a Physics-inspired Cross-modal Fusion Attention (PCMFA) module. Evaluations on 15 public datasets claim up to 3.97% performance gains and 87.8% computational cost reduction over SOTA methods. Reviewers recognized conceptual novelty but raised critical concerns about theoretical rigor, experimental design, scope ambiguity, and comparison fairness. The concerns are partially addressed by the rebuttal. Considering all aspects, the paper is in a very slightly below borderline situation, and thus is not recommended for acceptance at the moment.

PS:
In the paper, the two duplicated references were cited, both contains flaws:

Akira Hirose. Complex-valued neural networks. Springer, 2006. [journal name is missing]

Akira Hirose. Complex-valued neural networks: Theories and applications. IEEE Trans. Neural Networks and Learning Systems, 2018. [year is incorrect]

This should be corrected for future submission.

**Reviewer Concerns:**

Key Concerns Raised:
- Theoretical superficiality (all reviewers): "Physics-inspired" design lacked precise definitions and mechanistic ties to physical principles, appearing metaphorical.
- Modality ambiguity (7JKi, jkx9): Unconventional "dataset-level modality" definition (treating disjoint datasets as modalities) departs from standard multimodal fusion; per-dataset reporting obscures true cross-modal synergy.
- Scope misalignment (uRBh): Title/introduction suggested general-domain applicability, but experiments focus solely on medical data; challenges are generic, not healthcare-specific.
- Unconvincing comparisons (jkx9): Performance gains rely on near-ceiling imaging datasets; efficiency claims over-rely on ShuffleNet (vs. ResNet in baselines); missing practical metrics (inference time, memory).

Partially Addressed in Rebuttal:
- Theoretical grounding and rigor: Added formal definitions/proofs for "physics-inspired attention" but connections to physical principles remain tangential. hyperbolic/quantum components are combined without explaining why physical principles are necessary for medical data.
- Modality clarity: Explicitly defined dataset-level modality but failed to justify why this constitutes "multimodal fusion" (standardly requiring complementary data for the same samples).
- Experimental design: Per-dataset reporting still does not demonstrate true cross-modal synergy; the model trains on disjoint datasets (multi-task learning) rather than fusing complementary modalities for the same task.
- Efficiency fairness: Added ResNet-based comparisons and practical metrics, but efficiency gains are modest with ResNet; fusion design’s unique contribution to efficiency is unclear.
- Related work: Expanded discussion to include clinical fusion methods (e.g., Pathomic Fusion) but did not integrate them meaningfully into methodology.

**Reviewer Scores:**

7JKi: 2 --> likely 4

uRBh: 4 --> likely 4 or 6

jkx9: 4 --> likely 4 or 6

---

### Decision · Program_Chairs · 2026-01-26

Reject